# Large Language Models are Demonstration Pre-Selectors for Themselves

**Jiarui Jin** [1 2 * †] **Yuwei Wu** [3 *] **Haoxuan Li** [4] **Xiaoting He** [5] **Weinan Zhang** [1] **Yiming Yang** [3] **Yong Yu** [1]
**Jun Wang** [6] **Mengyue Yang**[⊠ 7]

## Abstract

In-context learning (ICL) with large language models (LLMs) delivers strong few-shot performance by choosing few-shot demonstrations from the entire training data. However, existing ICL methods, which rely on similarity or diversity scores to choose demonstrations, incur high computational costs due to repeatedly retrieval from large-scale datasets for each query. To this end, we propose FEEDER (FEw yet Essential Demonstration prE-selectoR), a novel *pre-selection* framework that identifies a core subset of demonstrations containing the most representative examples in the training data, tailored to specific LLMs. To construct this subset, we introduce the "sufficiency" and "necessity" metrics in the pre-selection stage and design a tree-based algorithm to identify representative examples efficiently. Once pre-selected, this representative subset can effectively replace the full training data, improving efficiency while maintaining comparable performance in ICL. Additionally, our pre-selected subset also benefits fine-tuning LLMs, where we introduce a bi-level optimization method that enhances training efficiency without sacrificing performance. Experiments with LLMs ranging from 300M to 8B parameters show that FEEDER can reduce training data size by over 20% while maintaining performance and seamlessly integrating with various downstream demonstration selection strategies in ICL.

## 1. Introduction

Large language models (LLMs), e.g., GPT (Brown et al., 2020), Gemma (Team et al., 2024), and Llama (Touvron et al., 2023), have demonstrated impressive performance across a wide range of tasks by employing few-shot inference, referred as in-context learning (ICL) (Brown et al., 2020; Dong et al., 2022). This approach avoids the computational expense associated with fine-tuning LLMs. Here, the core challenge is how to select the most representative demonstrations from large training data.

Early approaches (Qiu et al., 2022; Liu et al., 2021; Rubin et al., 2021; Wang et al., 2022) primarily selected demonstrations based on relevance, typically using similarity scores between each demonstration and the input question. However, recent studies (Levy et al., 2022; Köksal et al., 2022; Zhou et al., 2023) indicate that evaluating examples in isolation is suboptimal. Instead, they advocate for incorporating additional selection criteria, such as diversity, uncertainty, or clustering-based metrics, alongside similarity, to enhance demonstration selection effectiveness. As a result, the above enhancements in the demonstration selection process introduce significant computational overhead, particularly when the number of shots is large. Beyond efficiency, we argue that the effectiveness of selected demonstrations should also consider the specific LLM in use, as different LLMs exhibit varying capabilities and knowledge domains.

To this end, we introduce a new stage called the pre-selection stage and propose a demonstration pre-selector named FEEDER (FEw yet Essential Demonstration prE-selectoR). FEEDER functions as a representative subset selector, aiming to identify a representative subset from the full training data before the demonstration selection process, thereby enhancing both efficiency and effectiveness in demonstration selection. For this purpose, we introduce the concepts of "sufficiency" and "necessity", where sufficiency investigates whether incorporating a demonstration is representative of other samples, while necessity examines whether a demonstration provides redundant information, taking into account the LLM's capability, knowledge, and the selected samples. By directing LLMs to focus on the selected subset, FEEDER prevents them from processing unnecessary data, thereby improving efficiency. Since exhaustively enumerating and

---

*Equal contributions: Jiarui Jin and Yuwei Wu. † Work done during Jiarui Jin's visit at University College London. [1]Shanghai Jiao Tong University [2]Xiaohongshu Inc. [3]Carnegie Mellon University [4]Peking University [5]No Affiliation [6]University College London [7]University of Bristol. Correspondence to: Mengyue Yang <mengyue.yang@bristol.ac.uk>.

*Proceedings of the 42nd International Conference on Machine Learning*, Vancouver, Canada. PMLR 267, 2025. Copyright 2025 by the author(s).

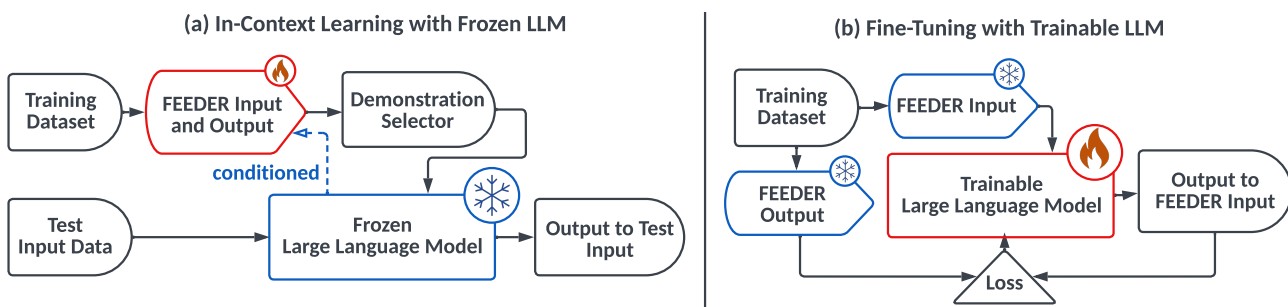

*Figure 1.* Overview of FEEDER that operates effectively within both in-context learning and fine-tuning settings. In the in-context learning setting, depicted in (a), we first *pre-select* a representative subset termed FEEDER from the training dataset, and then incorporate existing demonstration selector to get samples regarding specific test input. This pre-selected subset is characterized by its sufficiency and necessity conditioned on the frozen LLM. In the fine-tuning setting, shown in (b), FEEDER allows the LLM to be tuned on the fixed subset FEEDER, and this subset is intentionally pre-selected to be a faithful representation of the training dataset, with the dual objectives of maintaining data quality and minimizing computational expenses. The above two processes can be encapsulated into a bi-level optimization framework, allowing for iterative refinement of both the pre-selected FEEDER and the fine-tuned LLM.

evaluating all possible subsets is impractical, we develop a tree-based approximation algorithm that leverages the capability of the given LLM to assess whether each subset is sufficient and necessary for representing others.

Besides ICL, we further observe that our pre-selection process can enhance the fine-tuning process by allowing LLMs to achieve comparable or even superior performance while training on the pre-selected representative subset of the data, rather than the full data, where we introduce a bi-level optimization method that enhances training efficiency without sacrificing performance.

Our empirical evaluations encompass six LLM bases, ranging from 335M to 7B parameters, and include six demonstration selectors in the demonstration selection stage, applied to text classification, reasoning, and semantic parsing tasks. Our results demonstrate that our pre-selection process produces a subset that consistently reduces the data size for demonstration selection by over 20% across various datasets. Moreover, using this pre-selected subset in ICL with a simple similarity-based metric can outperform existing sophisticated metrics such as diversity-based approaches. Additionally, we show that this subset can enhance the fine-tuning process, enabling LLMs trained on the selected representative subset to achieve comparable or even superior performance.

To summarize, our pre-selector FEEDER offers three key advantages:

- The pre-selection stage effectively identifies a representative subset, eliminating unnecessary data and significantly reducing the high computational complexity of downstream demonstration selection.

- FEEDER evaluates the representativeness of demonstrations in terms of sufficiency and necessity, taking the LLM's capabilities and knowledge into consideration.

- The pre-selected subset is beneficial not only for ICL but also for accelerating the fine-tuning process while maintaining comparable or even superior performance.

## 2. Demonstration Pre-Selection

We begin by delineating two distinct contexts where FEEDER can operate: ICL and fine-tuning settings. Throughout this paper, we approach both scenarios from a data-centric perspective (Strickland, 2022), emphasizing the significance of *data quality* over *data quantity*.

In the ICL setting, we are given a training dataset $\mathcal{D}_{\text{TRAIN}} = \{(\boldsymbol{x}_n, \boldsymbol{y}_n)\}_{n=1}^N$ consisting of pairs of input data (e.g., questions) and output labels (e.g., answers). We are also given a test dataset $\mathcal{D}_{\text{TEST}} = \{(\boldsymbol{x}_m, \boldsymbol{y}_m)\}_{m=1}^M$, where we assume that $\mathcal{D}_{\text{TRAIN}}$ share the same support set (Yosida, 2012) with $\mathcal{D}_{\text{TEST}}$. Our goal is to develop a demonstration selector that extracts n-shot demonstrations from the training dataset, denoted as $\mathcal{D}_{\text{DEMO}} \subseteq \mathcal{D}_{\text{TRAIN}}$. We use $\Psi_{\text{LLM}} : \mathbb{X} \times \mathbb{D} \to \mathbb{Y}$ to represent a LLM using selected demonstrations as the context. Here, $\boldsymbol{x}_\cdot \in \mathbb{X}$ is an input text, $\boldsymbol{y}_\cdot \in \mathbb{Y}$ is the corresponding output, and $(\boldsymbol{x}_\cdot, \boldsymbol{y}_\cdot) \in \mathbb{D}$ is one demonstration. Formally, our objective is to minimize:

$$\mathcal{L}(\mathcal{D}_{\text{DEMO}}, \mathcal{D}_{\text{TEST}}) = \sum_{(\boldsymbol{x}_m, \boldsymbol{y}_m) \in \mathcal{D}_{\text{TEST}}} \ell\Big(\Psi_{\text{LLM}}^*(\boldsymbol{x}_m, \mathcal{D}_{\text{DEMO}}), \boldsymbol{y}_m\Big),$$
(1)

where $\ell(\cdot, \cdot)$ is the task-specific loss function, and $\Psi_{\text{LLM}}^*(\cdot)$ means that the LLM is frozen. However, since we do not have access to $\mathcal{D}_{\text{TEST}}$ during the training phase, it is impractical to optimize the demonstration selection directly by minimizing $\mathcal{L}(\mathcal{D}_{\text{DEMO}}, \mathcal{D}_{\text{TEST}})$.

Instead, we re-consider the demonstration selection task as a two-stage problem, where we first *pre-select* a subset of *high-quality* demonstrations from $\mathcal{D}_{\text{TRAIN}}$ as the selection pool, i.e., a FEEDER subset denoted as $\mathcal{D}_{\text{FEEDER}}$; and

then we apply existing demonstration selectors such as random or similarity-based selectors on $\mathcal{D}_{\texttt{FEEDER}}$, to choose the corresponding demonstrations as context for a specific test instance. Our key idea is that a high-quality training dataset $\mathcal{D}_{\texttt{FEEDER}}$ should be both representative of the entire training dataset $\mathcal{D}_{\texttt{TRAIN}}$ and as minimal in size as possible. Formally, we use the loss function $\mathcal{L}(\mathcal{D}_{\texttt{FEEDER}}, \mathcal{D}_{\texttt{TRAIN}})$ from Eq. (1) to evaluate our *pre-selector*, i.e., how well the representation of $\mathcal{D}_{\texttt{FEEDER}}$ aligns with $\mathcal{D}_{\texttt{TRAIN}}$. Then, our objective can be written as:

$$
\min_{\mathcal{D}_{\texttt{FEEDER}} \subseteq \mathcal{D}_{\texttt{TRAIN}}} |\mathcal{D}_{\texttt{FEEDER}}|,
$$
$$
\text{s.t. } \mathcal{L}(\mathcal{D}_{\texttt{FEEDER}}, \mathcal{D}_{\texttt{TRAIN}}) \leq \mathcal{L}(\mathcal{D}_{\texttt{TRAIN}}, \mathcal{D}_{\texttt{TRAIN}}). \tag{2}
$$

This formulation indices that $\mathcal{D}_{\texttt{FEEDER}}$ should be not only sufficient but also necessary to represent $\mathcal{D}_{\texttt{TRAIN}}$, thus removing redundant data points to save computation costs meanwhile maintaining LLM performance.

Our pre-selected subset of high-quality data $\mathcal{D}_{\texttt{FEEDER}}$ also can be applied to fine-tune LLMs. Concretely, instead of fine-tuning LLMs on the entire training dataset $\mathcal{D}_{\texttt{TRAIN}}$, $\mathcal{D}_{\texttt{FEEDER}}$ allows us to fine-tune LLMs with few but high-quality data, reducing computation costs. In this case, the LLM $\Psi_{\texttt{LLM}}$ is usually trainable, and our goal can be formulated as:

$$
\min_{\Psi_{\texttt{LLM}}} \mathbb{E}_{(\boldsymbol{x}_n, \boldsymbol{y}_n) \in \mathcal{D}_{\texttt{FEEDER}}^*} [\ell\big(\Psi_{\texttt{LLM}}(\boldsymbol{x}_n, \emptyset), \boldsymbol{y}_n\big)], \tag{3}
$$

where $\mathcal{D}_{\texttt{FEEDER}}^*$ means that the pre-selected $\mathcal{D}_{\texttt{FEEDER}}$ is fixed during fine-tuning.

---

**Algorithm 1** Bi-level Optimization
---
**Input:** Training dataset $\mathcal{D}_{\texttt{TRAIN}}$, LLM $\Psi_{\texttt{LLM}}$.
**Output:** Approximated subset $\widetilde{\mathcal{D}}_{\texttt{FEEDER}}$, tuned LLM $\Psi_{\texttt{LLM}}$.
Initialize $\widetilde{\mathcal{D}}_{\texttt{FEEDER}} = \mathcal{D}_{\texttt{TRAIN}}$.
**for** *each iteration* **do**

> Update $\widetilde{\mathcal{D}}_{\texttt{FEEDER}}$ by using our approximation algorithm with frozen LLM $\Psi_{\texttt{LLM}}$.
> Tune LLM $\Psi_{\texttt{LLM}}$ by using Eq. (3) as our loss function on fixed $\widetilde{\mathcal{D}}_{\texttt{FEEDER}}$.

---

Given the above analysis, we can further bridge the (pre)-selection of $\mathcal{D}_{\texttt{FEEDER}}$ and the LLM fine-tuning on $\mathcal{D}_{\texttt{FEEDER}}$ into a bi-level optimization framework. On the outer level, following Eq. (2), we optimize the selection of $\mathcal{D}_{\texttt{FEEDER}}$ in the context of a frozen LLM $\Psi_{\texttt{LLM}}^*$; while on the inner level, following Eq. (3), we optimize the LLM $\Psi_{\texttt{LLM}}$ using the fixed dataset $\mathcal{D}_{\texttt{FEEDER}}^*$. The bi-level optimization procedure described above is amenable to repetition, enabling iterative refinement of both the selected $\mathcal{D}_{\texttt{FEEDER}}$ and the tuned LLM. The overall process is summarized in Algorithm 1, and the construction of our FEEDER subset is detailed in the subsequent sections.

## 3. Connections to Existing Work

With the growing capabilities of LLMs, data (referred to as "demonstrations") selection has gained prominence, which involves selecting suitable examples as the context for in-context learning (Dong et al., 2022; Yang et al., 2023; Zhou et al., 2022) or filtering a subset from training examples for fine-tuning (Sachdeva et al., 2024; Zhou et al., 2024). Previous solutions have revolved around constructing either parameter-free selection mechanisms (Wang et al., 2022; Zemlyanskiy et al., 2022; Gao et al., 2023) or neural-based selection methods (Pasupat et al., 2021; Liu et al., 2021; Gupta et al., 2021; Rubin et al., 2021; Li et al., 2023). Recent investigations (Xia et al., 2024; Marion et al., 2023) focus on mining training examples for fine-tuning specific tasks, with Wang et al. (2024) and Wan et al. (2025) extending this approach to in-context learning. In contrast to previous methods that use LLMs as demonstration selectors, our work leverages the powerful few-shot inference capabilities of LLMs by employing them as *pre-selectors*. For this purpose, we introduce a pre-selection stage to examine "sufficiency" and "necessity" to identify a representative subset of training examples. The resulting FEEDER subset can serve a dual purpose: they can be used as candidate input contexts or to fine-tune the LLM. In both scenarios, FEEDER can significantly reduce the computation costs by substituting the entire training data with FEEDER subset.

## 4. Demonstration Pre-Selector with Sufficiency and Necessity Metrics

**Notations.** Let $X, C$ denote variables for the input and the context (i.e., pre-selected demonstrations). We introduce $Y$, a boolean variable, to represent whether the corresponding output is correct. For simplicity, we use $Y_{\boldsymbol{x}_n} = 1$ to denote $Y = 1|X = \boldsymbol{x}_n$, meaning that the LLM generates the correct output for the input $\boldsymbol{x}_n$. Similarly, $Y_{\boldsymbol{x}_n} = 0$, equivalent to $Y = 0|X = \boldsymbol{x}_n$, indicates that LLM produces an incorrect output for $\boldsymbol{x}_n$. For convenience, we introduce $S$, a variable to record the original status of the LLM before new plug-in and unplug operations (denoted as $\texttt{plug}(\cdot)$ and $\texttt{unplug}(\cdot)$ respectively). The connections between the above operations and the $\texttt{do}(\cdot)$ operation in causality are discussed in Appendix A1.

We begin by considering the relationship between two demonstrations, denoted as $(\boldsymbol{x}_n, \boldsymbol{y}_n)$ and $(\boldsymbol{x}_m, \boldsymbol{y}_m)$, and the sufficiency and necessity metrics based on particular LLMs as follows.

The sufficiency metric is introduced to assess whether plugging in one data point is adequate for the LLM to produce the correct answer to another data point. Formally, we define sufficiency between pair of demonstrations as:

**Definition 1** (Sufficiency Metric). *Given tuple $(X, Y, C, S)$,*

*a training sample $(\boldsymbol{x}_n, \boldsymbol{y}_n)$ is considered sufficient for another one $(\boldsymbol{x}_m, \boldsymbol{y}_m)$, if the following equation holds:*

$$Y_{\boldsymbol{x}_m} = 1|\texttt{plug}((\boldsymbol{x}_n, \boldsymbol{y}_n)); C, S = (Y_{\boldsymbol{x}_m} = 0), \quad (4)$$

*where $(\boldsymbol{x}_n, \boldsymbol{y}_n)$ is not included in $C$. It means that when plugging in $(\boldsymbol{x}_n, \boldsymbol{y}_n)$, it would correct the LLM's answer to $\boldsymbol{x}_m$.*

The necessity metric is introduced to assess whether it is necessary to retain a particular plugged-in data point to maintain the correct output of another data point. Its formal definition over pairs of demonstrations can be written as:

**Definition 2** (Necessary Metric). *Given tuple $(X, Y, C, S)$, a training sample $(\boldsymbol{x}_n, \boldsymbol{y}_n)$ is considered necessary for $(\boldsymbol{x}_m, \boldsymbol{y}_m)$, if the following equation holds:*

$$Y_{\boldsymbol{x}_m} = 0|\texttt{unplug}((\boldsymbol{x}_n, \boldsymbol{y}_n)); C, S = (Y_{\boldsymbol{x}_m} = 1), \quad (5)$$

*where $(\boldsymbol{x}_n, \boldsymbol{y}_n)$ is included in $C$. It means that prior to unplugging $(\boldsymbol{x}_n, \boldsymbol{y}_n)$, the LLM's output is correct. However, when we do unplug $(\boldsymbol{x}_n, \boldsymbol{y}_n)$ from the context, it causes the LLM to offer an incorrect output.*

The above definitions of sufficiency and necessity metrics, operating on the instance level, are further clarified with examples in Appendix A2.1.

Extending these definitions to the set level, a sufficient set signifies that plugging in a specific set is adequate to ensure the correct outputs for all examples in another set, while a necessary set implies that removing any example from this set would result in incorrect answers for at least one example within another set. Formal definitions for the above set-level metrics, along with examples, are available in Appendix A2.2.

Taking into account both the sufficiency and necessity metrics, we define a subset of the training dataset $\mathcal{D}_{\texttt{TRAIN}}$ as FEEDER subset $\mathcal{D}_{\texttt{FEEDER}}$, if it can be both sufficient and necessary to represent $\mathcal{D}_{\texttt{TRAIN}}$. We provide its formal definition along with some examples in Appendix A2.3. Strictly following the above definition to discover a FEEDER subset is impractical because the constraints are too stringent and the computational costs are prohibitively high with $O(2^N)$ computational complexity to enumerate all the possible subsets. Therefore, we propose an approximation algorithm for discovering a FEEDER subset with a tree-based algorithm.

Our tree-based approach leverages the capability of the LLM to assess the sufficiency and necessity for each sample in $\mathcal{D}_{\texttt{TRAIN}}$, which can filter out unnecessary portions of $\mathcal{D}_{\texttt{TRAIN}}$, while retaining a sufficient subset that effectively represents the entire train data $\mathcal{D}_{\texttt{TRAIN}}$. Concretely, our tree expands from the bottom to the top, where each node represents a set of instances. Formally, we use the variable $K$ to represent the depth of the tree, corresponding to the number of rounds.

**An Example of Approximation Algorithm for FEEDER**

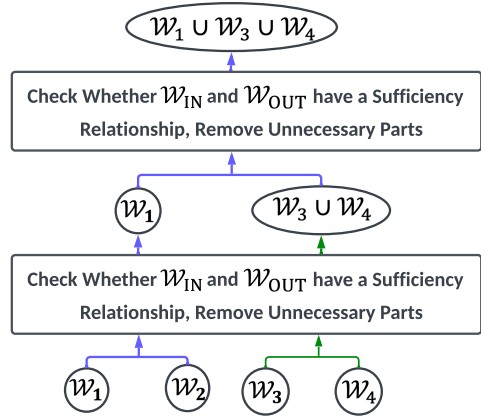

*Figure 2.* An illustrated example of our approximation algorithm for FEEDER. At each round (corresponding to each layer of the tree), we check whether there is a sufficiency relationship between each pair of nodes. After each check, we remove those unnecessary parts from $\mathscr{W}_\cdot$.

Specifically, we use $k = 1, 2, \ldots, K$ to refer to each $k$-th round; and during each $k$-th round, we generate the $(k+1)$-th layer of the tree. We denote $\mathscr{W}_k$ as the set of nodes after the $k$-th round. We initialize $\mathscr{W}_0$ by assigning all the samples in $\mathcal{D}_{\texttt{TRAIN}}$ as the bottom nodes:

$$\mathscr{W}_0 := \{\mathcal{W}_n := \{(\boldsymbol{x}_n, \boldsymbol{y}_n)\}|(\boldsymbol{x}_n, \boldsymbol{y}_n) \in \mathcal{D}_{\texttt{TRAIN}}\}. \quad (6)$$

During each $k$-th round, we generate $\mathscr{W}_k$ from $\mathscr{W}_{k-1}$. This is achieved by examining the sufficiency relationship between every pair of nodes in $\mathscr{W}_{k-1}$, denoted as $\mathcal{W}_i, \mathcal{W}_j \in \mathscr{W}_{k-1}$. In this evaluation, we assess whether the following equation holds true by assigning $\mathcal{W}_i$ and $\mathcal{W}_j$ as $\mathcal{W}_{\texttt{IN}}$ and $\mathcal{W}_{\texttt{OUT}}$, or vice versa:

$$Y_{(\{\boldsymbol{x}_n|\boldsymbol{x}_n \in \mathcal{W}_{\texttt{OUT}}\})} = \mathbf{1}_{|\mathcal{W}_{\texttt{OUT}}|}|\texttt{plug}(\mathcal{W}_{\texttt{IN}}); C, S, \quad (7)$$

where $C = \emptyset$ and $S$ is loosened to allow for any value. If the above equation holds, it signifies that plugging in $\mathcal{W}_{\texttt{IN}}$ is *sufficient* for the LLM to generate the correct output to any input in $\mathcal{W}_{\texttt{OUT}}$. In other words, once we have $\mathcal{W}_{\texttt{IN}}$ included in the plugged-in context, it is *unnecessary* to further include $\mathcal{W}_{\texttt{OUT}}$. Formally, we can derive the following equation that is equivalent to Eq. (7) as:

$$Y_{(\{\boldsymbol{x}_n|\boldsymbol{x}_n \in \mathcal{W}_{\texttt{OUT}}\})} = \mathbf{1}_{|\mathcal{W}_{\texttt{OUT}}|}|\texttt{unplug}(\mathcal{W}_{\texttt{OUT}}); C, S, \quad (8)$$

where $C = (\mathcal{W}_{\texttt{IN}} \cup \mathcal{W}_{\texttt{OUT}})$ and $S$ is loosened to be any value.

Concretely, there are three possible scenarios by examining each pair of nodes in $\mathscr{W}_{k-1}$ using Eq. (7): (i) If both $\mathcal{W}_i$ and $\mathcal{W}_j$ are sufficient for representing each other, then we select the one with fewer elements to append to $\mathscr{W}_k$. (ii) If only one of $\mathcal{W}_i$ and $\mathcal{W}_j$ is sufficient to represent the other, then we append the sufficient one to $\mathscr{W}_k$. (iii) If neither $\mathcal{W}_i$ nor $\mathcal{W}_j$ is sufficient to represent the other, we append $\mathcal{W}_i \cup \mathcal{W}_j$

to $\mathcal{W}_k$. After performing the above calculations for each pair of nodes, we remove them from $\mathcal{W}_{k-1}$. When there is only one element left in $\mathcal{W}_{k-1}$, it is directly appended to $\mathcal{W}_k$. This process continues until $\mathcal{W}$. contains only one element.

Then, we can effectively remove unnecessary samples from $\mathcal{D}_{\text{TRAIN}}$ by extending the tree structure above from the bottom to the top. Moreover, the complexity of the above algorithm with $K$ rounds (corresponding to a tree depth of $K + 1$) is $O(K \log_2^{|\mathcal{D}_{\text{TRAIN}}|})$. In practice, we investigate the impact of varying $K$ and find that setting $K = 1$ already produces excellent performance. This indicates that a one-shot inference by LLM is sufficient to assess the sufficiency between each pair of samples. Once the results of pairs of samples are computed, we take their union to form the resulting subset $\mathcal{D}_{\text{FEEDER}}$. Figure 2 illustrates the process for $K = 2$. When $K = 1$, the top-level check between $\mathcal{W}_1$ and $\mathcal{W}_3 \cup \mathcal{W}_4$ is no longer required. Instead, the resulting subset is $\mathcal{W}_1 \cup \mathcal{W}_3 \cup \mathcal{W}_4$ after the bottom-level check.

Moreover, our tree-based algorithm can be iterated across multiple runs to further refine the necessary components. Specifically, the FEEDER subset obtained from one run serves as the input for the next. We denote the number of runs as $R$, leading to an overall complexity of $O(RK \log_2^{|\mathcal{D}_{\text{TRAIN}}|})$. Empirical investigations show that varying $R$ has minimal impact, with a single run ($R = 1$) already achieving strong performance. Thus, our implementation complexity simplifies to $O(\log_2^{|\mathcal{D}_{\text{TRAIN}}|})$.

Our tree-based approximation algorithm is summarized in Algorithm 2 in Appendix A3. If we hypothesize that sufficiency is transitive among sets, our tree-based approximation algorithm can ensure that the remaining subset remains sufficient to represent the entire $\mathcal{D}_{\text{TRAIN}}$, as verified in the following proposition.

**Proposition 1.** *If we successively apply our tree-based approximation algorithm on $\mathcal{D}_{\text{TRAIN}}$ for a certain number of runs to obtain a subset (denoted as $\widetilde{\mathcal{D}}_{\text{FEEDER}}$), then $\mathcal{D}_{\text{FEEDER}}$ is sufficient to represent $\mathcal{D}_{\text{TRAIN}}$.*

We offer the proof of the above proposition in Appendix A3, which demonstrates that our tree-based algorithm can effectively remove unnecessary samples from $\mathcal{D}_{\text{TRAIN}}$ while ensuring that the resulting subset remains sufficient to represent the entire training dataset.

Additionally, we present another algorithm for finding an exact sufficient and necessary subset from $\mathcal{D}_{\text{TRAIN}}$, along with its proof and deployment discussion, in Appendices A4.1, Appendix A4.2, and A7.

# 5. Experiments

Our evaluations are mainly conducted on 6 text classification datasets: SST-2 (Socher et al., 2013), SST-5 (Socher et al., 2013), COLA (Warstadt et al., 2018), TREC (Voorhees & Tice, 2000), SUBJ (Pang & Lee, 2004), and FPB (Malo et al., 2014). These datasets cover a range of tasks from sentiment classification and linguistic analysis to textual entailment. We also further assess FEEDER on the reasoning dataset GSM8K (Cobbe et al., 2021), the semantic-parsing dataset SMCALFlow (Andreas et al., 2020), and the scientific question-answering dataset GPQA (Rein et al., 2024). For each dataset, we directly follow the official splits to obtain $\mathcal{D}_{\text{TRAIN}}$ and $\mathcal{D}_{\text{TEST}}$.

To evaluate the performance of our approach, we employed two GPT-2 variants (Radford et al., 2019): one with 335M parameters, and the other with 774M parameters; one GPT-neo with 1.3B parameters; one GPT-3 variant (Brown et al., 2020) with 6B parameters; one Gemma-2 variant (Team et al., 2024) with 2B parameters, one Llama-2 variant (Touvron et al., 2023) with 7B parameters, Llama-3 variant (Meta, 2024) with 8B parameters, and Qwen-2.5 variant (Yang et al., 2024) with 32B parameters, as the LLM base.

## 5.1. Performance on In-context Learning

Since our $\mathcal{D}_{\text{FEEDER}}$ works as a pre-selector, when applied in the in-context learning setting, we propose incorporating demonstration selectors into FEEDER. In other words, our evaluations follow an ablative approach, with the baseline involving the direct application of these demonstration selectors on $\mathcal{D}_{\text{TRAIN}}$. This baseline can be regarded as treating these methods both as pre-selectors and demonstration selectors. As discussed in Section 4, our $\mathcal{D}_{\text{FEEDER}}$ is applied using only a one-shot inference check (i.e., $K = 1$) and a single-round run (i.e., $R = 1$), unless otherwise stated.

Concretely, we conducted an evaluation of FEEDER in conjunction with following six demonstration selectors: (i) Random is the random selector, which selects input demonstration randomly from the selection pool; (ii) Similarity (Sorensen et al., 2022; Gonen et al., 2022) is the similarity-based selector, which selects relevant demonstrations in terms of the cosine similarity metric over the embedding vectors generated by a sentence transformer (Reimers & Gurevych, 2019); (iii) Diversity (Ye et al., 2022) is the diversity-based selector , which selects similar and diverse demonstrations in terms of maximal marginal relevance (Carbonell & Goldstein, 1998); (iv) Uncertainty (Köksal et al., 2022) is the uncertainty-based selector that conducts selections according to their uncertainty metric; (v) Clustering (Zhou et al., 2023) is the clustering-based selector that searches demonstrations by clustering. (vi) Latent (Wang et al., 2024) uses LLMs as latent variable models to learn latent variables for down-streaming in-context learning. Please refer to Appendix A5.1 for detailed descriptions of the above demonstration selectors.

Experimental results regarding in-context learning perfor-

*Table 1.* Performance comparisons on text classification datasets are conducted in the in-context learning setting. We report both the mean and variance of accuracy using 8 different seeds and 5 different permutations of n-shots. Refer to Appendix A5.2 for more extended results on LLMs GPT-2 variants and GPT-3 variants, datasets FPB, SST-5, TREC, and demonstration selectors Uncertainty, Clustering and Latent.

| $\Psi_{\text{LLM}}(\cdot)$ | $\mathcal{D}$ | $n$ | SUBJ | | | SST-2 | | | COLA | | |
|---|---|---|---|---|---|---|---|---|---|---|---|
| | | | Random | Similarity | Diversity | Random | Similarity | Diversity | Random | Similarity | Diversity |
| Gemma-2 (2B) | $\mathcal{D}_{\text{TRAIN}}$ | 1 | 45.0 (5.9) | 48.1 (0.6) | 48.1 (0.6) | 51.2 (6.8) | 52.2 (0.8) | 52.2 (0.8) | 37.5 (7.0) | 40.5 (1.3) | 40.5 (1.3) |
| | | 2 | 62.3 (6.9) | 82.5 (1.8) | 74.2 (1.3) | 71.5 (5.6) | 78.5 (1.5) | 75.9 (0.9) | 40.6 (5.9) | 62.5 (1.0) | 61.6 (0.5) |
| | | 5 | 68.0 (7.1) | 91.5 (1.2) | 84.2 (1.6) | 70.2 (5.6) | 80.5 (1.6) | 80.6 (0.7) | 46.5 (5.9) | 67.2 (1.8) | 65.6 (0.6) |
| | | 10 | 50.3 (8.2) | 86.2 (1.9) | 85.6 (0.8) | 68.2 (4.8) | 85.5 (1.5) | 76.3 (1.3) | 50.2 (7.4) | 69.8 (1.5) | 71.5 (1.2) |
| | $\mathcal{D}_{\text{FEEDER}}$ | 1 | **48.2** (4.2) | **49.5** (1.0) | **49.5** (1.0) | **52.6** (4.6) | **53.1** (0.8) | **53.1** (0.8) | **38.9** (5.2) | 39.6 (0.8) | 39.6 (0.8) |
| | | 2 | **65.2** (2.9) | **85.2** (1.0) | **80.3** (0.8) | **74.2** (4.9) | **82.1** (1.2) | **83.0** (0.7) | **52.5** (2.5) | **68.9** (2.1) | **67.8** (1.5) |
| | | 5 | **72.2** (6.2) | **94.5** (5.3) | **85.5** (0.7) | 72.0 (4.2) | **83.6** (2.1) | **84.5** (1.7) | **55.2** (4.8) | **77.6** (2.5) | **73.9** (2.3) |
| | | 10 | **60.5** (4.0) | **86.5** (2.5) | **88.4** (2.4) | **70.5** (5.6) | **92.6** (2.6) | **78.5** (5.3) | **58.6** (4.6) | **75.6** (2.9) | **76.6** (2.5) |
| GPT-3 (6B) | $\mathcal{D}_{\text{TRAIN}}$ | 1 | 44.9 (6.6) | 49.5 (0.1) | 49.5 (0.1) | 48.2 (2.9) | 47.0 (0.1) | 47.0 (0.1) | 38.9 (6.7) | 41.2 (0.2) | 41.2 (0.2) |
| | | 2 | 55.4 (3.5) | 85.5 (0.1) | 86.5 (0.2) | 68.1 (4.2) | 78.7 (0.2) | 77.5 (0.1) | 42.8 (4.0) | 45.5 (0.3) | 45.6 (0.2) |
| | | 5 | 51.2 (4.4) | 90.8 (0.2) | 82.7 (0.1) | 75.2 (3.3) | 80.7 (0.1) | 77.8 (0.2) | 48.5 (3.3) | 51.8 (0.3) | 52.1 (0.2) |
| | | 10 | 57.7 (4.8) | 87.3 (0.1) | 85.3 (0.1) | 72.1 (3.8) | 77.6 (0.1) | 76.5 (0.2) | 59.1 (4.2) | 60.3 (0.1) | 61.0 (0.2) |
| | $\mathcal{D}_{\text{FEEDER}}$ | 1 | 43.9 (4.2) | **51.2** (1.0) | **51.2** (1.0) | 49.6 (2.4) | **51.3** (1.6) | **51.3** (1.6) | **41.2** (2.1) | **43.8** (1.8) | **43.8** (1.8) |
| | | 2 | 65.7 (3.0) | **91.5** (1.1) | **88.8** (1.6) | 73.5 (2.5) | **85.7** (4.2) | 76.1 (2.1) | **61.8** (2.1) | **63.1** (1.5) | **60.1** (1.4) |
| | | 5 | 53.7 (3.8) | **92.9** (0.8) | **91.5** (1.4) | 77.6 (4.0) | 81.0 (1.3) | 79.4 (1.0) | 50.6 (2.7) | **63.3** (1.4) | **65.8** (1.4) |
| | | 10 | 58.0 (3.4) | **88.8** (0.9) | **87.8** (1.2) | **83.8** (2.8) | **86.4** (2.0) | **87.2** (1.3) | 59.7 (3.0) | **67.5** (1.9) | **68.4** (2.2) |
| Llama-2 (7B) | $\mathcal{D}_{\text{TRAIN}}$ | 1 | 42.9 (6.6) | 48.5 (0.1) | 48.5 (0.1) | 46.2 (2.7) | 49.1 (0.1) | 49.1 (0.1) | 40.1 (6.1) | 42.0 (0.2) | 42.0 (0.2) |
| | | 2 | 51.9 (4.4) | 90.7 (0.1) | 85.2 (0.2) | 67.8 (3.2) | 73.5 (0.2) | 74.5 (0.2) | 43.5 (4.5) | 47.4 (0.2) | 49.6 (0.1) |
| | | 5 | 51.6 (3.2) | 86.8 (0.2) | 82.9 (0.1) | 74.8 (3.8) | 81.2 (0.2) | 78.7 (0.2) | 50.2 (3.7) | 52.6 (0.2) | 48.2 (0.3) |
| | | 10 | 56.1 (4.6) | 81.3 (0.1) | 85.7 (0.1) | 73.2 (3.1) | 76.3 (0.1) | 77.1 (0.1) | 59.6 (4.3) | 55.3 (0.2) | 60.0 (0.4) |
| | $\mathcal{D}_{\text{FEEDER}}$ | 1 | **43.8** (4.3) | **49.7** (1.0) | **49.7** (1.0) | **47.2** (2.4) | **50.8** (1.7) | **50.8** (1.7) | **41.2** (2.1) | **43.8** (1.8) | **43.8** (1.8) |
| | | 2 | 54.8 (3.0) | **92.5** (1.1) | 84.8 (0.7) | 72.2 (3.1) | **82.5** (4.0) | **80.1** (2.6) | 50.8 (2.3) | **58.6** (1.7) | **53.5** (1.3) |
| | | 5 | 53.7 (3.8) | **87.9** (1.8) | **91.5** (1.4) | 78.3 (4.6) | **83.2** (1.1) | **80.1** (1.4) | 53.8 (2.8) | **65.3** (1.6) | **61.8** (1.4) |
| | | 10 | 58.0 (3.4) | **85.8** (0.9) | **87.8** (1.2) | **85.0** (2.2) | **87.1** (2.2) | **86.9** (1.0) | **60.5** (3.1) | **68.0** (1.7) | **68.4** (2.0) |

*Table 2.* Performance comparisons on reasoning GSM8K dataset and semantic-parsing SMCALFlow dataset are conducted in the in-context learning setting. We report both the mean and variance of accuracy using 8 different seeds and 5 different permutations of n-shots. Refer to Appendix A5.3 for more extended results on demonstration selectors Clustering and Latent.

| $\Psi_{\text{LLM}}(\cdot)$ | $\mathcal{D}$ | $n$ | GSM8K | | | | SMCALFlow | | | |
|---|---|---|---|---|---|---|---|---|---|---|
| | | | Random | Similarity | Diversity | Uncertainty | Random | Similarity | Diversity | Uncertainty |
| Gemma-2 (2B) | $\mathcal{D}_{\text{TRAIN}}$ | 1 | 6.54 (1.56) | 15.16 (0.17) | 15.16 (0.17) | 10.51 (0.78) | 8.54 (1.64) | 19.12 (0.15) | 19.12 (0.15) | 11.21 (0.89) |
| | | 2 | 8.56 (0.85) | 18.89 (0.85) | 19.52 (0.45) | 17.58 (0.27) | 9.56 (0.84) | 20.05 (0.36) | 22.50 (0.41) | 13.58 (0.77) |
| | | 5 | 15.30 (2.89) | 20.31 (0.58) | 21.56 (0.78) | 19.30 (0.90) | 18.56 (4.58) | 28.65 (0.95) | 27.89 (1.85) | 25.22 (3.56) |
| | | 10 | 17.45 (4.21) | 21.52 (0.49) | 20.85 (0.55) | 20.66 (1.84) | 19.85 (5.21) | 30.58 (1.04) | 28.56 (0.58) | 31.00 (0.88) |
| | $\mathcal{D}_{\text{FEEDER}}$ | 1 | **10.25** (0.51) | **16.25** (0.21) | **16.25** (0.21) | **11.12** (1.78) | **9.64** (0.55) | **20.54** (0.66) | **20.54** (0.66) | **15.25** (0.87) |
| | | 2 | **13.76** (0.48) | **19.68** (0.13) | **20.51** (1.55) | 16.85 (3.65) | **10.25** (0.52) | **23.73** (0.18) | **24.25** (2.65) | **17.58** (6.58) |
| | | 5 | **18.52** (5.21) | **22.58** (0.85) | **22.05** (0.77) | **20.20** (2.05) | **20.44** (5.12) | **30.54** (4.58) | **32.54** (5.21) | **28.95** (3.66) |
| | | 10 | **19.20** (5.22) | **22.20** (1.45) | **23.52** (2.20) | **22.10** (6.21) | **21.52** (2.01) | **31.48** (1.52) | **31.02** (2.54) | 30.01 (1.20) |
| Llama-2 (7B) | $\mathcal{D}_{\text{TRAIN}}$ | 1 | 2.45 (0.83) | 3.52 (0.88) | 3.52 (0.88) | 3.05 (0.25) | 2.25 (0.64) | 10.25 (0.85) | 10.25 (0.85) | 9.01 (0.33) |
| | | 2 | 2.65 (0.77) | 4.97 (0.18) | 5.62 (0.85) | 4.12 (0.47) | 4.97 (0.84) | 10.05 (2.36) | 10.52 (1.45) | 11.20 (1.54) |
| | | 5 | 3.54 (0.88) | 8.25 (0.89) | 7.25 (0.96) | 7.88 (0.64) | 7.52 (0.85) | 16.20 (1.85) | 15.28 (1.75) | 15.33 (1.30) |
| | | 10 | 4.25 (0.36) | 8.85 (0.85) | 9.21 (1.98) | 8.10 (1.11) | 8.70 (1.05) | 18.95 (1.25) | 19.55 (2.01) | 17.52 (2.66) |
| | $\mathcal{D}_{\text{FEEDER}}$ | 1 | **3.54** (0.51) | **4.44** (0.89) | **4.44** (0.89) | **3.36** (0.66) | **3.64** (0.55) | **10.89** (0.63) | **10.89** (0.63) | **10.02** (0.69) |
| | | 2 | **3.76** (0.48) | **5.68** (0.13) | **6.66** (0.58) | **4.85** (0.88) | **4.25** (0.52) | **12.03** (0.16) | **11.13** (1.10) | **12.50** (2.01) |
| | | 5 | **4.20** (1.23) | **9.22** (1.01) | **8.81** (0.98) | **8.20** (1.14) | **8.25** (1.25) | **17.20** (3.66) | **16.66** (5.20) | **16.06** (2.22) |
| | | 10 | **5.02** (1.51) | **10.22** (1.32) | **9.25** (0.79) | **9.45** (0.66) | **9.20** (0.77) | **20.11** (2.02) | **21.25** (3.36) | **20.22** (4.02) |
| Llama-3 (8B) | $\mathcal{D}_{\text{TRAIN}}$ | 1 | 78.24 (6.56) | 79.56 (3.42) | 79.56 (3.42) | 78.42 (3.76) | 12.37 (6.65) | 15.64 (2.34) | 15.64 (2.34) | 14.35 (4.56) |
| | | 2 | 79.55 (7.29) | 83.40 (4.53) | 83.67 (4.05) | 81.23 (3.53) | 13.21 (4.34) | 16.74 (3.45) | 17.43 (3.65) | 16.60 (4.62) |
| | | 5 | 81.45 (5.43) | 83.47 (5.63) | 84.52 (4.76) | 82.34 (5.34) | 14.53 (5.23) | 16.54 (2.35) | 17.87 (1.35) | 16.52 (3.21) |
| | | 10 | 82.31 (6.34) | 84.42 (3.24) | 84.53 (4.45) | 84.12 (4.44) | 14.63 (4.53) | 16.50 (2.21) | 18.64 (2.34) | 17.87 (2.23) |
| | $\mathcal{D}_{\text{FEEDER}}$ | 1 | **80.23** (4.43) | **81.21** (3.45) | **81.21** (3.45) | **79.64** (2.34) | **13.56** (3.22) | **16.55** (2.31) | **16.55** (2.31) | **15.40** (2.44) |
| | | 2 | **82.13** (4.76) | **84.43** (3.23) | **83.88** (3.33) | **82.22** (3.43) | **14.03** (3.35) | **17.45** (3.64) | **17.77** (3.20) | **17.00** (4.57) |
| | | 5 | **82.55** (5.96) | **85.03** (3.66) | **84.77** (3.77) | **83.56** (3.76) | **14.58** (3.45) | **18.22** (2.78) | **18.12** (2.01) | **17.53** (2.55) |
| | | 10 | **84.56** (2.33) | **85.79** (3.56) | **85.43** (4.55) | **84.98** (4.76) | **14.99** (4.65) | 16.66 (2.33) | **18.78** (3.42) | **18.01** (2.44) |

*Table 3.* Performance comparisons on text classification datasets are conducted in our bi-level optimization (i.e., fine-tuning) setting, where we tune the LLMs and evaluate their few-shot inference performance. We report both the mean and variance of accuracy using 8 different seeds and 5 different permutations of n-shots. Refer to Appendix A8.2 for more extended results on datasets FPB, SST-5, TREC.

| $\Psi_{\text{LLM}}(\cdot)$ | $\mathcal{D}$ | $n$ | SUBJ | | | SST-2 | | | COLA | | |
|---|---|---|---|---|---|---|---|---|---|---|---|
| | | | Random | Similarity | Diversity | Random | Similarity | Diversity | Random | Similarity | Diversity |
| GPT-2 (0.8B) | $\mathcal{D}_{\text{TRAIN}}$ | 1 | 67.8 (7.2) | 83.7 (0.1) | 83.7 (0.1) | 61.3 (8.1) | 71.6 (0.2) | 71.6 (0.2) | 59.3 (5.2) | 69.4 (0.2) | 69.4 (0.2) |
| | | 2 | 69.1 (4.3) | 88.7 (0.2) | 86.9 (0.2) | 73.5 (3.2) | 75.8 (0.5) | 74.2 (0.3) | 64.1 (5.7) | 74.1 (0.2) | 74.0 (0.3) |
| | | 5 | 70.8 (5.1) | 73.3 (0.1) | 72.7 (0.2) | 74.6 (4.1) | 82.8 (0.3) | 75.3 (0.2) | 60.9 (4.6) | 76.7 (0.3) | 76.4 (0.3) |
| | | 10 | 89.2 (4.1) | 94.0 (0.2) | 91.6 (0.2) | 70.8 (2.9) | 84.5 (0.2) | 77.4 (0.2) | 70.7 (3.8) | 75.7 (0.3) | 77.6 (0.5) |
| | $\mathcal{D}_{\text{FEEDER}}$ | 1 | 93.0 (4.3) | 93.5 (1.8) | 93.5 (1.8) | 89.5 (4.3) | 88.4 (1.6) | 88.4 (1.6) | 81.5 (3.3) | 82.6 (1.4) | 82.6 (1.4) |
| | | 2 | 96.1 (3.8) | 94.1 (1.3) | 92.6 (1.2) | 92.6 (2.8) | 94.4 (0.6) | 93.8 (0.7) | 90.2 (3.8) | 91.2 (1.7) | 90.8 (0.9) |
| | | 5 | 85.7 (3.5) | 94.7 (1.5) | 94.1 (1.1) | 87.5 (4.1) | 92.5 (1.7) | 93.7 (1.7) | 87.7 (3.2) | 89.6 (2.7) | 90.0 (3.9) |
| | | 10 | 90.5 (3.3) | 95.5 (1.3) | 95.6 (1.4) | 91.9 (2.9) | 93.1 (2.1) | 89.0 (1.4) | 91.3 (3.5) | 92.4 (1.8) | 93.5 (1.9) |
| GPT-neo (1.3B) | $\mathcal{D}_{\text{TRAIN}}$ | 1 | 72.7 (5.2) | 91.0 (0.1) | 91.0 (0.1) | 65.4 (4.4) | 72.5 (0.2) | 72.5 (0.2) | 61.8 (5.2) | 68.5 (0.2) | 68.5 (0.2) |
| | | 2 | 74.1 (4.3) | 93.7 (0.2) | 92.1 (0.3) | 74.5 (3.2) | 75.8 (0.4) | 76.4 (0.5) | 70.8 (5.7) | 63.9 (0.2) | 64.3 (0.4) |
| | | 5 | 71.8 (5.5) | 74.8 (0.3) | 75.8 (0.4) | 73.6 (4.1) | 77.8 (0.3) | 76.3 (0.2) | 68.7 (4.7) | 75.4 (0.8) | 74.9 (0.4) |
| | | 10 | 90.2 (4.0) | 93.6 (0.4) | 92.5 (0.4) | 72.8 (2.9) | 81.5 (0.2) | 78.8 (0.2) | 72.7 (3.4) | 76.7 (0.4) | 77.5 (0.7) |
| | $\mathcal{D}_{\text{FEEDER}}$ | 1 | 93.5 (4.3) | 94.1 (1.4) | 94.1 (1.4) | 91.2 (3.8) | 92.7 (1.5) | 92.7 (1.5) | 86.8 (3.3) | 89.6 (0.9) | 89.6 (0.9) |
| | | 2 | 95.5 (3.9) | 95.1 (1.3) | 96.6 (1.8) | 88.6 (2.4) | 93.4 (0.6) | 94.2 (0.5) | 84.2 (3.7) | 87.3 (0.7) | 89.5 (0.9) |
| | | 5 | 91.5 (3.8) | 95.7 (1.0) | 95.3 (1.4) | 89.4 (2.7) | 92.5 (1.8) | 93.7 (1.9) | 89.7 (3.2) | 92.4 (2.3) | 90.8 (1.8) |
| | | 10 | 92.8 (3.1) | 96.0 (1.4) | 94.8 (1.2) | 90.9 (2.0) | 93.6 (1.6) | 92.2 (1.8) | 89.3 (3.9) | 93.5 (1.7) | 94.4 (1.6) |

mance are reported in Tables 1, 2, and 4. We also present the reduction of our FEEDER in Figure 3. Our findings are summarized as follows.

- Our FEEDER working as a pre-selector can improve the performance of diverse demonstration selectors. Particularly, we can observe that in many cases, our FEEDER can incorporate with the similarity-based demonstration selector (i.e., $\mathcal{D}_{\text{FEEDER}}$+Similarity) can achieve comparable or even better performance than diversity or clustering-based demonstration selectors (i.e., $\mathcal{D}_{\text{TRAIN}}$+Diversity).
- By combining the results from Table 1 and Figure 3, it is evident that FEEDER enables the retention of almost half of the training samples while consistently achieving superior or comparable performance, which provides evidence supporting the efficacy of FEEDER as a proficient data pre-selection method for in-context learning.
- We also evaluate the few-shot performance on more complex tasks using LLMs Gemma-2, with the corresponding results reported in Table 2. The table demonstrates that, even though LLMs may not perform well on these tasks, our FEEDER can consistently enhance their performance.
- FEEDER performs well with a large number of shots. In Table 1, we can observe many cases where the LLM performance drops when the number of shots increases from 5 to 10 (e.g., Llama-2 on the COLA dataset). This may be caused by the introduction of noisy and redundant shots. Our FEEDER addresses this issue by evaluating the sufficiency and necessity of each demonstration. To further verify this claim, in Appendix A9.3, we duplicate the training dataset and evaluate GPT-neo's performance. Our results show that FEEDER minimizes the negative impact on the LLM, supporting its effectiveness in managing demonstration quality.

### 5.2. Performance on Bi-level Optimization

Here, we would like to verify whether our FEEDER can be beneficial to the fine-tuning setting. As formulated in Section 2, our pre-selection and the LLM fine-tuning can be integrated into a bi-level optimization framework. Specifically, in our evaluation, we assess the performance of FEEDER by initially fine-tuning the LLM on the pre-selected $\mathcal{D}_{\text{FEEDER}}$. Subsequently, we use the tuned LLM to generate a new $\mathcal{D}_{\text{FEEDER}}$, and evaluate the LLM within the in-context learning setting, using the new $\mathcal{D}_{\text{FEEDER}}$ as the selection pool. For comparison, our baseline is to initially fine-tune the LLM with $\mathcal{D}_{\text{TRAIN}}$ and then evaluate the LLM within ICL, using $\mathcal{D}_{\text{TRAIN}}$ as the selection pool.

The corresponding experimental results are reported in Table 3. FEEDER can improve the LLM fine-tuning performance within our bi-level framework. This emphasizes the potential for achieving enhanced performance by utilizing a small yet high-quality dataset for fine-tuning, while simultaneously reducing computational expenses, which aligns with the core-set selection literature (Feldman, 2020; Guo et al., 2022). By combining the results from Table 1 and Table 3, we can see that fine-tuning LLMs provides greater performance improvements compared to augmenting LLMs with contexts. Furthermore, our FEEDER achieves even better performance gains in the fine-tuning setting. One potential explanation is that in this scenario, fine-tuning can leverage input demonstrations more effectively than prompting can, and our high-quality FEEDER can therefore provide greater benefits.

We also investigate the performance of FEEDER with varying the number of runs (i.e., $R$) and different tree depths (i.e., the number of rounds $K$) in Appendix A9.2. FEEDER's

*Table 4.* Performance comparisons on reasoning GSM8K dataset and scientific question-answering GPQA dataset are conducted in the in-context learning setting. We report both the mean and variance of accuracy using 8 different seeds and 5 different permutations of n-shots.

| $\Psi_{\text{LLM}}(\cdot)$ | $\mathcal{D}$ | $n$ | GSM8K | | | | GPQA | | | |
|---|---|---|---|---|---|---|---|---|---|---|
| | | | Random | Similarity | Diversity | Uncertainty | Random | Similarity | Diversity | Uncertainty |
| Qwen-2.5 (32B) | $\mathcal{D}_{\text{TRAIN}}$ | 1 | 81.01 (4.31) | 82.14 (0.18) | 82.14 (0.18) | 81.02 (5.24) | 40.10 (1.64) | 42.04 (0.19) | 42.04 (0.19) | 42.00 (5.04) |
| | | 2 | 83.25 (5.63) | 84.19 (0.56) | 84.68 (0.56) | 84.01 (4.27) | 42.06 (3.65) | 44.52 (0.36) | 45.23 (0.41) | 42.85 (7.77) |
| | | 5 | 85.20 (3.52) | 89.52 (0.58) | 89.61 (0.84) | 86.02 (3.90) | 43.33 (5.88) | 46.20 (1.05) | 46.81 (1.95) | 46.70 (5.66) |
| | | 10 | 86.21 (4.28) | 90.41 (0.63) | 89.92 (0.58) | 90.20 (3.84) | 43.23 (5.21) | 46.85 (1.04) | 46.71 (0.98) | 45.28 (0.89) |
| | $\mathcal{D}_{\text{FEEDER}}$ | 1 | **81.85** (2.61) | **83.52** (0.11) | **83.52** (0.11) | **81.91** (3.78) | **40.51** (5.95) | **42.71** (0.66) | **42.71** (0.66) | **41.22** (6.87) |
| | | 2 | **84.52** (6.48) | **85.71** (0.33) | **86.03** (0.56) | **84.80** (5.65) | **43.50** (0.52) | **45.81** (0.38) | **45.66** (3.65) | **43.20** (6.58) |
| | | 5 | **86.70** (7.11) | **90.20** (0.96) | **90.12** (0.88) | **88.82** (4.38) | **44.94** (4.12) | **48.03** (5.58) | **48.08** (6.21) | **48.95** (4.66) |
| | | 10 | **87.58** (7.22) | **91.23** (3.45) | **90.71** (4.24) | **91.96** (7.71) | **44.55** (2.11) | **47.80** (4.52) | **47.93** (3.54) | **47.91** (6.26) |

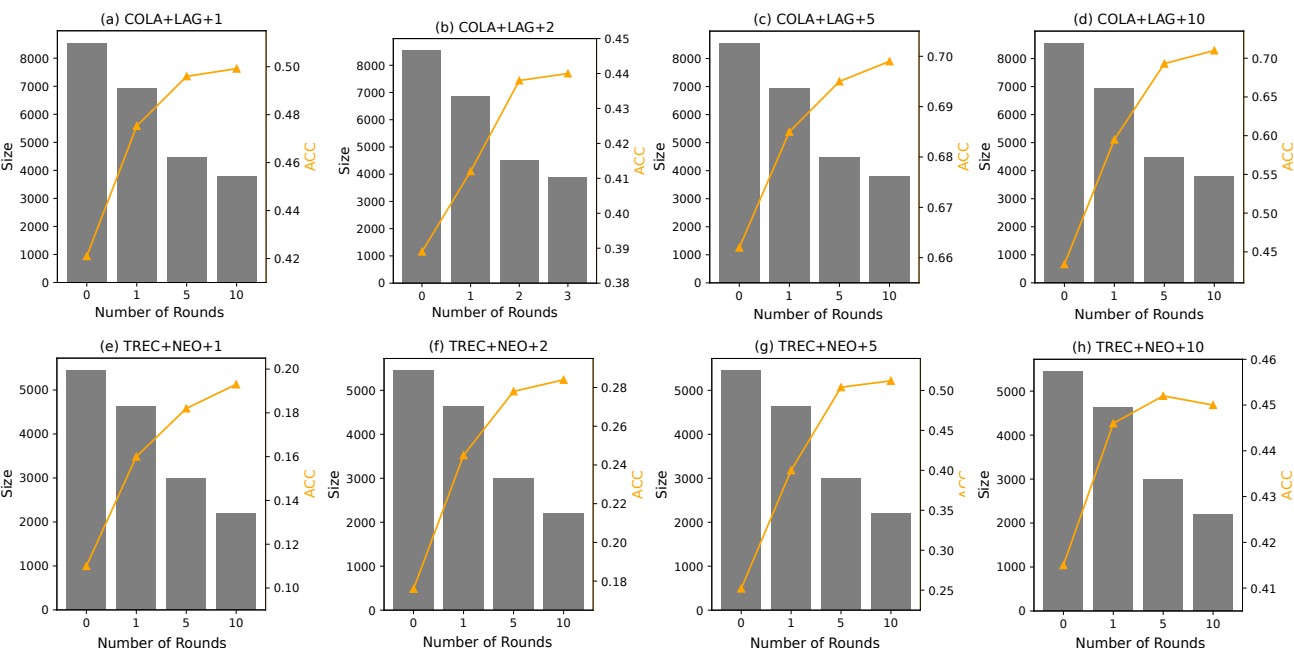

*Figure 3.* Performance comparisons for running our approximation algorithm to pre-select FEEDER with different runs $R$ are evaluated in terms of accuracy (denoted as ACC) with Random as the demonstration selector and the size of the resulting FEEDER subset (denoted as Size). Each sub-figure is entitled with Dataset+LLM base+n shots.

performance first rises and then drops with increasing $R$ or $K$. These findings further verify that identifying a representative subset from the training dataset - either by increasing the number of runs or the number of rounds - can significantly enhance the performance of the LLM. However, overly narrow subsets may limit the potential performance gains. This trend may be summarized as a trade-off between data quantity and data quality, and similar observations are reported in Chen et al. (2023).

Furthermore, we analyze scaling up FEEDER into larger LLMs and real-world datasets in Appendix A6.

### 5.3. Case Study

Subsequently, we conduct a case study to substantiate the central proposition of this paper: whether the assessment of the quality of demonstrations should depend on the specific

LLM in use. We consider the factual error made by Google Bard in the first demo[1]. We further prompt gpt-3.5-turbo to generate 5 *sufficient and necessary statements* for the fact. We evaluate separately using these statements as a prompt to gpt-3.5-turbo, and find that either one of the generated statements is sufficient and necessary to answer the question "What took the very first pictures of a planet outside of our own solar system?" We then evaluate the performance of gpt-j-6b with the above 5 statements, and find that only the 1-st or the 5-th statement is sufficient and necessary instance to answer the above question. Combining the results of gpt-j-6b and gpt-3.5-turbo verifies one of the core insights of our paper: the evaluation of prompting

---

[1] https://www.theverge.com/2023/2/8/23590864/google-ai-chatbot-bard-mistake-error-exoplanet-demo

a demonstration should consider the specific LLM in use. Refer to the detailed description in Appendix A10.2.

### 5.4. Time Complexity

Here, we provide some empirical evidence of the time complexity of the proposed method. As summarized in Algorithm 2 in Appendix A3 and discussed in Section 4, there are two key hyperparameter settings for reducing the time cost of Algorithm 2: the number of iterations (i.e., $K$) and the number of rounds (i.e., $R$). In our main experiment, we set $K = 1$ and $R = 1$, meaning that we perform only one-shot inference for sufficiency checks in each round of Algorithm 2 and execute the algorithm for a single round. We investigate the performance differences arising from varying $K$ and $R$ in Appendix A9.2 and Section 5.2 respectively. Additionally, we report the time complexity associated with different values of $K$ and $R$ on COLA and TREC datasets in Figure 4. From the figure, we observe that as the number of samples decreases, the time consumption of Algorithm 2 also decreases. Furthermore, we note that increasing the number of rounds has a great impact on reducing the time complexity. This may be attributed to the fact that two-shot inference for sufficiency-satisfying Eq. (7)-is significantly more challenging than a one-shot inference check. By further combining Figure 4 and Figure 3 in Section 5.1, we observe that the time consumption is nearly linear with respect to the size of the data samples.

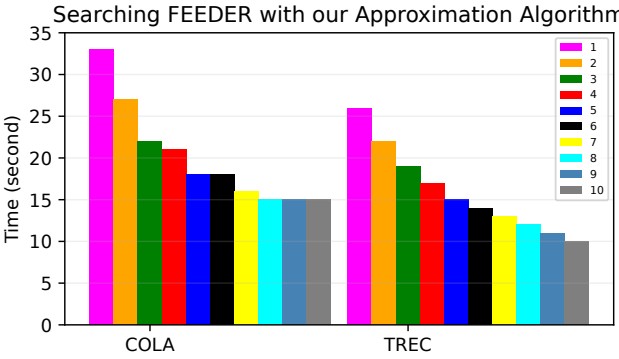

*Figure 4.* Time complexity of searching FEEDER using our approximation algorithm for different runs on COLA and TREC datasets using varying the number of rounds $R$ and varying the number of rounds $K$.

Consider two hyper-parameter settings in our approximation algorithm: the number of rounds $R$ and the number of iterations $K$, both designed to balance performance and computational efficiency. As detailed in Figures 3 and 4, the time complexity of our method scales almost linearly with the number of samples, making these parameters critical for practical applications. While Figure 3 illustrates the performance changes across different values of $R$, Figure A6 in Appendix A5.4 explores the impact of varying $K$. Inter-

estingly, Figure A6 reveals a similar but more robust trend compared to Figure 3. This robustness could be attributed to the inherent strength of the two-shot inference process for sufficiency, as defined in Eq. (7). The two-shot inference introduces a more rigorous evaluation mechanism than the one-shot inference check, enabling a stronger filtering of unnecessary samples.

In practice, we deploy our approximation algorithm with $K = 1$ and $R = 1$, which provides an optimal trade-off between computational efficiency and model performance. This configuration ensures that the pre-selection process remains practical while maintaining competitive accuracy.

## 6. Conclusion and Future Work

In this paper, we present a novel demonstration *pre-selector* FEEDER, designed to leverage LLM's capabilities and domain knowledge to identify high-quality demonstration and provide an approximate approach for their discovery. Our experimental results showcase the significant advantages of FEEDER across diverse LLM bases in both ICL and bi-level optimization for fine-tuning LLMs. In the future, it would be valuable to explore the use of larger LLMs and extend the applications of FEEDER to areas such as data safety and data management.

## Acknowledgment

The Shanghai Jiao Tong University team is partially supported by National Key R&D Program of China (No. 2022ZD0114804), Shanghai Municipal Science and Technology Major Project (No. 2021SHZDZX0102) and National Natural Science Foundation of China (No. 62322603, No. 62177033). This work is also supported in part by National Natural Science Foundation of China (No. 623B2002) and Beijing Natural Science Foundation (L257007).

## Impact Statement

The objective of this paper is to develop a pre-selection method over the training data as an intermediary process to enhance the accuracy of factual knowledge in the model's outputs. It is essential to note that our FEEDER, pre-selected from the training dataset without external trustworthy corpora, relies on the capability of the given LLM itself. This characteristic may potentially amplify existing biases in the model weights of LLMs.

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

# A1. Connections to Existing Approaches

## A1.1. Connections to Causality

The concepts of sufficiency and necessity have a broad application scope, especially in causality (Pearl, 1980; 2009), where sufficiency and necessity are proposed to define the causal relationship between two binary variables. Let $X$ and $Y$ denote a pair of variables. Then, the probability of sufficiency measures the capacity of setting $X = \texttt{true}$ to produce $Y = \texttt{true}$, while the probability of necessity measures the changing the value of $X$ from $X = \texttt{true}$ to $X = \texttt{false}$ would cause the value of $Y$ changing from $Y = \texttt{true}$ to $Y = \texttt{false}$.

In this paper, we adopt the concepts of sufficiency and necessity in the context of demonstration selection, where we investigate whether prompting certain data points is sufficient or necessary for the given LLM to generate correct answers for input questions. For this purpose, we introduce the plugging-in operation, denoted as $\texttt{plug}(\cdot)$, to examine sufficiency, and the unplugging operation, denoted as $\texttt{unplug}(\cdot)$, to examine necessity. Both of these operations are analogous to the do operation in causality, denoted as $\texttt{do}(\cdot)$, which indicates that the system operates under the condition that certain variables are controlled by external forces. To be more specific, in our setting, the external force can be explained as follows. We have the choice to either plug in or unplug certain data points, thereby altering what is already plugged into the LLM. Our approach shares similarities with the counterfactual idea in causality, which explores hypothetical scenarios by considering what might happen if certain variables are set with different values. In our case, we investigate the impact of plugged-in data that includes data points differing from the historical (i.e., factual) setting. Notably, a significant distinction between our approach and the counterfactual setting in causality lies in the fact that we do not need to estimate "counterfactual" situations; instead, we can directly conduct evaluations.

## A1.2. Connections to Demonstration Selection

In the context of few-shot inference, a central challenge lies in selecting the appropriate training samples as extra input during inference. These samples are often referred to as demonstrations or prompts (Levy et al., 2022; Liu et al., 2021; Dong et al., 2022). The underlying assumption is that the training dataset serves as a support set (Yosida, 2012) for test samples. Previous studies (Wang et al., 2022; Rubin et al., 2021) have demonstrated that introducing similar training samples can enhance the performance of LLMs on test instances. Gao et al. (2023) enhances these approaches by retrieving candidates whose ground label lies in top-2 zero-shot predictions. However, as pointed out in Levy et al. (2022), existing methods often treat each data point in isolation, neglecting the collective impact of multiple data points. For instance, retrievers based on similarity metrics may select redundant data points together. To address this limitation, Levy et al. (2022) proposes to consider the diversity among the data points, to avoid the case where too "similar" data points are selected together. Further, Rubin et al. (2021) trains an LLM as a contrastive scorer as well as a demonstration referrer, and Li et al. (2023) advances this framework through unified training across various datasets.

In this paper, we present a novel perspective, asserting that the quality of demonstrations is contingent on the specific LLM in use. Namely, a high-quality demonstration for one LLM might be deemed low-quality for another. Leveraging this insight, we introduce sufficiency and necessity as new set-level metrics. Our approach offers several advantages: Firstly, sufficiency and necessity measure the quality of data points based on the specific LLM, in contrast to generic similarity and diversity metrics. Secondly, our proposed sufficiency and necessity extend to the set level, enabling the consideration of data points as a cohesive whole. In our framework, "similarity" is akin to "sufficiency" signifying that plugging in data points can enhance LLM performance, while "diversity" is akin to "necessity" suggesting that each data point should play an indispensable role.

Recent studies (Xia et al., 2024; Marion et al., 2023) focus on mining training examples for fine-tuning on specific tasks, while Wang et al. (2024) extends this idea to in-context learning. Unlike these approaches, which use LLMs to select demonstrations tailored to specific test datasets, our work leverages LLMs as demonstration pre-selectors, identifying a core subset of the training data that remains independent of the test datasets, thus eliminating the need for re-computation across different test datasets.

## A1.3. Connections to Core-set Selection

Core-set selection (Feldman, 2020; Guo et al., 2022), a long-standing problem in machine learning, focuses on identifying a subset of the most informative training samples. Previous research (Dor et al., 2020) has surveyed and evaluated state-of-the-art approaches for models like BERT (Devlin et al., 2018), encompassing strategies such as random sampling, uncertainty-sampling (using entropy metric) (Lewis, 1995; Gal & Ghahramani, 2016) and diversity sampling (using diversity metric) (Gissin & Shalev-Shwartz, 2019).

FEEDER, in contrast to these prior papers mainly using active learning, is designed to select core sets, which can serve as additional input contexts (i.e., in-context learning setting) or be used for fine-tuning LLMs (i.e., fine-tuning setting). FEEDER defines "informative training samples" as those samples that specifically enhance the LLM's performance on a given task.

## A1.4. Connections to Prompt Optimization

Prompting provides a natural way for humans to interact with; and due to its flexibility, prompting has been widely used as a genre method for various natural language processing tasks (Schick & Schütze, 2020; Brown et al., 2020; Sanh et al., 2021). However, using prompting effectively with LLMs requires careful design, either done manually (Reynolds & McDonell, 2021) or automatically (Gao, 2021; Shin et al., 2020), as LLMs do not interpret prompts in the same way humans do (Webson & Pavlick, 2021; Lu et al., 2021; Cheng et al., 2025). While numerous successful methods (Liu et al., 2021; Lester et al., 2021; Qin & Eisner, 2021) for prompt tuning rely on optimizing a continuous space through gradient-based techniques, this approach becomes impractical as many powerful LLMs are only accessible through APIs that may not offer gradient access.

Our FEEDER approach can be seen as a discrete pre-search method for prompts, distinct from existing methods for prompt generation (Gao, 2021; Ben-David et al., 2021), prompt scoring (Davison et al., 2019), and prompt paraphrasing (Jiang et al., 2020; Yuan et al., 2021), which aim to optimize instructions by directly searching the natural language hypothesis space. Instead, our approach leverages the causal dependencies among candidate demonstrations, focusing on searching for the most informative demonstrations as prompts, in terms of sufficiency and necessity.

## A2. A Family of Analysis on Sufficiency and Necessity Metrics

**Notations.** Let $X, C$ denote variables for the input and the context (i.e., previously plugged-in demonstrations). We use $Y$, a boolean variable, to denote whether the output to the input is correct. Concretely, we use $Y_{\boldsymbol{x}} = 1$ to denote $Y = 1|X = \boldsymbol{x}$, meaning that the LLM generates the correct output to the input $\boldsymbol{x}$. Similarly, $Y_{\boldsymbol{x}} = 0$, equivalent to $Y = 0|X = \boldsymbol{x}$, indicates that the LLM produces the incorrect output to $\boldsymbol{x}$. For clarity, we introduce $S$, a variable to record the original status of the LLM before *new* plug-in and unplug operations (denoted as $\texttt{plug}(\cdot)$ and $\texttt{unplug}(\cdot)$ respectively), e.g., $C = ((\boldsymbol{x}, \boldsymbol{y})), S = (Y_{\boldsymbol{x}} = 1)$ means that without plugging-in any new data or unplugging any plugged-in data, the plugged-in data is $(\boldsymbol{x}, \boldsymbol{y})$ and the LLM's performance is $Y_{\boldsymbol{x}} = 1$.

### A2.1. Instance Level Metrics

Here, two instances are considered, represented as $(\boldsymbol{x}_n, \boldsymbol{y}_n)$ and $(\boldsymbol{x}_m, \boldsymbol{y}_m)$.

The sufficiency metric is introduced to assess whether plugging in one data point is sufficient to enable the LLM to generate the correct output for the other one. Formally, the sufficiency relationship is defined as follows:

**Definition A1** (Instance-level Sufficiency Metric). *Given tuple* $(X, Y, C, S)$*, data point* $(\boldsymbol{x}_n, \boldsymbol{y}_n)$ *is sufficient for* $(\boldsymbol{x}_m, \boldsymbol{y}_m)$*, if the following equation holds:*

$$Y_{\boldsymbol{x}_m} = 1|\texttt{plug}((\boldsymbol{x}_n, \boldsymbol{y}_n)); C, S, \qquad (9)$$

*where* $(\boldsymbol{x}_n, \boldsymbol{y}_n)$ *is not included in* $C$ *and* $S$ *can be any value. It means that when plugging in* $(\boldsymbol{x}_n, \boldsymbol{y}_n)$*, it would correct the LLM's answer to* $\boldsymbol{x}_m$*.*

**Example A1.** Let $\boldsymbol{x}_m, \boldsymbol{x}_n$ be *Which country does Sherlock Holmes live?* and *Which city does Sherlock Holmes live?* Then, after informing the LLM of the correct answer of $\boldsymbol{x}_n$ (e.g., $\boldsymbol{y}_n$ is *Sherlock Holmes lives in London*), the LLM can deduce the correct answer of $\boldsymbol{x}_m$ (e.g., $\boldsymbol{y}_m$ is *Sherlock Holmes lives in the United Kingdom*). In this case, the LLM is using the city where Sherlock Holmes lives to infer the country in which he lives.

The necessity metric is introduced to assess whether the presence of one plugged-in data point is necessary for preserving the correct output in relation to another. Formally, this is expressed as:

**Definition A2** (Instance-level Necessity Metric). *Given tuple* $(X, Y, C, S)$*, we say that data point* $(\boldsymbol{x}_n, \boldsymbol{y}_n)$ *is necessary for* $(\boldsymbol{x}_m, \boldsymbol{y}_m)$*, if the following equation holds:*

$$Y_{\boldsymbol{x}_m} = 0|\texttt{unplug}((\boldsymbol{x}_n, \boldsymbol{y}_n)); C, S, \qquad (10)$$

*where* $(\boldsymbol{x}_n, \boldsymbol{y}_n)$ *is included in* $C$ *and* $S = (Y_{\boldsymbol{x}_m} = 1)$*. It means that before unplugging* $(\boldsymbol{x}_n, \boldsymbol{y}_n)$*, the LLM's answer to* $\boldsymbol{x}_m$ *is correct. However, when we do unplug* $(\boldsymbol{x}_n, \boldsymbol{y}_n)$*, it causes the LLM to offer an incorrect output to* $\boldsymbol{x}_m$*.*

**Example A2.** Consider $\boldsymbol{x}_m$ as *Which city does Sherlock Holmes live?* and $\boldsymbol{x}_n$ as *What is the detailed address of Sherlock Holmes lives?*. Assume the LLM has no prior knowledge about Sherlock Holmes until the introduction of the plugged-in data $(\boldsymbol{x}_n, \boldsymbol{y}_n)$, where $\boldsymbol{y}_n$ is *221B Baker Street, London*. After plugging in $(\boldsymbol{x}_n, \boldsymbol{y}_n)$, the LLM is capable of generating the correct output $\boldsymbol{y}_m$ (e.g., *Sherlock Holmes lives in London*) in response to $\boldsymbol{x}_m$. If we were to unplug $(\boldsymbol{x}_n, \boldsymbol{y}_n)$, the LLM would provide an incorrect output for $\boldsymbol{x}_m$, such as *Sherlock Holmes lives in New York*.

In an ideal scenario, ensuring optimal LLM performance entails the extraction of data points that are both sufficient and necessary.

**Definition A3** (Instance-level Sufficiency and Necessity Metric). *Given tuple* $(X, Y, C)$*, we say that data point* $(\boldsymbol{x}_n, \boldsymbol{y}_n)$ *is both sufficient and necessary for* $(\boldsymbol{x}_m, \boldsymbol{y}_m)$*, if the following equation holds:*

$$\begin{aligned}&\Big(Y_{\boldsymbol{x}_m} = 1|\texttt{plug}((\boldsymbol{x}_n, \boldsymbol{y}_n)); C = \emptyset\Big) \\ &\wedge \Big(Y_{\boldsymbol{x}_m} = 0|\texttt{unplug}((\boldsymbol{x}_n, \boldsymbol{y}_n)); C = ((\boldsymbol{x}_n, \boldsymbol{y}_n))\Big),\end{aligned} \qquad (11)$$

*which indicates that plugging in data point $(\boldsymbol{x}_n, \boldsymbol{y}_n)$ can respond to the LLM's answering $\boldsymbol{x}_m$ in both ways. We omit $S$ here, because we can derive the original status of the necessary instance based on the condition of the sufficiency instance.*

We further demonstrate that neither of the aforementioned quantities (i.e., sufficiency and necessity) is adequate for determining the other, indicating that they are not entirely independent. This is illustrated in the following lemma.

**Lemma 1** (Connection between Sufficiency and Necessity). *Supposing that we only consider using the data point $(\boldsymbol{x}_n, \boldsymbol{y}_n)$ as the plug in data, and only care about the LLM's performance regarding the input question $\boldsymbol{x}_m$, then overall there are only two situations here: (i) $(\boldsymbol{x}_n, \boldsymbol{y}_n)$ is plugged-in, and (ii) $(\boldsymbol{x}_n, \boldsymbol{y}_n)$ is not plugged-in. Based on the above assumption, we re-write (i) as plugging-in $(\boldsymbol{x}_n, \boldsymbol{y}_n)$ when there is no plugged-in data (i.e., $\texttt{plug}((\boldsymbol{x}_n, \boldsymbol{y}_n)); C = \emptyset$, and re-write (ii) as unplugging $(\boldsymbol{x}_n, \boldsymbol{y}_n)$ when there is plugged-in data $(\boldsymbol{x}_n, \boldsymbol{y}_n)$ (i.e., $\texttt{unplug}((\boldsymbol{x}_n, \boldsymbol{y}_n)); C = ((\boldsymbol{x}_n, \boldsymbol{y}_n)))$. For convenience, we use $E^*$ and $E$ to denote (i) and (ii) respectively; and we use $Y^*$ and $Y$ to denote $Y_{\boldsymbol{x}_1} = 1$ and $Y_{\boldsymbol{x}_1} = 0$. Then, we have: $E^* \vee E = \texttt{true}$, $E^* \wedge E = \texttt{false}$, $Y^* \vee Y = \texttt{true}$, $Y^* \wedge Y = \texttt{false}$.*

*We define* PS *as the probability of being sufficient as:*

$$\begin{aligned} \texttt{PS} :=& \Pr\Big(Y_{\boldsymbol{x}_m} = 1 | \texttt{plug}((\boldsymbol{x}_n, \boldsymbol{y}_n)); C = \emptyset\Big) \\ =& \Pr(Y^* | E^*). \end{aligned} \tag{12}$$

*We define* PN *as the probability of being necessary as:*

$$\begin{aligned} \texttt{PN} :=& \Pr\Big(Y_{\boldsymbol{x}_m} = 0 | \texttt{unplug}((\boldsymbol{x}_n, \boldsymbol{y}_n)); C = ((\boldsymbol{x}_n, \boldsymbol{y}_n))\Big) \\ =& \Pr(Y | E). \end{aligned} \tag{13}$$

*We further define* PNS *as the probability of being sufficient and necessary as:*

$$\texttt{PNS} := \Pr(Y^* | E^*, Y | E). \tag{14}$$

*Then,* PS, PN, PNS *satisfy the following relationship:*

$$\texttt{PSN} = \Pr(Y, E) \cdot \texttt{PS} + \Pr(Y^*, E^*) \cdot \texttt{PN}. \tag{15}$$

*Proof.* Based on the earlier delineation of $Y^*$, $Y$, $E^*$, and $E$, we can express:

$$\begin{aligned} Y^* | E^* \wedge Y | E =& (Y^* | E^* \wedge Y | E) \wedge (E \vee C^*) \\ =& (Y^* | E^* \wedge Y \wedge E) \vee (Y | E \wedge Y^* \wedge E^*). \end{aligned} \tag{16}$$

Taking probabilities on both sides and using the disjointedness of $E^*$ and $E$, we have:

$$\begin{aligned} \texttt{PSN} =& \Pr(Y^* | E^*, Y | E) \\ =& \Pr(Y | E, Y^*, E^*) + \Pr(Y^* | E^*, Y, E) \\ =& \Pr(Y, E) \cdot \texttt{PS} + \Pr(Y^*, E^*) \cdot \texttt{PN}. \end{aligned} \tag{17}$$

$\square$

## A2.2. Set Level Metrics

We extend Definitions A1 and A2 to the set level as:

**Definition A4** (Set-level Sufficiency Metric). *Given tuple $(X, Y, C, S)$, the input set $\mathcal{D}_{\texttt{IN}}$ is sufficient for output set $\mathcal{D}_{\texttt{OUT}}$, if the following equation holds:*

$$Y_{(\{\boldsymbol{x}_n | \boldsymbol{x}_n \in \mathcal{D}_{\texttt{OUT}}\})} = \mathbf{1}_{|\mathcal{D}_{\texttt{OUT}}|} | \texttt{plug}(\mathcal{D}_{\texttt{IN}}); C, S. \tag{18}$$

*where $\mathcal{D}_{\texttt{IN}}$ is not included in $C$ and $S$ can be any value. $\mathbf{1}_{|\mathcal{D}_{\texttt{OUT}}|}$ denotes $\mathbf{1}_{|\mathcal{D}_{\texttt{OUT}}|}$-dimensional vectors whose elements are all 1s. It indicates that when plugging in $\mathcal{D}_{\texttt{IN}}$, it guarantees that the LLM's output to any input question in $\mathcal{D}_{\texttt{OUT}}$ is correct.*

**Definition A5** (Set-level Necessity Metric). *Given tuple $(X, Y, C, S)$, the input set $\mathcal{D}_{\texttt{IN}}$ is necessary for output set $\mathcal{D}_{\texttt{OUT}}$, if the following equation holds:*

$$Y_{(\{\boldsymbol{x}_n | \boldsymbol{x}_n \in \mathcal{D}_{\texttt{OUT}}\})} \neq \mathbf{1}_{|\mathcal{D}_{\texttt{OUT}}|} | \texttt{unplug}(\mathcal{D}'_{\texttt{IN}}); C, S, \tag{19}$$

*where $\mathcal{D}_{\texttt{IN}}$ is included in $C$, $S = (Y_{(\{\boldsymbol{x}_n | \boldsymbol{x}_n \in \mathcal{D}_{\texttt{OUT}}\})} = \mathbf{1}_{|\mathcal{D}_{\texttt{OUT}}|})$, and $\mathcal{D}'_{\texttt{IN}}$ can be any subset of $\mathcal{D}_{\texttt{IN}}$. $\mathbf{1}_{|\mathcal{D}_{\texttt{OUT}}|}$ denotes $\mathbf{1}_{|\mathcal{D}_{\texttt{OUT}}|}$-dimensional vectors whose elements are all 1s. It means that before unplugging any subset of $\mathcal{D}_{\texttt{IN}}$, there is plugged-in data $\mathcal{D}_{\texttt{IN}}$ and the LLM's output to any input in $\mathcal{D}_{\texttt{OUT}}$ is correct. When we unplug any subset of $\mathcal{D}_{\texttt{IN}}$, then it would cause the LLM's output to at least one input in $\mathcal{D}_{\texttt{OUT}}$ to be incorrect.*

From the above description, when we refer to a set as a sufficient set, we are stating that the collective set of data points is sufficient. On the other hand, when we characterize a set as a necessary set, we mean that each individual data point within the set is necessary.

**Example A3.** Let $\mathcal{D}_{\texttt{OUT}} = \{(\boldsymbol{x}_m, \boldsymbol{y}_m)\}$ and $\mathcal{D}_{\texttt{IN}} = \{(\boldsymbol{x}_i, \boldsymbol{y}_i), (\boldsymbol{x}_j, \boldsymbol{y}_j)\}$. We assign $\boldsymbol{x}_m$ and $\boldsymbol{y}_m$ as *Which country does Sherlock Holmes live?* and *Sherlock Holmes lives in the United Kingdom*. Let $\boldsymbol{x}_i$ and $\boldsymbol{y}_i$ denote *Which street does Sherlock Holmes live?* and *Baker street*. We assign $\boldsymbol{x}_j$ and $\boldsymbol{y}_j$ as *Where is Baker street?* and *Bake street is located in London*. Supposing that the LLM does not know that Bake Street is located in the United Kingdom, then solely plugging in either $(\boldsymbol{x}_i, \boldsymbol{y}_i)$ or $(\boldsymbol{x}_j, \boldsymbol{y}_j)$ is not sufficient for the LLM to get the right answer to the input question $\boldsymbol{x}_m$. In this regard, it is easy to derive that $\mathcal{D}_{\texttt{IN}}$ is both a sufficient and necessary set for $\mathcal{D}_{\texttt{OUT}}$ when both (i) plugging in $\mathcal{D}_{\texttt{IN}}$ is sufficient to maintain the right answer for $\mathcal{D}_{\texttt{OUT}}$; and (ii) unplugging any subset of $\mathcal{D}_{\texttt{IN}}$ can not maintain the right answer for $\mathcal{D}_{\texttt{OUT}}$, are satisfied.

## A2.3. FEEDER Set

Next, we explore the problem of defining a subset within the given dataset $\mathcal{D}_{\texttt{TRAIN}}$ that is both sufficient and necessary to represent $\mathcal{D}_{\texttt{TRAIN}}$. This subset is termed FEEDER (FEw yet Essential DEmonstRations).

**Definition A6** (FEEDER Set). *Given tuple $(X, Y, C, S)$ and $\mathcal{D}_{\text{TRAIN}}$, a subset of $\mathcal{D}_{\text{TRAIN}}$, is considered as a FEEDER set (denoted as $\mathcal{D}_{\text{FEEDER}}$), if the following conditions are satisfied:*

*(i)* $Y_{(\boldsymbol{x}_1..., \boldsymbol{x}_N)} = \boldsymbol{1}_N | \text{plug}(\mathcal{D}_{\text{FEEDER}}); C = \emptyset, S = (Y_{(\boldsymbol{x}_1..., \boldsymbol{x}_N)} \neq \boldsymbol{1}_N)$ *holds.*

*(ii)* $Y_{(\boldsymbol{x}_1..., \boldsymbol{x}_N)} \neq \boldsymbol{1}_N | \text{unplug}(\mathcal{D}'_{\text{FEEDER}}); C = \mathcal{D}_{\text{TRAIN}}, S = (Y_{(\boldsymbol{x}_1..., \boldsymbol{x}_N)} = \boldsymbol{1}_N)$ *holds for any subset of $\mathcal{D}_{\text{FEEDER}}$ (denoted as $\mathcal{D}'_{\text{FEEDER}}$).*

$\boldsymbol{1}_N$ *denotes $N$-dimensional vectors whose elements are all 1s. (i) and (ii) respectively imply that plugging in $\mathcal{D}_{\text{FEEDER}}$ is sufficient and necessary to maintain the LLM generating correct output.*

**Example A4.** If we merge $\mathcal{D}_{\text{IN}}$ and $\mathcal{D}_{\text{OUT}}$ exemplified in Example A3 into one set $\mathcal{D}$, namely let $\mathcal{D} = \mathcal{D}_{\text{IN}} \cup \mathcal{D}_{\text{OUT}}$, then in this case, it is easy to derive that $\mathcal{D}_{\text{IN}}$ is a FEEDER set (denoted as $\mathcal{D}_{\text{FEEDER}}$) for $\mathcal{D}$.

The above definition of the FEEDER set is overly strict, as identifying it would require enumerating all possible subsets. To mitigate this complexity, we introduce the following approximation algorithm.

---

**Algorithm 2** Approximation Algorithm for FEEDER

**Input:** Training dataset $\mathcal{D}_{\text{TRAIN}}$.
**Output:** An approximated FEEDER set $\widetilde{\mathcal{D}}_{\text{FEEDER}}$.
Initialize $k = 1$.
Initialize $\mathscr{W}_0 = \{\mathcal{W}_n = \{(\boldsymbol{x}_n, \boldsymbol{y}_n)\} | (\boldsymbol{x}_n, \boldsymbol{y}_n) \in \mathcal{D}_{\text{TRAIN}}\}$.
**repeat**
    **for** *each pair $(\mathcal{W}_i, \mathcal{W}_j)$ where $\mathcal{W}_i, \mathcal{W}_j \in \mathscr{W}_{k-1}$* **do**
        Check $Y_{(\{\boldsymbol{x}_n | \boldsymbol{x}_n \in \mathcal{W}_j\})} = \boldsymbol{1}_{|\mathcal{W}_j|} | \text{plug}(\mathcal{W}_i); C, S$ (a), where $C = \emptyset$ and $S$ can be any value.
        Check $Y_{(\{\boldsymbol{x}_n | \boldsymbol{x}_n \in \mathcal{W}_i\})} = \boldsymbol{1}_{|\mathcal{W}_i|} | \text{plug}(\mathcal{W}_j); C, S$ (b), where $C = \emptyset$ and $S$ can be any value.
        **Case I** (Both (a) and (b) hold), if $|\mathcal{W}_i| \geq |\mathcal{W}_j|$, append $\mathcal{W}_j$ to $\mathscr{W}_k$; otherwise, append $\mathcal{W}_i$ to $\mathscr{W}_k$.
        **Case II** (Either one of (a) and (b) holds), if (a) holds, append $\mathcal{W}_i$ to $\mathscr{W}_k$; otherwise, append $\mathcal{W}_j$ to $\mathscr{W}_k$.
        **Case III** (Neither (a) nor (b) hold), append $\mathcal{W}_i \cup \mathcal{W}_j$ to $\mathscr{W}_k$.
        Remove $\mathcal{W}_i, \mathcal{W}_j$ from $\mathscr{W}_{k-1}$, i.e., $\mathscr{W}_{k-1} = \mathscr{W}_{k-1} - \{\mathcal{W}_i, \mathcal{W}_j\}$.
    **if** $|\mathscr{W}_{k-1}| = 1$ **then**
        Append only element in $\mathscr{W}_{k-1}$ to $\mathscr{W}_k$.
    Grow tree from bottom to top via $k = k + 1$.
**until** $|\mathscr{W}_k| = 1$, *and we assume the current round is $K$*;
Let $\mathcal{W}_{\text{SUFFICIENT}}$ denote only one element (i.e. the root node) in $\mathscr{W}_K$.
Assign $\widetilde{\mathcal{D}}_{\text{FEEDER}}$ as $\mathcal{W}_{\text{SUFFICIENT}}$, i.e., $\mathcal{D}_{\text{OUT}} = \mathcal{W}_{\text{SUFFICIENT}}$.

---

**Algorithm 3** Exact Algorithm for FEEDER

**Input:** Training dataset $\mathcal{D}_{\text{TRAIN}}$.
**Output:** An exact FEEDER set $\widetilde{\mathcal{D}}_{\text{FEEDER}}$.
Initialize $k = 1$.
Initialize $\mathscr{H}_0 = \emptyset$.
**for** *each instance $(\boldsymbol{x}_n, \boldsymbol{y}_n) \in \mathcal{D}_{\text{TRAIN}}$* **do**
    Check $Y_{(\{\boldsymbol{x}_{n'} | \boldsymbol{x}_{n'} \in \mathcal{D}_{\text{TRAIN}}\})} = \boldsymbol{1}_{|\mathcal{D}_{\text{TRAIN}}|} | \text{unplug}((\boldsymbol{x}_n, \boldsymbol{y}_n)); C, S$ (a), $C = \mathcal{D}_{\text{TRAIN}}, S = (Y_{(\{\boldsymbol{x}_{n'} | \boldsymbol{x}_{n'} \in \mathcal{D}_{\text{TRAIN}}\})} = \boldsymbol{1}_{|\mathcal{D}_{\text{TRAIN}}|})$.
    If (a) holds, let $\mathcal{H}_n = \{(\boldsymbol{x}_n, \boldsymbol{y}_n)\}$ and append $\mathcal{H}_n$ to $\mathscr{H}_0$.
**repeat**
    **for** *each pair $(\mathcal{H}_i, \mathcal{H}_j)$ where $\mathcal{H}_i, \mathcal{H}_j \in \mathscr{H}_{k-1}$* **do**
        Check $Y_{(\{\boldsymbol{x}_n | \boldsymbol{x}_n \in \mathcal{D}_{\text{TRAIN}}\})} = \boldsymbol{1}_{|\mathcal{D}_{\text{TRAIN}}|} | \text{unplug}(\mathcal{H}_i \cup \mathcal{H}_j); C, S$ (b), where $C = \mathcal{D}_{\text{TRAIN}}$ and $S = (Y_{(\{\boldsymbol{x}_{n'} | \boldsymbol{x}_{n'} \in \mathcal{D}_{\text{TRAIN}}\})} = \boldsymbol{1}_{|\mathcal{D}_{\text{TRAIN}}|})$.
        If (b) holds, generate a new node $\mathcal{H}_i \cup \mathcal{H}_j$, append it to $\mathscr{H}_k$, and assign $\mathcal{H}_i \cup \mathcal{H}_j$ with MAINTAIN signals; otherwise, append $\mathcal{H}_i$ and $\mathcal{H}_j$ to $\mathscr{H}_k$.
    Assign $\mathcal{H}_{\text{MAX}} = \arg\max_{\mathcal{H}. \in \mathscr{H}_k} |\mathcal{H}.|$ with MAINTAIN signal.
    Remove the nodes without MAINTAIN signals in $\mathscr{H}_k$.
    Grow tree from bottom to top via $k = k + 1$.
**until** $|\mathscr{H}_k| = 1$ *where we assume the current round is $K$*;
Let $\mathcal{H}_{\text{UNNCESSARY}}$ denote only one element (i.e. the root node) in $\mathscr{H}_K$.
Assign $\widetilde{\mathcal{D}}_{\text{FEEDER}}$ as removing $\mathcal{H}_{\text{UNNCESSARY}}$ from $\mathcal{D}_{\text{TRAIN}}$, i.e., $\widetilde{\mathcal{D}}_{\text{FEEDER}} = \mathcal{D}_{\text{TRAIN}} - \mathcal{H}_{\text{UNNCESSARY}}$.

---

## A3. Approximated Extraction of FEEDER

**Definition A7** (Transitivity Inference over Sets). *We assume that sufficiency is transitive over sets. Formally, for any three sets, denoted as $\mathcal{D}_A$, $\mathcal{D}_B$, and $\mathcal{D}_C$, if $\mathcal{D}_A$ is a sufficient set of $\mathcal{D}_B$ and $\mathcal{D}_B$ is a sufficient set of $\mathcal{D}_C$, then we can conclude that $\mathcal{D}_A$ is a sufficient set of $\mathcal{D}_C$.*

We also establish case studies in Appendix A10.1 to verify the feasibility of the above assumption.

For convenience, we use $\mathcal{D}_{\text{IN}} = \{(\boldsymbol{x}_n, \boldsymbol{y}_n)\}_{n=1}^{N_{\text{IN}}}$ to denote the input set for our tree algorithm, and we use $\mathcal{D}_{\text{OUT}}$ to denote the corresponding output. The tree expands from the bottom to the top. We use the variable $K$ to represent the depth of these trees, which corresponds to the number of iterations. To be more specific, we use $k = 1, 2, \ldots, K$ to refer to each $k$-th iteration, and during each $k$-th iteration, we generate the $(k+1)$-th layer of the tree.

Concretely, we leverage the transitivity of sufficiency to build the tree, where each node is a set of samples. Formally, we denote $\mathscr{W}_k$ as the set of nodes after the $k$-th iteration. We initialize $\mathscr{W}_0$ by assigning all the candidate samples in $\mathcal{D}_{\text{IN}}$

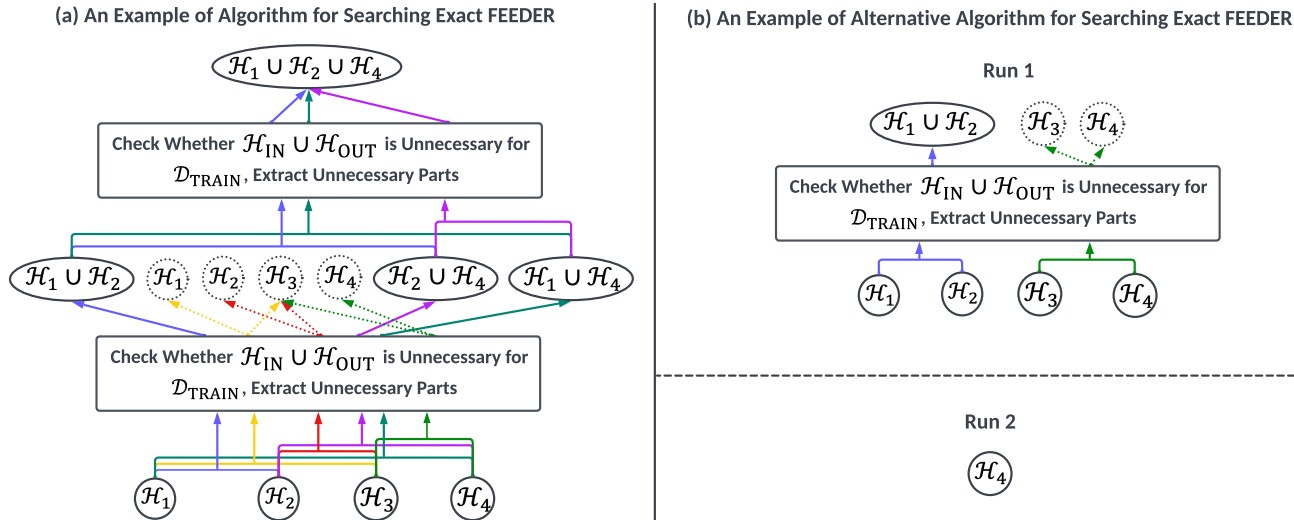

**(a) An Example of Algorithm for Searching Exact FEEDER**

**(b) An Example of Alternative Algorithm for Searching Exact FEEDER**

*Figure A1.* An illustrated example of our algorithm for deriving an exact FEEDER set. As shown in (a), we check the necessity of the conjunction of each pair of nodes, and we do not remove them from $\mathscr{H}$; instead, we assign MAINTAIN signals to newly generated nodes and the node with the maximum size, and those nodes without MAINTAIN signals, circled with dashed lines, would be removed from $\mathcal{H}$. In (b), we propose an alternative algorithm by removing nodes after checking the necessity, and we repeat the above process for multiple runs, at the beginning of each run, we unplug all the previously selected data points. The repeat should stop until there is no or only one node in $\mathscr{H}_0$ (i.e., $\mathcal{H}_4$), and therefore, the result in (b) is $\mathcal{H}_1 \cup \mathcal{H}_2 \cup \mathcal{H}_4$, same as the result in (a).

as the bottom nodes:

$$\mathscr{W}_0 := \{\mathcal{W}_n := \{(\boldsymbol{x}_n, \boldsymbol{y}_n)\}|(\boldsymbol{x}_n, \boldsymbol{y}_n) \in \mathcal{D}_{\text{IN}}\}. \quad (20)$$

During each $k$-th round, we generate $\mathscr{W}_k$ by examining the sufficiency relationship between every pair of nodes, denoted as $\mathcal{W}_i, \mathcal{W}_j \in \mathscr{W}_{k-1}$. In this evaluation, we assess whether the following equation holds true by assigning $\mathcal{W}_i$ and $\mathcal{W}_j$ as $\mathcal{W}_{\text{IN}}$ and $\mathcal{W}_{\text{OUT}}$, or vice versa.

$$Y_{(\{\boldsymbol{x}_n|\boldsymbol{x}_n \in \mathcal{W}_{\text{OUT}}\})} = \mathbf{1}_{|\mathcal{W}_{\text{OUT}}|} | \text{plug}(\mathcal{W}_{\text{IN}}); C, S, \quad (21)$$

where $C = \emptyset$ and $S$ is loosened to allow for any value. $\mathbf{1}_{|\mathcal{W}_{\text{OUT}}|}$ is $\mathbf{1}_{|\mathcal{W}_{\text{OUT}}|}$-dimensional vectors whose elements are all 1s. It signifies that plugging in $\mathcal{W}_{\text{IN}}$ is sufficient for the LLM to generate the correct output to any input in $\mathcal{W}_{\text{OUT}}$. In other words, once we have $\mathcal{W}_{\text{IN}}$ included in the plugged-in context, it is unnecessary to further include $\mathcal{W}_{\text{OUT}}$. Formally, we can derive the following equation from Eq. (21):

$$Y_{(\{\boldsymbol{x}_n|\boldsymbol{x}_n \in \mathcal{W}_{\text{OUT}}\})} = \mathbf{1}_{|\mathcal{W}_{\text{OUT}}|} | \text{unplug}(\mathcal{W}_{\text{OUT}}); C, S, \quad (22)$$

where $C = (\mathcal{W}_{\text{IN}} \cup \mathcal{W}_{\text{OUT}})$ and $S$ is loosened to be any value. Concretely, there are three possible scenarios by examining each pair of nodes in $\mathscr{W}_{k-1}$: (i) If both $\mathcal{W}_i$ and $\mathcal{W}_j$ are sufficient sets for each other, then we select the one with fewer elements to append to $\mathscr{W}_k$. (ii) If only one of $\mathcal{W}_i$ and $\mathcal{W}_j$ is a sufficient set for the other, then we append the sufficient set to $\mathscr{W}_k$. (iii) If neither $\mathcal{W}_i$ nor $\mathcal{W}_j$ is a sufficient set, we append $\mathcal{W}_i \cup \mathcal{W}_j$ to $\mathscr{W}_k$. After performing the above calculations for each pair of nodes, we remove them from $\mathscr{W}_{k-1}$.

When there is only one element left in $\mathscr{W}_{k-1}$, it is directly appended to $\mathscr{W}_k$. This process continues until $\mathscr{W}$ contains only one element, which is denoted as $\mathcal{W}_{\text{SUFFICIENT}} \in \mathscr{W}_K$. We then assign $\mathcal{D}_{\text{OUT}}$ as $\mathcal{D}_{\text{OUT}} = \mathcal{W}_{\text{SUFFICIENT}}$.

The time complexity of running the above tree algorithm for one round is $O(\log_2^{|\mathcal{D}_{\text{IN}}|})$ to generate one layer of the tree. To effectively remove the unnecessary part, we can repeat the above process for multiple rounds by using the output of the previous round (i.e., $\mathcal{D}_{\text{OUT}}$) as the input for the subsequent round (i.e., $\mathcal{D}_{\text{IN}}$) to build the tree from the bottom to the top. Therefore, the overall complexity of building a tree (i.e., executing Algorithm 2 for a single run) is $O(K \log_2^{|\mathcal{D}_{\text{IN}}|})$, where $K$ is the number of rounds. It follows that the time complexity for multiple runs is $O(RK \log_2^{|\mathcal{D}_{\text{IN}}|})$, where $R$ denotes the number of runs.

Our tree algorithm can also maintain the remaining set to be sufficient to represent the entire $\mathcal{D}_{\text{TRAIN}}$, as verified in the following proposition.

**Proposition 1** ($\widetilde{\mathcal{D}}_{\text{FEEDER}}$ obtained by Algorithm 2 is an Approximation of $\mathcal{D}_{\text{FEEDER}}$). *If we successively apply Algorithm 2 on $\mathcal{D}_{\text{TRAIN}}$ for multiple rounds to obtain a subset (denoted as $\widetilde{\mathcal{D}}_{\text{FEEDER}}$), then $\widetilde{\mathcal{D}}_{\text{FEEDER}}$ is sufficient to represent $\mathcal{D}_{\text{TRAIN}}$.*

*Proof.* In the tree generation process, each parent node is established as a sufficient set for every leaf node within the tree. More precisely, as shown in **Case I**, **Case II** and **Case III** of Algorithm 2, three scenarios exist for creating a parent

node for each pair of leaf nodes. In cases (i) and (ii), the parent node corresponds to the leaf node which serves as a sufficient set for the other node. In case (iii), the parent node results from the conjunction of two leaf nodes, inherently forming a sufficient set capable of representing either of the two leaf nodes.

According to our assumption of the sufficiency transitivity, for each data point in $\mathcal{D}_{\text{TRAIN}}$, the root node of the tree is a sufficient set for each leaf node in the tree. Formally, we have:

$$Y_{\{\boldsymbol{x}_n | \boldsymbol{x}_n \in \mathcal{D}_{\text{TRAIN}}\}} = \mathbf{1}_{|\mathcal{D}_{\text{TRAIN}}|} | \text{plug}(\widetilde{\mathcal{D}}_{\text{FEEDER}}); C, S, \quad (23)$$

where $C = \emptyset$ and $S$ can be any value. This means that the resulting set $\widetilde{\mathcal{D}}_{\text{FEEDER}}$ is a sufficient set of $\mathcal{D}_{\text{TRAIN}}$. $\qquad\square$

## A4. Exact Extraction of FEEDER

To extract an exact FEEDER set $\mathcal{D}_{\text{FEEDER}}$ from $\mathcal{D}_{\text{TRAIN}}$, we need to explicitly check the necessity among all the candidate samples, and remove those unnecessary parts. We do not directly apply this algorithm in practice, due to its high computation costs. We provide a solution for integrating the algorithm into our FEEDER and report the corresponding results in Appendix A7.

### A4.1. Exact Extraction of FEEDER via Necessity Checks

Our intuition behind constructing a tree for checking necessity is based on the inherent transitivity property of necessity. Formally, it can be expressed as: If unplugging $\mathcal{D}_{\text{A}}$ could cause the outputs to at least one input in $\mathcal{D}_{\text{C}}$ from correct to incorrect, then unplugging $\mathcal{D}_{\text{A}} \cup \mathcal{D}_{\text{B}}$ also can not maintain the outputs to all the input in $\mathcal{D}_{\text{C}}$ correct. Namely, if unplugging a subset would degrade the performance, then unplugging the whole set would also degrade the performance.

Similar to the tree for explicitly checking sufficiency introduced in Appendix A3, each node in the tree for checking necessity also represents a set of samples. For convenience, we also use $\mathcal{D}_{\text{IN}} = \{(\boldsymbol{x}_n, \boldsymbol{y}_n)\}_{n=1}^{N_{\text{IN}}}$ to denote the input set and $\mathcal{D}_{\text{OUT}}$ for the corresponding output. We use $\mathscr{H}_k$ to denote a set of nodes after the $k$-th round.

We initialize $\mathscr{H}_0$ by identifying all samples in $\mathcal{D}_{\text{IN}}$ for which unplugging them individually does not affect the LLM's performance. Formally, we construct $\mathscr{H}_0$ as $\mathscr{H}_0 := \{\mathcal{H}_n := \{(\boldsymbol{x}_n, \boldsymbol{y}_n)\}\}$ where $(\boldsymbol{x}_n, \boldsymbol{y}_n) \in \mathcal{D}_{\text{IN}}$ satisfies:

$$Y_{(\{\boldsymbol{x}_{n'} | \boldsymbol{x}_{n'} \in \mathcal{D}_{\text{IN}}\})} = \mathbf{1}_{|\mathcal{D}_{\text{IN}}|} | \text{unplug}((\boldsymbol{x}_n, \boldsymbol{y}_n)); C, S, \quad (24)$$

where $C = \mathcal{D}_{\text{IN}}$ and $S$ is loosened to allow for any value. During each $k$-th round, we generate $\mathscr{H}_k$ by examining the necessity relationship between each pair of nodes (denoted as $\mathcal{H}_i, \mathcal{H}_j \in \mathscr{H}_{k-1}$). Here, we further verify whether solely unplugging $\mathcal{H}_i \cup \mathcal{H}_j$ does not impact the LLM's performance. Formally, we check whether the following equation

holds:

$$Y_{(\{\boldsymbol{x}_{n'} | \boldsymbol{x}_{n'} \in \mathcal{D}_{\text{IN}}\})} = \mathbf{1}_{|\mathcal{D}_{\text{IN}}|} | \text{unplug}(\mathcal{H}_i \cup \mathcal{H}_j); C, S, \quad (25)$$

where $C = \mathcal{D}_{\text{IN}}$ and $S$ is loosened to allow for any value. This determines whether plugging $\mathcal{H}_i \cup \mathcal{H}_j$ is unnecessary for maintaining the correct outputs to all inputs in $\mathcal{D}_{\text{IN}}$. If the above equation holds, we create a new node $\mathcal{H}_i \cup \mathcal{H}_j$ and add it to $\mathscr{H}_k$, labeling it with a MAINTAIN signal. Otherwise, we add both $\mathcal{H}_i$ and $\mathcal{H}_j$ to $\mathscr{H}_k$. After this computation, we identify $\mathcal{H}_{\text{MAX}} = \arg\max_{\mathcal{H}. \in \mathscr{H}_k} |\mathcal{H}.|$ and label it with a MAINTAIN signal. Subsequently, we remove the nodes in $\mathscr{H}_k$ that lack MAINTAIN signals. This process continues until $\mathscr{H}.$ contains only one element, denoted as $\mathcal{H}_{\text{UNNECESSARY}} \in \mathscr{H}_K$. Finally, we calculate $\mathcal{D}_{\text{OUT}}$ as $\mathcal{D}_{\text{OUT}} = \mathcal{D}_{\text{IN}} - \mathcal{H}_{\text{UNNECESSARY}}$.

### A4.2. Exact Extraction of FEEDER via Iterative Sufficiency Checks

Consider that at each round, we need to check the necessity for $O(\texttt{C}_{N_{\text{IN}}}^2)$ times (where $\texttt{C}.$ denotes a combination operator), this becomes impractical. To this end, we develop an alternative algorithm. Specifically, at each $k$-th round, we remove all the checked nodes (i.e., $\mathcal{H}_i$ and $\mathcal{H}_j$ from $\mathscr{H}_k$, similar to our approximation algorithm in Appendix A3). Then, it requires $O(\log_2^{|\mathcal{D}_{\text{IN}}|})$ computations to finish one round. To obtain an exact FEEDER, we need to keep repeating the above process until there is no or only one left in $\mathcal{H}_0$. While practical, we also can set a maximum number of rounds to approximate.

**Proposition 2** ($\widetilde{\mathcal{D}}_{\text{FEEDER}}$ obtained by either Algorithm 3 or Algorithm 4 is an Exact $\mathcal{D}_{\text{FEEDER}}$). *If we successively apply either Algorithm 3 or Algorithm 4 on $\mathcal{D}_{\text{TRAIN}}$ for multiple rounds to obtain a subset (denoted as $\widetilde{\mathcal{D}}_{\text{FEEDER}}$), then $\widetilde{\mathcal{D}}_{\text{FEEDER}}$ is sufficient and necessary to represent $\mathcal{D}_{\text{TRAIN}}$.*

*Proof.* According to Definition A6, it is straightforward to see that to prove the above proposition is equivalent to proving that $\widetilde{\mathcal{D}}_{\text{FEEDER}}$ is a sufficient set of $\mathcal{D}_{\text{TRAIN}}$ and a necessary set of $\mathcal{D}_{\text{TRAIN}}$.

We begin by proving sufficiency. Either Algorithm 3 or 4 preserves the sufficiency during checking the necessity, as we are always guaranteeing $Y_{(\{\boldsymbol{x}_n | \boldsymbol{x}_n \in \mathcal{D}_{\text{TRAIN}}\})} = \mathbf{1}_{|\mathcal{D}_{\text{TRAIN}}|}$, when removing the unnecessary parts.

In other words, we have: $Y_{(\{\boldsymbol{x}_n | \boldsymbol{x}_n \in \mathcal{D}_{\text{TRAIN}}\})} = \mathbf{1}_{|\mathcal{D}_{\text{TRAIN}}|} | \text{unplug}(\mathcal{D}_{\text{TRAIN}} - \mathcal{H}_{\text{UNNECESSARY}}); C, S$, where $C = \mathcal{D}_{\text{TRAIN}}$ and $S$ can be any value. It can be rewritten as:

$$Y_{(\{\boldsymbol{x}_n | \boldsymbol{x}_n \in \mathcal{D}_{\text{TRAIN}}\})} = \mathbf{1}_{|\mathcal{D}_{\text{TRAIN}}|} | \text{plug}(\widetilde{\mathcal{D}}_{\text{FEEDER}}); C, S, \quad (26)$$

where $C = \emptyset$ and $S$ can be any value. It shows that plugging in $\widetilde{\mathcal{D}}_{\text{FEEDER}}$ is sufficient for representing $\mathcal{D}_{\text{TRAIN}}$.

*Table A1.* Performance comparisons on text classification datasets are conducted in the in-context learning setting. We report both the mean and variance of accuracy using 8 different seeds and 5 different permutations of n-shots. This table is extended from Table 1.

| $\Psi_{\text{LLM}}(\cdot)$ | $\mathcal{D}$ | $n$ | SUBJ | | | SST-2 | | | COLA | | |
|---|---|---|---|---|---|---|---|---|---|---|---|
| | | | Random | Similarity | Diversity | Random | Similarity | Diversity | Random | Similarity | Diversity |
| GPT-2 (0.3B) | $\mathcal{D}_{\text{TRAIN}}$ | 1 | 41.3 (7.2) | 41.1 (0.1) | 41.1 (0.1) | 48.9 (4.6) | 24.5 (0.2) | 24.5 (0.2) | 29.0 (5.4) | 38.8 (0.1) | 38.8 (0.1) |
| | | 2 | 47.3 (7.2) | 62.8 (0.1) | 71.9 (0.2) | 51.2 (5.8) | 65.7 (0.1) | 62.5 (0.2) | 30.9 (4.6) | 38.5 (0.2) | 36.2 (0.1) |
| | | 5 | 51.8 (5.5) | 85.8 (0.3) | 70.1 (0.2) | 62.6 (5.6) | 79.4 (0.2) | 61.7 (0.1) | 39.4 (5.8) | 49.3 (0.1) | 47.0 (0.2) |
| | | 10 | 62.4 (5.0) | 88.0 (0.2) | 78.2 (0.1) | 50.9 (4.9) | 83.8 (0.3) | 76.9 (0.2) | 31.6 (4.6) | 52.5 (0.2) | 58.8 (0.2) |
| | $\mathcal{D}_{\text{FEEDER}}$ | 1 | **42.8** (2.4) | **44.9** (1.1) | **44.9** (1.1) | **49.8** (4.2) | **48.1** (1.9) | **48.1** (1.9) | **29.6** (4.1) | 35.1 (1.5) | 35.1 (1.5) |
| | | 2 | **55.9** (3.3) | **63.4** (1.6) | **74.7** (0.9) | **67.3** (4.4) | **67.7** (1.4) | **64.7** (1.5) | **31.3** (2.2) | **41.7** (1.2) | 34.9 (1.9) |
| | | 5 | **57.5** (4.0) | **86.9** (0.7) | 69.8 (1.0) | **70.3** (4.4) | 77.9 (1.2) | **68.5** (1.9) | 35.2 (2.0) | **57.3** (1.2) | **54.6** (1.7) |
| | | 10 | **63.5** (4.4) | **88.7** (1.5) | **79.7** (2.0) | **75.2** (6.2) | 83.0 (1.7) | **77.2** (1.5) | **59.3** (3.8) | **68.7** (2.4) | **68.5** (2.9) |
| MED (0.8B) | $\mathcal{D}_{\text{TRAIN}}$ | 1 | 42.5 (5.2) | 43.6 (0.1) | 43.6 (0.1) | 49.0 (4.3) | 42.3 (0.2) | 42.3 (0.2) | 42.1 (5.7) | 48.3 (0.1) | 48.3 (0.1) |
| | | 2 | 58.1 (6.3) | 88.3 (0.2) | 87.0 (0.3) | 68.0 (5.2) | 70.7 (0.1) | 59.6 (0.2) | 41.1 (4.2) | 36.8 (0.2) | 37.7 (0.1) |
| | | 5 | 66.7 (4.5) | 86.2 (0.2) | 86.7 (0.1) | 49.1 (4.3) | 80.6 (0.1) | 67.5 (0.2) | 46.2 (4.7) | 53.8 (0.2) | 48.5 (0.3) |
| | | 10 | 48.6 (6.0) | 85.9 (0.1) | 73.9 (0.2) | 71.1 (4.5) | 84.6 (0.1) | 73.1 (0.2) | 43.4 (4.5) | 55.5 (0.2) | 56.1 (0.4) |
| | $\mathcal{D}_{\text{FEEDER}}$ | 1 | **45.8** (5.1) | **46.4** (0.4) | **46.4** (0.4) | **49.1** (3.0) | **47.7** (1.3) | **47.7** (1.3) | **46.6** (3.8) | 45.1 (1.1) | 45.1 (1.1) |
| | | 2 | **63.1** (4.5) | **89.7** (1.5) | 86.8 (1.3) | **69.8** (3.8) | **73.0** (2.9) | 61.2 (2.1) | 36.6 (3.5) | **37.0** (2.8) | 34.6 (2.0) |
| | | 5 | **73.4** (4.3) | **88.2** (1.9) | **88.8** (1.7) | 59.3 (2.4) | 80.9 (1.3) | **69.6** (1.7) | **59.2** (3.3) | **68.6** (1.6) | **66.6** (1.7) |
| | | 10 | 52.0 (3.8) | **87.4** (1.3) | 75.6 (1.2) | **76.0** (3.0) | **86.7** (1.4) | **75.6** (1.8) | **59.3** (4.8) | **68.8** (2.0) | **68.9** (1.8) |
| NEO (1.3B) | $\mathcal{D}_{\text{TRAIN}}$ | 1 | 42.8 (3.9) | 42.1 (0.1) | 42.1 (0.1) | 49.2 (3.7) | 33.8 (0.1) | 33.8 (0.1) | 25.5 (3.4) | 36.5 (0.2) | 36.5 (0.2) |
| | | 2 | 48.5 (4.2) | 88.3 (0.2) | 72.6 (0.3) | 76.8 (3.5) | 81.5 (0.1) | 76.3 (0.4) | 30.7 (3.1) | 55.5 (0.2) | 56.5 (0.4) |
| | | 5 | 51.6 (5.0) | 90.5 (0.2) | 81.7 (0.2) | 65.1 (3.5) | 80.8 (0.2) | 66.1 (0.3) | 40.0 (3.6) | 55.9 (0.1) | 52.5 (0.2) |
| | | 10 | 48.5 (5.8) | 85.9 (0.3) | 81.9 (0.1) | 69.8 (4.8) | 84.1 (0.1) | 69.7 (0.1) | 39.6 (4.5) | 59.3 (0.3) | 63.4 (0.1) |
| | $\mathcal{D}_{\text{FEEDER}}$ | 1 | **43.2** (4.0) | **46.3** (1.0) | **46.3** (1.0) | **49.3** (5.1) | **48.3** (1.9) | **48.3** (1.9) | **28.3** (5.4) | 34.8 (1.3) | 34.8 (1.3) |
| | | 2 | **62.6** (3.5) | **89.4** (1.5) | **73.8** (2.1) | 75.1 (2.8) | **82.6** (2.1) | **78.5** (1.9) | **59.3** (3.7) | **64.7** (1.4) | **64.7** (1.6) |
| | | 5 | **69.4** (5.6) | **91.2** (1.8) | **82.9** (1.3) | **73.2** (4.2) | **82.9** (2.7) | **71.6** (2.4) | **58.7** (3.2) | **67.2** (2.4) | **65.8** (1.8) |
| | | 10 | **58.7** (3.3) | **87.2** (1.7) | **84.3** (2.8) | **72.4** (3.4) | **85.8** (2.5) | **71.8** (2.9) | **59.8** (2.8) | **68.8** (1.4) | **68.9** (1.3) |
| Gemma-2 (2B) | $\mathcal{D}_{\text{TRAIN}}$ | 1 | 45.0 (5.9) | 48.1 (0.6) | 48.1 (0.6) | 51.2 (6.8) | 52.2 (0.8) | 52.2 (0.8) | 37.5 (7.0) | 40.5 (1.3) | 40.5 (1.3) |
| | | 2 | 62.3 (6.9) | 82.5 (1.8) | 74.2 (1.3) | 71.5 (5.6) | 78.5 (1.5) | 75.9 (0.9) | 40.6 (5.9) | 62.5 (1.0) | 61.6 (0.5) |
| | | 5 | 68.0 (7.1) | 91.5 (1.2) | 84.2 (1.6) | 70.2 (5.6) | 80.5 (1.6) | 80.6 (0.7) | 46.5 (5.9) | 67.2 (1.8) | 65.6 (0.6) |
| | | 10 | 50.3 (8.2) | 86.2 (1.9) | 85.6 (0.8) | 68.2 (4.8) | 85.5 (1.5) | 76.3 (1.3) | 50.2 (7.4) | 69.8 (1.5) | 71.5 (1.2) |
| | $\mathcal{D}_{\text{FEEDER}}$ | 1 | **48.2** (4.2) | **49.5** (1.0) | **49.5** (1.0) | **52.6** (4.6) | **53.1** (0.8) | **53.1** (0.8) | **38.9** (5.2) | 39.6 (0.8) | 39.6 (0.8) |
| | | 2 | **65.2** (2.9) | **85.2** (1.0) | **80.3** (0.8) | **74.2** (4.9) | **82.1** (1.2) | **83.0** (0.7) | **52.5** (2.5) | **68.9** (2.1) | **67.8** (1.5) |
| | | 5 | **72.2** (6.2) | **94.5** (5.3) | **85.5** (0.7) | 72.0 (4.2) | **83.6** (2.1) | **84.5** (1.7) | **55.2** (4.8) | **77.6** (2.5) | **73.9** (2.3) |
| | | 10 | **60.5** (4.0) | **86.5** (2.5) | **88.4** (2.4) | **70.5** (5.6) | **92.6** (2.6) | **78.5** (5.3) | **58.6** (4.6) | **75.6** (2.9) | **76.6** (2.5) |
| GPT-3 (6B) | $\mathcal{D}_{\text{TRAIN}}$ | 1 | 44.9 (6.6) | 49.5 (0.1) | 49.5 (0.1) | 48.2 (2.9) | 47.0 (0.1) | 47.0 (0.1) | 38.9 (6.7) | 41.2 (0.2) | 41.2 (0.2) |
| | | 2 | 55.4 (3.5) | 85.5 (0.1) | 86.5 (0.2) | 68.1 (4.2) | 78.7 (0.2) | 77.5 (0.1) | 42.8 (4.0) | 45.5 (0.3) | 45.6 (0.2) |
| | | 5 | 51.2 (4.4) | 90.8 (0.2) | 82.7 (0.1) | 75.2 (3.3) | 80.7 (0.1) | 77.8 (0.2) | 48.5 (3.3) | 51.8 (0.3) | 52.1 (0.2) |
| | | 10 | 57.7 (4.8) | 87.3 (0.1) | 85.3 (0.1) | 72.1 (3.8) | 77.6 (0.1) | 76.5 (0.2) | 59.1 (4.2) | 60.3 (0.1) | 61.0 (0.2) |
| | $\mathcal{D}_{\text{FEEDER}}$ | 1 | **43.9** (4.2) | **51.2** (1.0) | **51.2** (1.0) | **49.6** (2.4) | **51.3** (1.6) | **51.3** (1.6) | **41.2** (2.1) | **43.8** (1.8) | **43.8** (1.8) |
| | | 2 | **65.7** (3.0) | **91.5** (1.1) | **88.8** (1.6) | **73.5** (2.5) | **85.7** (4.2) | 76.1 (2.1) | **61.8** (2.1) | **63.1** (1.5) | **60.1** (1.4) |
| | | 5 | 53.7 (3.8) | **92.9** (0.8) | **91.5** (1.4) | **77.6** (4.0) | **81.0** (1.3) | **79.4** (1.0) | **50.6** (2.7) | **63.3** (1.4) | **65.8** (1.4) |
| | | 10 | **58.0** (3.4) | **88.8** (0.9) | **87.8** (1.2) | **83.8** (2.8) | **86.4** (2.0) | **87.2** (1.3) | **59.7** (3.0) | **67.5** (1.9) | **68.4** (2.2) |
| Llama-2 (7B) | $\mathcal{D}_{\text{TRAIN}}$ | 1 | 42.9 (6.6) | 48.5 (0.1) | 48.5 (0.1) | 46.2 (2.7) | 49.1 (0.1) | 49.1 (0.1) | 40.1 (6.1) | 42.0 (0.2) | 42.0 (0.2) |
| | | 2 | 51.9 (4.4) | 90.7 (0.1) | 85.2 (0.2) | 67.8 (3.2) | 73.5 (0.2) | 74.5 (0.2) | 43.5 (4.5) | 47.4 (0.2) | 49.6 (0.1) |
| | | 5 | 51.6 (3.2) | 86.8 (0.2) | 82.9 (0.1) | 74.8 (3.8) | 81.2 (0.2) | 78.7 (0.2) | 50.2 (3.7) | 52.6 (0.2) | 48.2 (0.3) |
| | | 10 | 56.1 (4.6) | 81.3 (0.1) | 85.7 (0.1) | 73.2 (3.1) | 76.3 (0.1) | 77.1 (0.1) | 59.6 (4.3) | 55.3 (0.2) | 60.0 (0.4) |
| | $\mathcal{D}_{\text{FEEDER}}$ | 1 | **43.8** (4.3) | **49.7** (1.0) | **49.7** (1.0) | **47.2** (2.4) | **50.8** (1.7) | **50.8** (1.7) | **41.2** (2.1) | **43.8** (1.8) | **43.8** (1.8) |
| | | 2 | **54.8** (3.0) | **92.5** (1.1) | 84.8 (0.7) | **72.2** (3.1) | **82.5** (4.0) | **80.1** (2.6) | **50.8** (2.3) | **58.6** (1.7) | **53.5** (1.3) |
| | | 5 | 53.7 (3.8) | **87.9** (1.8) | **91.5** (1.4) | 78.3 (4.6) | **83.2** (1.1) | **80.1** (1.4) | **53.8** (2.8) | **65.3** (1.6) | **61.8** (1.4) |
| | | 10 | **58.0** (3.4) | **85.8** (0.9) | **87.8** (1.2) | **85.0** (2.2) | **87.1** (2.2) | **86.9** (1.0) | **60.5** (3.1) | **68.0** (1.7) | **68.4** (2.0) |

*Table A2.* A complementary table to Table A1 presents the corresponding results for the demonstration selectors Uncertainty, Clustering, Latent.

| $\Psi_{\text{LLM}}(\cdot)$ | $\mathcal{D}$ | $n$ | SUBJ | | | SST-2 | | | COLA | | |
|---|---|---|---|---|---|---|---|---|---|---|---|
| | | | Uncertainty | Clustering | Latent | Uncertainty | Clustering | Latent | Uncertainty | Clustering | Latent |
| GPT-3 (6B) | $\mathcal{D}_{\text{TRAIN}}$ | 1 | 53.5 (6.3) | 49.3 (4.4) | 51.5 (2.1) | 49.0 (2.9) | 47.5 (1.5) | 47.8 (1.1) | 42.0 (6.5) | 39.8 (1.5) | 40.2 (1.2) |
| | | 2 | 87.8 (3.7) | 86.5 (4.1) | 86.3 (3.5) | 75.6 (4.2) | 80.1 (2.2) | 79.0 (2.4) | 49.6 (4.0) | 46.8 (5.0) | 47.5 (3.3) |
| | | 5 | 90.7 (4.5) | 88.2 (4.4) | 89.4 (4.2) | 81.8 (3.3) | 82.2 (3.3) | 80.7 (4.4) | 55.4 (3.5) | 56.4 (4.3) | 58.8 (3.3) |
| | | 10 | 88.3 (4.8) | 90.7 (3.8) | 91.3 (4.1) | 80.5 (3.8) | 78.8 (3.9) | 76.8 (4.1) | 58.4 (4.2) | 62.1 (3.6) | 61.5 (4.5) |
| | $\mathcal{D}_{\text{FEEDER}}$ | 1 | **55.3** (4.2) | **50.9** (4.4) | 50.2 (3.2) | **50.3** (2.4) | **48.4** (3.4) | **48.3** (2.6) | **43.8** (2.1) | **40.8** (3.5) | **42.5** (5.1) |
| | | 2 | **89.8** (3.0) | **89.7** (3.5) | **89.5** (2.5) | **77.1** (2.5) | **82.5** (3.5) | 83.0 (3.2) | **60.0** (2.1) | **57.8** (4.4) | **58.1** (3.5) |
| | | 5 | **92.3** (3.8) | **92.0** (2.4) | **91.8** (2.9) | **81.2** (4.0) | 80.8 (3.8) | 80.4 (2.9) | **62.4** (2.7) | **61.6** (3.7) | **62.3** (2.4) |
| | | 10 | **90.8** (3.4) | **92.0** (2.4) | **91.8** (2.9) | **81.2** (2.8) | 80.8 (3.8) | 80.4 (2.9) | **62.4** (3.0) | **62.7** (3.1) | **62.5** (2.5) |
| Llama-2 (7B) | $\mathcal{D}_{\text{TRAIN}}$ | 1 | 49.0 (6.6) | 48.5 (5.6) | 47.5 (5.1) | 49.2 (2.7) | 48.2 (3.7) | 48.7 (3.1) | 40.1 (6.1) | 41.1 (4.1) | 41.0 (3.2) |
| | | 2 | 89.2 (4.4) | 87.8 (3.5) | 88.7 (4.1) | 75.1 (3.2) | 72.5 (2.2) | 74.7 (4.2) | 48.5 (4.5) | 45.2 (4.0) | 46.4 (1.2) |
| | | 5 | 82.9 (3.2) | 80.1 (2.2) | 83.8 (1.2) | 83.7 (3.8) | 81.5 (3.0) | 82.2 (1.2) | 53.2 (3.7) | 51.2 (2.5) | 52.6 (2.2) |
| | | 10 | 86.2 (4.6) | 82.1 (4.4) | 83.3 (2.1) | 76.4 (3.1) | 75.2 (3.7) | 74.8 (4.1) | 63.5 (4.3) | 62.6 (4.0) | 60.3 (2.2) |
| | $\mathcal{D}_{\text{FEEDER}}$ | 1 | **49.7** (4.3) | 45.8 (4.3) | **48.7** (5.1) | **51.8** (2.4) | **48.4** (3.5) | **50.3** (2.7) | **43.0** (2.1) | **42.2** (2.5) | **42.8** (1.8) |
| | | 2 | **91.8** (3.0) | **90.8** (3.4) | **91.5** (2.4) | **78.1** (3.1) | **73.5** (3.1) | **76.5** (4.0) | **49.5** (2.3) | **48.8** (2.3) | **50.6** (2.7) |
| | | 5 | **89.5** (3.8) | **88.7** (4.8) | **86.9** (2.8) | **84.1** (4.6) | **82.3** (4.5) | **83.8** (4.1) | **60.8** (2.8) | **58.8** (3.8) | **59.3** (2.6) |
| | | 10 | **88.8** (3.4) | **88.0** (4.4) | **86.8** (2.9) | **80.9** (2.2) | **85.1** (2.0) | **83.4** (2.2) | **67.4** (3.1) | **64.5** (3.4) | **66.0** (2.7) |

Next, we investigate necessity. Our goal is to prove unplugging any data point in $\widetilde{\mathcal{D}}_{\text{FEEDER}}$ would lead to a degradation of the LLM's performance. For convenience, we use $(\boldsymbol{x}_n, \boldsymbol{y}_n) \in \mathcal{D}_{\text{TRAIN}}$ to denote an arbitrary data point. If we applying Algorithm 3 to execute the search for an exact $\mathcal{D}_{\text{FEEDER}}$, then $(\boldsymbol{x}_n, \boldsymbol{y}_n)$ must be in $\mathscr{H}_0$, or out of $\mathscr{H}_0$.

If $(\boldsymbol{x}_n, \boldsymbol{y}_n)$ is not an element in $\mathscr{H}_0$, then according to the computing process of $\mathscr{H}_0$ (i.e., lines 3 to 3 in Algorithm 3), unplugging $(\boldsymbol{x}_n, \boldsymbol{y}_n)$ it would definitely cause the LLM's performance on $\mathcal{D}_{\text{TRAIN}}$ from $Y_{(\{\boldsymbol{x}_n|\boldsymbol{x}_n \in \mathcal{D}_{\text{TRAIN}}\})} = \mathbf{1}_{|\mathcal{D}_{\text{TRAIN}}|}$ to $Y_{(\{\boldsymbol{x}_n|\boldsymbol{x}_n \in \mathcal{D}_{\text{TRAIN}}\})} \neq \mathbf{1}_{|\mathcal{D}_{\text{TRAIN}}|}$.

If $(\boldsymbol{x}_n, \boldsymbol{y}_n)$ is an element in $\mathscr{H}_0$, then $(\boldsymbol{x}_n, \boldsymbol{y}_n)$ must be in $\mathscr{H}_{\text{UNNECESSARY}}$; otherwise, according to lines 3 to 3 in Algorithm 3, $\mathscr{H}_{\text{UNNECESSARY}} \cup \{(\boldsymbol{x}_n, \boldsymbol{y}_n)\}$ should be $\mathscr{H}_{\text{MAX}}$ and always stay in $\mathscr{H}$. until becoming the root node (i.e., $\mathscr{H}_{\text{UNNECESSARY}}$ should be updated to be $\mathscr{H}_{\text{UNNECESSARY}} \cup \{(\boldsymbol{x}_n, \boldsymbol{y}_n)\}$). Thus, $(\boldsymbol{x}_n, \boldsymbol{y}_n)$ must be in $\mathscr{H}_{\text{UNNECESSARY}}$. However, all the data points in $\mathscr{H}_{\text{UNNECESSARY}}$ are removed from $\mathcal{D}_{\text{TRAIN}}$, causing a contradiction. Hence, unplugging $(\boldsymbol{x}_n, \boldsymbol{y}_n)$ would change the LLM's performance, namely necessity holds.

Then, we consider applying Algorithm 4 for searching an exact $\mathcal{D}_{\text{FEEDER}}$. Similarly, if $(\boldsymbol{x}_n, \boldsymbol{y}_n)$ is not selected when checking the necessity, then unplugging $(\boldsymbol{x}_n, \boldsymbol{y}_n)$ would definitely cause a degradation of the LLM's performance.

If $(\boldsymbol{x}_n, \boldsymbol{y}_n)$ is selected during checking the necessity, then $(\boldsymbol{x}_n, \boldsymbol{y}_n)$ must be included in $\mathcal{D}_r$; otherwise, $\mathcal{D}_r$ would continue to update, since the condition of stopping iteration is that there is no or only one unnecessary node. However, all the data points are removed from $\mathcal{D}_{\text{TRAIN}}$, causing a contradiction. Hence, unplugging $(\boldsymbol{x}_n, \boldsymbol{y}_n)$ would change the LLM's performance, namely necessity holds.

Combining the above analysis of sufficiency and necessity, we can conclude that $\mathcal{D}_{\text{FEEDER}}$ is an exact FEEDER for $\mathcal{D}_{\text{TRAIN}}$. □

## A5. FEEDER **in In-context Learning Setting**

### A5.1. Demonstration Selectors

As described in Section 5.1, when applied in the ICL setting, our $\mathcal{D}_{\text{FEEDER}}$ is assessed by serving as the selection pool, replacing $\mathcal{D}_{\text{TRAIN}}$ for existing demonstration selectors.

The first one is a random selector, denoted as Random, which randomly selects samples from the selection pool.

The second one is a similarity-based selector, denoted as Similarity, which selects samples similar to the test samples. Formally, let $\mathcal{D}_{\text{SELECT}}$ denote the selection pool. Then, for each test sample $\boldsymbol{x}_m$, the metric of similarity can be written as:

$$\text{SIM}(\boldsymbol{x}_m, \boldsymbol{x}_n) = \text{COS}(\text{TRANSFORMER}(\boldsymbol{x}_m), \text{TRANSFORMER}(\boldsymbol{x}_n)), \quad (27)$$

where $\boldsymbol{x}_n \in \mathcal{D}_{\text{SELECT}}$, $\text{COS}(\cdot)$ is a cosine similarity metric, and $\text{TRANSFORMER}(\cdot)$ denotes a sentence transformer (Reimers & Gurevych, 2019). Here, we directly use the Sentence Transformers library[2] from Hugging Face in our implementation. Then, we are able to select $N_{\text{shot}}$ samples with maximum SIM values from $\mathcal{D}_{\text{SELECT}}$.

The third one is a diversity-based selector, denoted as Diversity, where we adopt the maximal marginal relevance method (Carbonell & Goldstein, 1998) as the metric of Di-

[2] https://huggingface.co/sentence-transformers

*Table A3.* Performance comparisons on text classification datasets are conducted in the in-context learning setting. We report both the mean and variance of accuracy using 8 different seeds and 5 different permutations of n-shots. This table is extended from Table 1.

| $\Psi_{\text{LLM}}(\cdot)$ | $\mathcal{D}$ | $n$ | FPB | | | SST-5 | | | TREC | | |
|---|---|---|---|---|---|---|---|---|---|---|---|
| | | | Random | Similarity | Diversity | Random | Similarity | Diversity | Random | Similarity | Diversity |
| GPT-2 (0.3B) | $\mathcal{D}_{\text{TRAIN}}$ | 1 | 27.2 (6.1) | 25.3 (0.1) | 25.3 (0.1) | 14.5 (6.1) | 22.7 (0.2) | 22.7 (0.2) | 19.4 (6.4) | 42.8 (0.1) | 42.8 (0.1) |
| | | 2 | 27.4 (6.2) | 45.8 (0.2) | 40.4 (0.1) | 18.0 (5.8) | 25.6 (0.1) | 23.7 (0.2) | 21.4 (4.7) | 57.2 (0.2) | 51.4 (0.1) |
| | | 5 | 26.3 (4.5) | 55.9 (0.1) | 44.7 (0.2) | 26.5 (5.3) | 32.3 (0.2) | 27.8 (0.1) | 37.6 (5.1) | 66.0 (0.3) | 61.4 (0.3) |
| | | 10 | 27.8 (5.1) | 63.1 (0.1) | 50.7 (0.1) | 14.9 (3.9) | 35.3 (0.1) | 30.4 (0.2) | 53.0 (5.2) | 71.4 (0.2) | 65.8 (0.3) |
| | $\mathcal{D}_{\text{FEEDER}}$ | 1 | **28.4** (3.4) | **28.8** (2.1) | **28.8** (2.1) | **15.4** (5.2) | **23.7** (1.7) | **23.7** (1.7) | **37.4** (3.6) | **48.4** (1.6) | **48.4** (1.6) |
| | | 2 | **35.5** (4.3) | **47.4** (2.6) | 37.9 (1.9) | **20.9** (4.7) | **27.9** (1.1) | **25.8** (1.3) | 27.6 (3.2) | **58.8** (2.2) | **52.1** (1.9) |
| | | 5 | **28.3** (3.0) | 54.6 (1.7) | **47.9** (1.0) | **28.6** (3.4) | **33.2** (1.8) | 27.4 (1.7) | **40.8** (3.0) | **67.4** (1.2) | **61.8** (1.3) |
| | | 10 | **39.6** (3.4) | **66.5** (2.3) | **51.8** (1.2) | **17.6** (2.2) | **36.9** (1.9) | 29.8 (1.7) | 44.6 (2.8) | **74.6** (1.4) | **67.6** (1.9) |
| GPT-2 (0.8B) | $\mathcal{D}_{\text{TRAIN}}$ | 1 | 33.8 (5.2) | 29.9 (0.1) | 29.9 (0.1) | 14.2 (4.9) | 25.2 (0.1) | 25.2 (0.1) | 21.0 (4.6) | 53.2 (0.2) | 53.2 (0.2) |
| | | 2 | 27.0 (6.1) | 55.4 (0.2) | 49.9 (0.3) | 18.1 (5.1) | 29.7 (0.1) | 24.4 (0.2) | 28.2 (4.4) | 62.6 (0.2) | 60.6 (0.2) |
| | | 5 | 27.2 (4.8) | 64.3 (0.1) | 45.1 (0.3) | 25.6 (4.8) | 34.1 (0.1) | 30.8 (0.1) | 35.4 (5.7) | 63.4 (0.1) | 64.6 (0.1) |
| | | 10 | 47.0 (5.5) | 65.5 (0.2) | 52.9 (0.1) | 28.7 (4.2) | 38.7 (0.1) | 36.6 (0.1) | 43.2 (4.8) | 66.0 (0.1) | 68.8 (0.1) |
| | $\mathcal{D}_{\text{FEEDER}}$ | 1 | 33.8 (4.4) | **32.6** (0.7) | **32.6** (0.7) | **18.7** (3.0) | **25.5** (2.2) | **25.5** (2.2) | **22.4** (3.8) | 52.6 (2.1) | 52.6 (2.1) |
| | | 2 | **37.5** (4.7) | 54.8 (1.1) | 47.6 (1.3) | **25.2** (3.8) | 29.7 (1.9) | 24.1 (2.1) | **34.6** (3.5) | **64.2** (1.8) | 59.4 (2.0) |
| | | 5 | **38.9** (3.3) | **64.5** (1.3) | **48.0** (2.7) | **39.3** (2.9) | **35.2** (1.1) | **31.0** (1.2) | **45.4** (3.3) | **65.5** (1.5) | **64.9** (1.7) |
| | | 10 | **63.5** (2.8) | **66.7** (1.6) | **53.1** (1.5) | **39.6** (3.0) | **39.8** (1.8) | **37.8** (1.6) | **55.8** (3.8) | **70.4** (2.0) | 68.6 (1.7) |
| GPT-neo (1.3B) | $\mathcal{D}_{\text{TRAIN}}$ | 1 | 54.9 (3.9) | 61.6 (0.1) | 61.6 (0.1) | 12.8 (2.7) | 20.2 (0.1) | 20.2 (0.1) | 11.0 (3.2) | 57.2 (0.2) | 57.2 (0.2) |
| | | 2 | 53.6 (4.0) | 66.8 (0.2) | 60.0 (0.1) | 17.9 (3.6) | 26.9 (0.1) | 22.7 (0.1) | 17.6 (3.1) | 52.6 (0.2) | 42.2 (0.2) |
| | | 5 | 28.2 (4.0) | 68.2 (0.1) | 60.4 (0.1) | 19.0 (3.9) | 29.2 (0.1) | 25.1 (0.1) | 25.2 (3.8) | 66.4 (0.1) | 61.8 (0.1) |
| | | 10 | 49.0 (4.8) | 75.8 (0.1) | 71.1 (0.2) | 12.7 (2.8) | 33.7 (0.2) | 31.9 (0.1) | 41.6 (4.4) | 70.6 (0.1) | 69.0 (0.1) |
| | $\mathcal{D}_{\text{FEEDER}}$ | 1 | **58.1** (4.7) | **61.8** (1.4) | **61.8** (1.4) | **18.5** (2.1) | **20.6** (1.8) | **20.6** (1.4) | **18.2** (2.4) | 56.4 (1.3) | 56.4 (1.3) |
| | | 2 | **61.4** (3.3) | 64.1 (1.5) | 58.8 (1.1) | **19.7** (2.7) | **27.4** (2.1) | **22.8** (1.8) | **27.8** (2.7) | **54.0** (1.4) | **44.5** (1.6) |
| | | 5 | **43.2** (2.6) | **68.8** (1.8) | **62.7** (1.3) | **19.2** (3.2) | **30.2** (2.7) | **26.4** (2.4) | **50.4** (3.2) | **68.0** (1.4) | **62.6** (1.9) |
| | | 10 | **61.4** (2.3) | 74.8 (1.9) | **71.9** (1.8) | **15.4** (2.4) | **37.0** (1.5) | **34.5** (1.9) | **45.2** (2.9) | **72.8** (1.4) | **69.8** (1.5) |
| Gemma-2 (2B) | $\mathcal{D}_{\text{TRAIN}}$ | 1 | 58.2 (5.7) | 62.5 (0.1) | 62.5 (0.1) | 21.5 (3.9) | 22.5 (0.1) | 22.5 (0.1) | 21.9 (3.4) | 52.3 (0.1) | 52.3 (0.1) |
| | | 2 | 59.2 (5.9) | 66.2 (0.4) | 65.8 (0.3) | 26.5 (3.6) | 42.5 (0.6) | 42.2 (0.6) | 35.6 (4.4) | 60.0 (0.2) | 59.1 (0.1) |
| | | 5 | 48.6 (3.6) | 76.6 (0.4) | 78.8 (0.6) | 26.6 (2.5) | 48.8 (0.3) | 41.2 (0.4) | 55.8 (2.9) | 82.2 (0.2) | 71.1 (0.6) |
| | | 10 | 35.2 (6.5) | 79.5 (0.4) | 78.8 (0.2) | 36.6 (4.4) | 50.2 (0.8) | 43.3 (0.4) | 51.1 (3.3) | 84.3 (0.5) | 75.0 (0.4) |
| | $\mathcal{D}_{\text{FEEDER}}$ | 1 | **59.9** (4.4) | **64.6** (0.6) | **64.6** (0.6) | **22.6** (4.3) | **25.8** (1.3) | **25.8** (1.3) | **26.2** (1.8) | **55.1** (1.8) | **55.1** (1.8) |
| | | 2 | 55.4 (2.4) | **67.8** (1.8) | **67.0** (1.1) | **28.7** (2.3) | **45.4** (1.0) | **46.8** (1.1) | **40.8** (1.5) | **63.6** (1.3) | **62.8** (1.6) |
| | | 5 | **52.2** (3.4) | **88.0** (4.6) | **80.1** (3.2) | **30.5** (2.0) | **52.6** (1.9) | **54.4** (1.4) | **60.4** (2.5) | **87.8** (1.6) | **73.0** (1.2) |
| | | 10 | **39.1** (5.1) | **81.3** (3.3) | **83.8** (2.4) | **36.8** (2.2) | **62.5** (1.5) | **54.9** (1.3) | **58.1** (5.2) | **88.9** (1.8) | **83.4** (1.4) |
| GPT-3 (6B) | $\mathcal{D}_{\text{TRAIN}}$ | 1 | 30.7 (5.5) | 55.3 (0.1) | 55.3 (0.1) | 19.6 (3.6) | 20.5 (0.1) | 20.5 (0.1) | 21.4 (4.4) | 50.7 (0.1) | 50.7 (0.1) |
| | | 2 | 33.4 (4.9) | 64.9 (0.4) | 65.5 (0.3) | 24.1 (3.0) | 30.5 (0.4) | 31.6 (0.3) | 34.4 (4.0) | 58.8 (0.2) | 60.7 (0.1) |
| | | 5 | 40.6 (3.0) | 75.0 (0.4) | 74.9 (0.1) | 24.1 (2.5) | 32.5 (0.3) | 35.6 (0.2) | 51.8 (2.9) | 71.2 (0.2) | 70.6 (0.4) |
| | | 10 | 25.9 (6.5) | 78.5 (0.4) | 79.5 (0.2) | 35.5 (4.2) | 38.9 (0.1) | 40.5 (0.3) | 49.5 (3.6) | 72.5 (0.1) | 73.0 (0.2) |
| | $\mathcal{D}_{\text{FEEDER}}$ | 1 | **31.2** (4.8) | 54.8 (0.8) | 54.8 (0.8) | **20.6** (3.1) | **27.8** (1.3) | **27.8** (1.3) | **32.2** (1.8) | **52.1** (1.8) | **52.1** (1.8) |
| | | 2 | **35.4** (2.4) | **65.8** (1.8) | **67.1** (0.9) | **28.7** (2.3) | **33.4** (1.4) | **33.0** (1.1) | **44.8** (2.5) | **60.1** (1.5) | **61.8** (1.4) |
| | | 5 | **42.2** (3.4) | **77.9** (3.6) | **78.4** (3.2) | **28.5** (2.0) | **35.6** (1.3) | **37.4** (1.4) | **53.4** (2.7) | **75.8** (1.6) | **72.2** (1.2) |
| | | 10 | **39.1** (5.1) | **80.3** (3.3) | **82.8** (2.4) | **36.8** (2.2) | **41.5** (1.5) | **40.9** (1.3) | **54.1** (5.2) | **76.9** (1.8) | **80.4** (1.4) |
| Llama-2 (7B) | $\mathcal{D}_{\text{TRAIN}}$ | 1 | 29.0 (4.7) | 47.1 (0.1) | 47.1 (0.1) | 28.6 (2.9) | 29.7 (0.1) | 29.7 (0.1) | 35.2 (3.7) | 54.2 (0.1) | 54.2 (0.1) |
| | | 2 | 27.4 (3.4) | 68.4 (0.2) | 67.1 (0.3) | 35.9 (3.1) | 33.9 (0.1) | 33.5 (0.3) | 45.0 (4.0) | 69.4 (0.1) | 63.6 (0.1) |
| | | 5 | 39.7 (3.2) | 80.3 (0.2) | 78.9 (0.1) | 37.9 (2.3) | 38.3 (0.2) | 37.0 (0.1) | 53.0 (3.6) | 79.0 (0.2) | 70.4 (0.3) |
| | | 10 | 37.9 (2.6) | 87.4 (0.3) | 86.5 (0.2) | 38.4 (3.8) | 37.5 (0.1) | 40.0 (0.2) | 58.0 (2.3) | 83.4 (0.1) | 79.2 (0.1) |
| | $\mathcal{D}_{\text{FEEDER}}$ | 1 | **33.7** (5.3) | **51.7** (0.8) | **51.7** (0.8) | 27.6 (2.4) | **32.3** (1.5) | **32.3** (1.3) | **41.2** (2.1) | **56.8** (1.8) | **56.8** (1.8) |
| | | 2 | **39.6** (5.0) | **68.7** (1.5) | **69.8** (0.7) | **39.5** (2.5) | 32.6 (1.2) | 32.7 (1.1) | **53.8** (2.3) | 68.6 (1.7) | 63.5 (1.3) |
| | | 5 | **45.6** (4.8) | **87.9** (4.8) | **79.5** (3.5) | **39.2** (2.0) | **38.7** (1.3) | **39.4** (1.0) | **58.2** (2.8) | **82.8** (1.6) | **71.8** (1.4) |
| | | 10 | 37.8 (6.4) | 87.1 (3.9) | **87.8** (2.2) | **39.7** (2.8) | **39.0** (1.0) | **41.6** (1.3) | **59.8** (3.1) | **86.0** (1.9) | **83.4** (2.0) |

*Table A4.* A complementary table to Table A3 presents the corresponding results for the demonstration selectors Uncertainty, Clustering, Latent.

| $\Psi_{\text{LLM}}(\cdot)$ | $\mathcal{D}$ | $n$ | FPB | | | SST-5 | | | TREC | | |
|---|---|---|---|---|---|---|---|---|---|---|---|
| | | | Uncertainty | Clustering | Latent | Uncertainty | Clustering | Latent | Uncertainty | Clustering | Latent |
| GPT-3 (6B) | $\mathcal{D}_{\text{TRAIN}}$ | 1 | 55.8 (6.3) | 56.3 (4.0) | 58.0 (2.5) | 29.0 (2.9) | 27.5 (1.5) | 25.8 (1.1) | 52.0 (6.5) | 49.8 (1.5) | 50.2 (1.2) |
| | | 2 | 67.8 (3.7) | 66.5 (4.1) | 66.3 (3.5) | 35.6 (4.2) | 36.1 (2.2) | 34.0 (2.4) | 59.6 (4.0) | 60.8 (5.0) | 58.5 (3.3) |
| | | 5 | 76.7 (4.5) | 78.2 (4.4) | 79.4 (4.2) | 41.8 (3.3) | 42.2 (3.3) | 40.7 (4.4) | 65.4 (3.5) | 66.4 (4.3) | 65.8 (3.3) |
| | | 10 | 78.3 (4.8) | 80.7 (3.8) | 81.3 (4.1) | 40.5 (3.8) | 38.8 (3.9) | 36.8 (4.1) | 78.4 (4.2) | 72.1 (3.6) | 71.5 (4.5) |
| | $\mathcal{D}_{\text{FEEDER}}$ | 1 | **56.3** (4.2) | **57.9** (4.4) | **58.2** (3.2) | **32.3** (2.4) | **29.4** (3.4) | **28.3** (2.6) | **53.8** (2.1) | **50.8** (3.5) | **52.5** (5.1) |
| | | 2 | **69.8** (3.0) | **69.7** (3.5) | **69.5** (2.5) | **37.1** (2.5) | **42.5** (3.5) | **38.2** (3.2) | **60.1** (2.1) | 57.8 (4.8) | **59.1** (3.5) |
| | | 5 | **82.3** (3.8) | **82.0** (2.4) | **81.8** (2.9) | **44.2** (4.0) | **45.8** (3.8) | **44.4** (2.9) | **68.4** (2.7) | **66.6** (3.7) | **67.3** (2.4) |
| | | 10 | **80.8** (3.4) | **83.0** (2.4) | **83.8** (2.9) | **42.2** (2.8) | **40.8** (3.8) | **40.4** (2.9) | **82.4** (3.0) | **74.7** (3.1) | **73.5** (2.5) |
| Llama-2 (7B) | $\mathcal{D}_{\text{TRAIN}}$ | 1 | 49.0 (6.6) | 47.5 (5.6) | 47.5 (5.1) | 36.2 (2.4) | 37.2 (3.7) | 38.7 (4.1) | 55.1 (6.1) | 54.1 (4.0) | 54.0 (3.3) |
| | | 2 | 68.2 (4.8) | 67.8 (3.5) | 68.7 (4.1) | 35.1 (4.2) | 32.5 (2.0) | 34.7 (4.2) | 67.5 (4.5) | 68.2 (4.0) | 66.4 (1.3) |
| | | 5 | 80.9 (3.2) | 81.6 (2.2) | 83.8 (1.2) | 36.7 (3.8) | 38.5 (3.0) | 39.2 (1.2) | 68.2 (3.7) | 69.2 (2.5) | 67.3 (2.2) |
| | | 10 | 86.2 (4.6) | 85.1 (4.4) | 87.3 (2.1) | 36.4 (3.1) | 35.2 (3.7) | 39.8 (4.1) | 86.5 (4.3) | 85.6 (4.0) | 87.3 (2.2) |
| | $\mathcal{D}_{\text{FEEDER}}$ | 1 | **51.2** (4.8) | **48.9** (4.3) | **48.7** (5.1) | **41.8** (2.4) | **44.4** (3.5) | **43.3** (2.7) | **58.0** (2.1) | **62.2** (2.5) | **62.8** (1.8) |
| | | 2 | **71.8** (3.0) | **72.8** (3.4) | **73.5** (2.4) | **45.1** (3.1) | **45.3** (3.1) | **46.5** (4.0) | **69.5** (2.3) | **70.8** (2.3) | **70.6** (2.7) |
| | | 5 | **88.5** (3.8) | **85.7** (4.8) | **86.9** (2.8) | **42.1** (4.6) | **42.3** (4.5) | **40.8** (4.1) | **72.8** (2.8) | **75.8** (3.8) | **69.3** (2.6) |
| | | 10 | **88.8** (3.4) | **91.1** (4.4) | **89.8** (2.9) | **46.9** (2.2) | **50.1** (2.0) | **53.0** (2.2) | **87.4** (3.1) | **88.5** (3.4) | **89.0** (2.7) |

*Table A5.* Performance comparisons on text classification datasets are conducted in the fine-tuning setting, where we tune the LLMs and evaluate their few-shot inference performance. We report both the mean and variance of accuracy using 8 different seeds and 5 different permutations of n-shots. This table is extended from Table 3.

| $\Psi_{\text{LLM}}(\cdot)$ | $\mathcal{D}$ | $n$ | FPB | | | SST-5 | | | TREC | | |
|---|---|---|---|---|---|---|---|---|---|---|---|
| | | | Random | Similarity | Diversity | Random | Similarity | Diversity | Random | Similarity | Diversity |
| GPT-2 (0.3B) | $\mathcal{D}_{\text{TRAIN}}$ | 1 | 58.3 (5.7) | 68.4 (0.1) | 67.4 (0.1) | 55.5 (4.8) | 60.2 (0.4) | 58.4 (0.2) | 59.2 (5.2) | 70.0 (0.1) | 68.0 (0.1) |
| | | 2 | 58.5 (5.2) | 72.3 (0.4) | 70.1 (0.2) | 58.5 (4.2) | 60.4 (0.6) | 61.2 (0.4) | 57.7 (5.2) | 70.1 (0.2) | 70.3 (0.4) |
| | | 5 | 67.8 (5.1) | 66.2 (0.4) | 65.7 (0.3) | 58.6 (5.2) | 60.4 (0.7) | 61.8 (0.5) | 66.3 (4.5) | 72.8 (0.4) | 70.2 (0.5) |
| | | 10 | 58.2 (4.4) | 63.3 (0.6) | 65.6 (0.3) | 61.4 (4.3) | 60.4 (0.4) | 61.8 (0.2) | 60.9 (3.8) | 71.3 (0.5) | 72.5 (0.9) |
| | $\mathcal{D}_{\text{FEEDER}}$ | 1 | **65.0** (5.5) | **77.3** (1.3) | **73.3** (1.3) | **61.7** (4.2) | **74.8** (1.8) | **74.4** (0.8) | **63.9** (4.0) | **74.3** (0.7) | **75.3** (0.7) |
| | | 2 | **62.2** (3.4) | **75.0** (1.1) | **74.3** (1.5) | **62.3** (3.4) | **63.4** (1.8) | **62.6** (1.2) | **60.1** (3.5) | **76.1** (1.7) | **74.4** (0.9) |
| | | 5 | **70.4** (3.2) | **78.8** (1.6) | **76.4** (1.0) | **62.4** (4.2) | **62.2** (1.4) | **66.4** (1.3) | **68.8** (3.2) | **77.2** (3.3) | **76.6** (2.9) |
| | | 10 | **62.3** (3.3) | **80.6** (1.3) | **78.6** (1.9) | **63.9** (4.5) | **78.6** (1.9) | **71.0** (1.2) | **68.7** (2.7) | **72.2** (1.7) | **75.7** (1.9) |
| GPT-2 (0.8B) | $\mathcal{D}_{\text{TRAIN}}$ | 1 | 60.3 (4.7) | 73.4 (0.1) | 73.4 (0.1) | 57.5 (5.1) | 64.3 (0.2) | 64.3 (0.2) | 61.1 (5.2) | 77.3 (0.1) | 77.3 (0.1) |
| | | 2 | 62.5 (5.2) | 75.3 (0.4) | 75.1 (0.3) | 62.5 (4.2) | 65.4 (0.6) | 66.2 (0.4) | 62.7 (5.2) | 78.1 (0.2) | 79.3 (0.4) |
| | | 5 | 71.8 (5.1) | 72.2 (0.4) | 70.1 (0.3) | 63.6 (5.2) | 67.4 (0.7) | 68.6 (0.6) | 64.3 (4.5) | 76.8 (0.4) | 74.2 (0.5) |
| | | 10 | 63.2 (4.4) | 67.3 (0.6) | 68.6 (0.3) | 66.4 (4.3) | 68.4 (0.4) | 67.8 (0.6) | 66.9 (3.8) | 78.3 (0.5) | 75.5 (0.9) |
| | $\mathcal{D}_{\text{FEEDER}}$ | 1 | **69.0** (5.3) | **81.3** (1.3) | **81.3** (1.3) | **59.8** (4.2) | **72.8** (0.8) | **72.8** (0.8) | **65.9** (4.0) | **83.3** (0.7) | **83.3** (0.7) |
| | | 2 | **73.2** (3.4) | **82.0** (1.1) | **83.3** (1.5) | **65.3** (3.4) | **73.4** (1.8) | **72.6** (1.2) | **62.1** (3.5) | **80.1** (1.7) | **82.2** (0.9) |
| | | 5 | **74.4** (3.4) | **84.8** (1.6) | **86.4** (1.4) | **67.4** (3.9) | **77.5** (1.0) | **76.7** (1.4) | **69.8** (3.2) | **83.2** (3.3) | **84.6** (2.9) |
| | | 10 | **75.3** (3.3) | **85.6** (1.3) | **87.6** (1.9) | **58.9** (3.5) | **78.6** (1.7) | **79.0** (1.2) | **69.7** (2.7) | **86.2** (1.7) | **85.7** (1.9) |
| GPT-neo (1.3B) | $\mathcal{D}_{\text{TRAIN}}$ | 1 | 62.7 (5.7) | 78.4 (0.1) | 78.4 (0.1) | 60.3 (4.1) | 66.6 (1.4) | 66.6 (1.4) | 63.3 (5.2) | 79.5 (0.4) | 79.5 (0.4) |
| | | 2 | 63.1 (4.6) | 74.2 (0.3) | 73.1 (0.2) | 64.5 (3.2) | 66.8 (0.8) | 68.4 (0.7) | 63.5 (5.7) | 81.2 (0.4) | 81.4 (0.6) |
| | | 5 | 70.8 (5.1) | 73.3 (0.1) | 72.7 (0.2) | 63.6 (4.1) | 70.8 (0.4) | 70.8 (0.4) | 67.8 (4.7) | 80.6 (0.5) | 82.0 (0.4) |
| | | 10 | 62.2 (4.4) | 63.0 (0.6) | 69.6 (0.5) | 65.8 (2.9) | 69.5 (0.3) | 68.8 (0.6) | 68.1 (3.8) | 78.8 (0.4) | 82.4 (0.5) |
| | $\mathcal{D}_{\text{FEEDER}}$ | 1 | **73.0** (4.4) | **83.5** (1.5) | **83.5** (1.5) | **63.3** (3.1) | **72.7** (1.3) | **72.7** (1.3) | **64.6** (3.2) | **84.6** (0.8) | **84.6** (0.8) |
| | | 2 | **76.1** (3.8) | **84.1** (1.4) | **82.5** (1.7) | **65.6** (2.7) | **76.4** (0.7) | **78.6** (0.8) | **64.2** (3.7) | **85.5** (0.7) | **86.3** (0.9) |
| | | 5 | **75.7** (3.5) | **90.7** (1.5) | **88.1** (1.9) | **67.4** (2.9) | **79.5** (1.8) | **79.7** (1.5) | **70.8** (3.2) | **88.2** (2.3) | **89.6** (1.9) |
| | | 10 | **77.5** (3.3) | **92.6** (1.3) | **90.6** (1.8) | **68.9** (2.0) | **82.6** (1.7) | **80.0** (1.6) | **73.7** (2.7) | **91.2** (1.7) | **86.7** (1.9) |

versity. Formally, we have:

$$\text{DIV}(\boldsymbol{x}_m, \boldsymbol{x}_n) = \text{SIM}(\boldsymbol{x}_m, \boldsymbol{x}_n) - \eta \cdot \max_{\boldsymbol{x}_{n'} \in \mathcal{D}'_{\text{SELECT}}} \text{SIM}(\boldsymbol{x}_m, \boldsymbol{x}_{n'}),$$

$$(28)$$

where $\boldsymbol{x}_n \in \mathcal{D}_{\text{SELECT}} - \mathcal{D}'_{\text{SELECT}}$, and $\mathcal{D}'_{\text{SELECT}}$ denotes the set of previously selected instances. We can see that Diversity prefers the instance that is both similar to the test samples meanwhile distant to previously selected instances. $\eta$ is a hyper-parameter to balance the above two parts. We set $\eta = 1$ in our experiment.

The fourth one is an uncertainty-based selector (Köksal et al., 2022), denoted as Uncertainty, which conducts selections according to their uncertainty metric;

The fifth one is a clustering-based selector (Zhou et al., 2023), denoted as Clustering, which searches demonstrations by clustering.

The sixth one uses LLMs as latent variable models (Wang et al., 2024), denoted as Latent, which learns latent variables for down-streaming ICL.

In our experiment, we run our approximation algorithm for 1 run to get $\mathcal{D}_{\text{FEEDER}}$, and then treat $\mathcal{D}_{\text{FEEDER}}$ as the selection pool for the above demonstration selectors. In our results, we report both the mean and variance of accuracy using 8 different seeds and 5 different permutations of n-shots.

We also want to emphasize that since our pre-selector and pre-selection process are novel, we evaluate the performance of FEEDER in an ablation fashion. Specifically, our results (denoted as $\mathcal{D}_{\text{FEEDER}}$ in the $\mathcal{D}$ column) can be interpreted as FEEDER + X (where X represents any demonstration retriever described above), meaning that FEEDER is used for pre-selection of input demonstrations, and X is used to select specific demonstrations considering the target inputs. Our baseline (denoted as $\mathcal{D}_{\text{TRAIN}}$ in the $\mathcal{D}$ column) can be formulated as X + X, meaning X is used for both pre-selection of input demonstrations and for selecting specific demonstrations with regard to the target inputs.

### A5.2. Additional Results with Diverse Datasets

We report performance comparison results on text classification datasets SUBJ, SST-2, and COLA datasets in Table 1. We include the results of FPB, SST-5, and TREC datasets in Table A3, whose trend is consistent with our results in Table 1. These results further verify the superiority of our FEEDER in ICL.

Besides three basic demonstration selectors, denoted as Random, Similarity, and Diversity, we also examine the performance of FEEDER with some recently proposed demonstration selectors, denoted as Uncertainty, Clustering, and Latent. We summarize the corresponding results in Table A4, whose trend is consistent with our results in Table A2. Over-

all, compared to using the entire training dataset $\mathcal{D}_{\text{TRAIN}}$ as the retrieval pool, treating its core set $\mathcal{D}_{\text{FEEDER}}$ as the retrieval pool can improve the LLM performance at most cases. These results are consistent with the analysis reported in Section 5.1, which together verify that our FEEDER collaborating with various demonstration selectors works well in ICL.

### A5.3. Additional Results with Diverse Demonstration Selectors

We report performance comparison results on the reasoning dataset GSM8K and the semantic parsing dataset SM-CALFlow in Table 2. The corresponding results for additional demonstration selectors, Clustering and Latent, are provided in Table A6, showing a similar trend. Together, these results further demonstrate the superiority of our FEEDER framework in the in-context learning setting.

### A5.4. Performance Comparison with Different Number of Iterations

We report performance comparison results of FEEDER with different runs $R$ in Figure 3 in Section 5.1. Here, we further evaluate the performance of FEEDER with different iterations $K$. The corresponding results are depicted in Figure A6. This figure shows a similar trend to Figure 3, which also indicates that our method can allow the LLM to achieve comparable performance with fewer samples.

Combining all the above results, we observe that both increasing the tree depth (i.e., the number of iterations $K$) in each round and increasing the number of rounds $R$ contribute to reducing the size of the resulting FEEDER set. However, there are notable trade-offs between these two approaches. Increasing the tree depth is computationally more expensive but offers greater robustness, as it minimizes the risk of mistakenly filtering out useful samples. On the other hand, increasing the number of rounds is relatively inexpensive but carries a higher likelihood of discarding valuable data points due to less rigorous evaluations.

## A6. Scaling Up FEEDER into Real-world Applications

### A6.1. Scaling up FEEDER to larger LLMs.

As the LLM scales up in size (e.g., scaling up to Llama-65B (Touvron et al., 2023) and Gemma-70B (Team et al., 2024)), the execution of our approximation algorithm for searching $\mathcal{D}_{\text{FEEDER}}$ can become exceedingly time-consuming. In response to this challenge, we propose a strategy wherein a smaller LLM is employed to generate a FEEDER set, which is then stored and utilized by the larger LLM. To assess the viability of this approach, we conducted an experiment

*Table A6.* Performance comparisons on reasoning GSM8K dataset and semantic-parsing SMCALFlow dataset are conducted in the in-context learning setting. We report both the mean and variance of accuracy using 8 different seeds and 5 different permutations of n-shots. This table is extended from Table 2.

| $\Psi_{\text{LLM}}(\cdot)$ | $\mathcal{D}$ | $n$ | GSM8K | | SMCALFlow | |
|---|---|---|---|---|---|---|
| | | | Clustering | Latent | Clustering | Latent |
| Gemma-2 (2B) | $\mathcal{D}_{\text{TRAIN}}$ | 1 | 16.17 (0.18) | 16.20 (0.19) | 20.02 (0.21) | 19.54 (0.14) |
| | | 2 | 19.89 (0.96) | 20.52 (0.15) | 22.58 (0.45) | 23.05 (0.36) |
| | | 5 | 21.31 (0.84) | 23.56 (0.66) | 29.30 (0.90) | 28.65 (0.95) |
| | | 10 | 22.52 (0.49) | 23.85 (0.65) | 30.12 (1.11) | 31.11 (0.91) |
| | $\mathcal{D}_{\text{FEEDER}}$ | 1 | **17.25** (0.21) | **16.68** (0.24) | **21.12** (1.78) | **20.89** (1.21) |
| | | 2 | **20.68** (0.83) | **21.01** (0.85) | **22.85** (2.65) | **25.03** (0.18) |
| | | 5 | **22.55** (0.75) | 23.05 (0.77) | **31.20** (1.15) | **29.54** (4.58) |
| | | 10 | **22.75** (0.85) | **24.02** (2.20) | **32.10** (2.01) | **32.48** (1.52) |
| GPT-3 (6B) | $\mathcal{D}_{\text{TRAIN}}$ | 1 | 2.95 (0.12) | 2.87 (0.25) | 9.95 (0.79) | 9.21 (0.85) |
| | | 2 | 4.78 (0.33) | 4.21 (0.25) | 10.12 (0.46) | 10.14 (0.88) |
| | | 5 | 7.21 (0.78) | 8.00 (1.05) | 12.31 (1.11) | 12.15 (1.30) |
| | | 10 | 8.05 (1.20) | 7.44 (1.25) | 14.14 (1.57) | 13.99 (1.54) |
| | $\mathcal{D}_{\text{FEEDER}}$ | 1 | **4.10** (0.22) | **3.25** (0.24) | **12.52** (1.13) | **11.42** (1.02) |
| | | 2 | **4.26** (0.64) | **4.55** (0.82) | **11.73** (0.54) | **12.05** (0.80) |
| | | 5 | **8.85** (1.28) | **8.14** (0.87) | **13.58** (1.44) | **12.44** (1.69) |
| | | 10 | **9.52** (1.88) | **8.50** (1.21) | **15.08** (1.91) | **16.50** (1.25) |
| Llamma-2 (7B) | $\mathcal{D}_{\text{TRAIN}}$ | 1 | 3.68 (0.89) | 3.98 (0.88) | 10.12 (0.95) | 9.25 (0.85) |
| | | 2 | 5.20 (0.38) | 5.55 (0.85) | 11.05 (1.36) | 12.52 (1.45) |
| | | 5 | 7.58 (0.89) | 7.52 (0.96) | 15.18 (1.15) | 15.30 (1.20) |
| | | 10 | 9.85 (0.85) | 9.21 (0.98) | 17.95 (1.25) | 18.55 (2.01) |
| | $\mathcal{D}_{\text{FEEDER}}$ | 1 | **4.25** (0.21) | **4.17** (0.89) | **11.89** (0.51) | **12.05** (0.63) |
| | | 2 | **5.88** (0.63) | **6.02** (0.58) | **13.03** (0.16) | **14.13** (1.10) |
| | | 5 | **8.22** (1.01) | **9.17** (0.98) | **18.20** (3.66) | **19.66** (5.20) |
| | | 10 | **10.17** (1.22) | **9.65** (0.83) | **22.11** (1.22) | **21.25** (1.26) |

comparing the performance of using GPT-2 variants and GPT-neo as the LLMs for obtaining a FEEDER set, and then we use this set as the retrieval pool to acquire demonstrations for GPT-neo. Results summarized in Table A7 demonstrate that even when $\mathcal{D}_{\text{FEEDER}}$ is pre-selected by a small LLM, it contributes to improved performance, compared to using $\mathcal{D}_{\text{TRAIN}}$, as reported in Table 1. This observation suggests the potential feasibility of employing a more compact LLM for pre-selecting $\mathcal{D}_{\text{FEEDER}}$ to enhance the performance of a larger LLM.

### A6.2. Scaling up FEEDER by Incremental Update.

Notice that numerous real-world datasets are temporal and require frequent updates. Re-running the tree based approximation algorithm for FEEDER over all samples can be excessively time-consuming. To address this, we design an incremental approach, treating the unchanged portion as a plug-and-play FEEDER set and the LLM as a whole, forming a new "LLM". Therefore, we can apply FEEDER solely to compute incremental data for the modified part, encompassing newly added and modified data points. Also, a significant challenge of FEEDER arises from the temporal nature of many real-world datasets, some of which require frequent updates, potentially on a daily basis. The conventional approach of recalculating a FEEDER over all

unchanged and changed samples can be time-consuming in such dynamic scenarios. To address this challenge, we introduce an incremental update algorithm for FEEDER, enabling the efficient re-computation of only the changed portions, including newly added and modified samples.

As depicted in Figure A3, once a FEEDER set for the original dataset is generated, we treat the unchanged part of plug-and-play plugged data and the LLM as a whole (depicted by the dashed box) as a new "LLM". Subsequently, we apply FEEDER exclusively to compute incremental data for the changed part, covering newly added and modified data points. This strategy aims to enhance the efficiency and responsiveness of FEEDER in the context of evolving and temporal datasets.

## A7. Integrating Algorithm 4 in FEEDER

One limitation to directly applying Algorithm 3 or 4 is that $\mathcal{D}_{\text{TRAIN}}$ is too large to be directly used as input demonstrations. For this purpose, we incorporate running Algorithm 4 for one round into our FEEDER as follows. As shown in Figure A2, we place Algorithm 4 after the demonstration selector to filter out the unnecessary parts from the pre-selected data. Concretely, we first select $n$ samples from our FEEDER set (i.e., $\mathcal{D}_{\text{FEEDER}}$), then filter retrieved samples

*Table A7.* Performance comparisons among using different LLMs GPT-2, GPT-3, GPT-neo as the base for acquiring a FEEDER set and using GPT-neo for inference on COLA dataset are conducted in the in-context learning setting. We report both the mean and variance of accuracy using 8 different seeds and 5 different permutations of n-shots.

| $\Psi_{\text{LLM}}(\cdot)$ | $\mathcal{D}$ | $n$ | GPT-2 (0.8B) | | | GPT-3 (6B) | | | GPT-neo (1.3B) | | |
|---|---|---|---|---|---|---|---|---|---|---|---|
| | | | Random | Similarity | Diversity | Random | Similarity | Diversity | Random | Similarity | Diversity |
| GPT-neo (1.3B) | $\mathcal{D}_{\text{FEEDER}}$ | 1 | 23.7 (5.7) | 31.0 (1.3) | 31.0 (1.3) | 25.3 (4.1) | 34.6 (1.8) | 34.6 (1.8) | 28.3 (5.4) | 34.8 (1.3) | 34.8 (1.3) |
| | | 2 | 45.1 (5.6) | 49.7 (1.4) | 46.1 (0.8) | 58.5 (3.2) | 57.8 (1.2) | 56.4 (1.0) | 69.3 (3.7) | 64.7 (1.4) | 64.7 (1.6) |
| | | 5 | 49.4 (4.6) | 58.1 (2.5) | 59.1 (1.9) | 54.6 (3.8) | 64.5 (1.1) | 61.7 (2.4) | 68.7 (3.2) | 67.2 (2.4) | 65.8 (1.8) |
| | | 10 | 59.4 (4.6) | 62.4 (1.5) | 65.8 (1.5) | 60.6 (3.8) | 64.7 (1.8) | 66.0 (1.4) | 69.8 (2.8) | 68.8 (1.4) | 68.9 (1.3) |

**Integrating Exact Extractor of FEEDER into FEEDER**

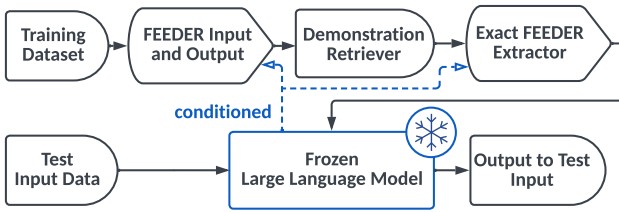

*Figure A2.* Integrating our extraction algorithm for FEEDER (i.e., Algorithm 4) into our in-context learning framework (as introduced in Figure 1(a)).

by running Algorithm 4 for one round (treating the set of retrieved samples as $\mathcal{D}_{\text{IN}}$). Then, re-select $n - |\mathcal{D}_{\text{OUT}}|$ where $\mathcal{D}_{\text{OUT}}$ indicates the output of Algorithm 4.

**Scaling Up FEEDER by Incremental Update**

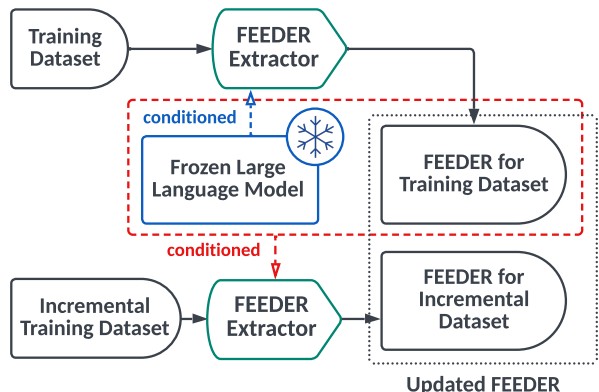

*Figure A3.* In order to scale up FEEDER for real-world applications dealing with dynamic data, we introduce an incremental update algorithm. This algorithm is designed to efficiently handle changes in training examples, avoiding the need to recompute over unchanged training examples.

## A8. FEEDER **in Fine-tuning Setting**

### A8.1. Implementation Details

As summarized in Algorithm 1 in Section 2, we can integrate our FEEDER selection and LLM fine-tuning into a bi-level optimization problem. To evaluate the performance

of our bi-level optimization, we first run Algorithm 1 for one run to get a pre-selected FEEDER set (i.e., $\mathcal{D}_{\text{FEEDER}}$) and a tuned LLM. Then, we update our FEEDER set with the tuned LLM and evaluate the performance of LLM in the in-context learning setting (i.e., few-shot inference), where we allow the LLM to retrieve relevant information from the pre-selected FEEDER set or the training dataset.

Concretely, our baseline is to first tune the LLM on the entire training dataset (i.e., $\mathcal{D}_{\text{TRAIN}}$) and then do few-shot inference on the test dataset (i.e., $\mathcal{D}_{\text{TEST}}$) with $\mathcal{D}_{\text{TRAIN}}$ as the retrieval pool. In contrast, ours is to first pre-select a FEEDER set (i.e., $\mathcal{D}_{\text{FEEDER}}$) from $\mathcal{D}_{\text{TRAIN}}$ and then tune the LLM on $\mathcal{D}_{\text{FEEDER}}$. Our FEEDER set is updated according to the tuned LLM using Algorithm 2 for 1 run, and our approach is evaluated on $\mathcal{D}_{\text{TEST}}$ with the updated $\mathcal{D}_{\text{FEEDER}}$ as the retrieval pool.

We conduct the fine-tuning pipeline in this manner to not only verify the superiority of our FEEDER but also to validate our bi-level optimization framework, which is able to tune both the FEEDER set and the LLMs in each loop.

We list some key hyper-parameters for fine-tuning as follows. The batch size is set as 32, the warm steps is set as 100, the learning rate is set as $5 \times 10^{-5}$, and the weight decay is set as 0.01. All our experiments are conducted with NVIDIA A100s[3].

### A8.2. Additional Results with Diverse Datasets

We report performance comparison results on text classification datasets SUBJ, SST-2, and COLA datasets in Table 3. We include the results of FPB, SST-5, and TREC datasets in Table A5, whose trend is consistent with our analysis in Section 5.2. These results further verify the superiority of our FEEDER in the fine-tuning setting.

*Table A8.* Performance comparisons between using randomly-selected $\mathcal{D}^*_{\text{TRAIN}}$ (where $|\mathcal{D}^*_{\text{TRAIN}}| = |\mathcal{D}_{\text{TRAIN}}|$) as the base for acquiring a FEEDER set and using GPT-neo for inference on SST-2, SST-5, and COLA datasets are conducted in ICL. We report both the mean and variance of accuracy using 8 different seeds and 5 different permutations of n-shots.

| $\Psi_{\text{LLM}}(\cdot)$ | $\mathcal{D}$ | $n$ | SST-2 | | | SST-5 | | | COLA | | |
|---|---|---|---|---|---|---|---|---|---|---|---|
| | | | Random | Similarity | Diversity | Random | Similarity | Diversity | Random | Similarity | Diversity |
| GPT-neo (1.3B) | $\mathcal{D}_{\text{TRAIN}}$ | 2 | 76.8 (3.5) | 81.5 (0.1) | 76.3 (0.4) | 17.9 (3.6) | 26.9 (0.1) | 22.7 (0.1) | 30.7 (3.1) | 55.5 (0.2) | 56.5 (0.4) |
| | | 5 | 65.1 (3.5) | 80.8 (0.2) | 66.1 (0.3) | 19.0 (3.9) | 29.2 (0.1) | 25.1 (0.1) | 40.0 (3.6) | 55.9 (0.1) | 52.5 (0.2) |
| | $\mathcal{D}^*_{\text{TRAIN}}$ | 2 | 73.2 (3.6) | 77.8 (2.3) | 72.4 (2.4) | 14.5 (3.8) | 23.3 (3.6) | 20.0 (1.0) | 28.3 (5.4) | 48.8 (3.3) | 49.7 (3.1) |
| | | 5 | 62.4 (3.5) | 77.6 (3.3) | 62.2 (2.2) | 16.6 (2.8) | 25.5 (2.1) | 27.7 (2.8) | 33.8 (4.4) | 50.2 (3.4) | 48.7 (2.8) |
| | $\mathcal{D}_{\text{FEEDER}}$ | 2 | **75.1** (2.8) | **82.6** (2.1) | **78.5** (1.9) | **19.7** (2.7) | **27.4** (2.1) | **22.8** (1.8) | **59.3** (3.7) | **64.7** (1.4) | **64.7** (1.6) |
| | | 5 | **73.2** (4.2) | **82.9** (2.7) | **71.6** (2.4) | **19.2** (3.2) | **30.2** (1.1) | **26.4** (2.4) | **58.7** (3.2) | **67.2** (2.4) | **65.8** (1.8) |

## A9. In-Depth Analysis of FEEDER

### A9.1. Performance Gap between using FEEDER and RAN as Pre-Selector

As our paper introduces a new pre-selection stage before the demonstration selection process, we also include an ablation study that randomly selects the same number of samples to form a randomly selected training dataset, denoted as $\mathcal{D}^*_{\text{TRAIN}}$, which matches the sample size of $\mathcal{D}_{\text{FEEDER}}$. The corresponding results are reported in Table A8. A comparison of Table A8 with Tables 1 and A3 indicates that replacing the entire training dataset with randomly selected samples significantly degrades LLM performance. In contrast, the FEEDER-selected samples act as a core set that summarizes the key information of the entire training dataset. By focusing on high-value samples, our approach enables LLMs to achieve better performance, effectively leveraging the essential knowledge within the dataset.

### A9.2. Performance Gap among Using Different Numbers of Rounds and Different Depths of Tree

FEEDER's performance first rises and then drops with increasing tree algorithm runs $R$. Figure A4 visualizes the impact of employing different numbers of runs of our approximation algorithm (as described in Section 4) to derive $\mathcal{D}_{\text{FEEDER}}$ for fine-tuning NEO. For ease of comparison, the results of fine-tuning NEO on $\mathcal{D}_{\text{TRAIN}}$ are also included with the blue line. The observations suggest that fine-tuning with a smaller dataset with high data quality can enhance performance, but excessively reducing the dataset size may not lead to the desired outcomes. Also, it also indicates that fine-tuning LLMs on "unnecessary" data samples would not help.

As described in Section 4, we set the tree depth to 2 (corresponding to $K = 1$), utilizing the one-shot inference capability of LLMs as the sufficiency filter to eliminate unnecessary samples. To further explore the performance im-

[3] https://www.nvidia.com/en-us/data-center/a100/

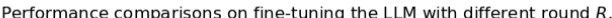

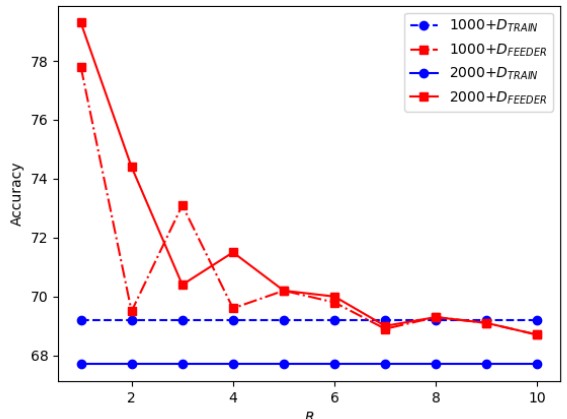

*Figure A4.* Performance comparisons on fine-tuning NEO with running our approximation algorithm to pre-select $\mathcal{D}_{\text{FEEDER}}$ with different run $R$. Our evaluation operates on COLA dataset in the zero-shot setting after fine-tuning on 1000 and 2000 batches.

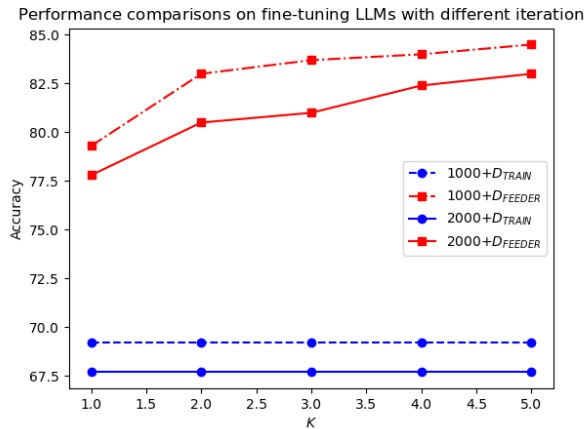

*Figure A5.* Performance comparisons on fine-tuning NEO with running our approximation algorithm to pre-select $\mathcal{D}_{\text{FEEDER}}$ with different round $K$. Our evaluation operates on COLA dataset in the zero-shot setting after fine-tuning on 1000 and 2000 batches.

pact of varying tree depths, we investigate the performance gap associated with different depths of the tree. Figure A5 visualizes the impact of employing different numbers of runs of our approximation algorithm (as outlined in Section 4) to derive $\mathcal{D}_{\texttt{FEEDER}}$ for fine-tuning NEO. For ease of comparison, the results of fine-tuning NEO on $\mathcal{D}_{\texttt{TRAIN}}$ are also presented as a baseline (depicted by the blue line). The results suggest that fine-tuning with a smaller, high-quality dataset can significantly enhance performance. However, when comparing to Figure A4, we observe that increasing the tree depth leads to more "smoothing" changes in the LLM performance. There are two potential explanations for this phenomenon: (i) The hyper-parameter $K$, which controls the tree depth, typically changes within a relatively small scope compared to $R$ due to its high computational cost and diminishing returns. While increasing $K$ initially enhances the filtering process by leveraging deeper evaluations of sufficiency, the marginal improvements in the quality or size of the resulting FEEDER set decreases as $K$ grows. (ii) Increasing the tree depth corresponds to performing n-shot inference to satisfy the sufficiency condition described in Eq. (7). This is significantly more challenging than a one-shot inference check and results in a much smaller reduction in the number of samples in the training dataset. (iii) Leveraging the n-shot inference capability of LLMs may yield more robust results. Specifically, the unnecessary samples filtered out by an n-shot sufficiency check are more likely to be genuinely unnecessary, thereby ensuring a higher-quality training set for fine-tuning.

### A9.3. Performance Gap between our Approximately Computed FEEDER Set and Exact FEEDER Set

As described in Section 4, our approximation algorithm ensures the sufficiency of the resulting FEEDER set but does not guarantee the necessity of each sample within it. To address this, we employ the integration method outlined in Appendix A7, which ensures that the selected demonstrations are both sufficient and necessary. We denote this refined set as $\mathcal{D}_{\texttt{FEEDER}}^{*}$. We compare the performance of few-shot preference using $\mathcal{D}_{\texttt{FEEDER}}$, $\mathcal{D}_{\texttt{FEEDER}}^{*}$, and $\mathcal{D}_{\texttt{TRAIN}}$, with the results summarized in Table A9. The results indicates that $\mathcal{D}_{\texttt{FEEDER}}^{*}$ achieves a slight improvement in LLM performance compared to $\mathcal{D}_{\texttt{FEEDER}}$, further validating the effectiveness of integrating sufficiency and necessity in the pre-selection process.

We further evaluate the robustness of our $\mathcal{D}_{\texttt{FEEDER}}^{*}$ and $\mathcal{D}_{\texttt{FEEDER}}$ by duplicating the training dataset $\mathcal{D}_{\texttt{TRAIN}}$. The duplicated dataset is denoted as $\mathcal{D}_{\texttt{TRAIN}}'$, and the corresponding resulting sets derived using our approximation and integration methods are denoted as $\mathcal{D}_{\texttt{FEEDER}}'$ and $\mathcal{D}_{\texttt{FEEDER}}^{*'}$ respectively. The results of this evaluation are summarized in Table A9. From the table, we observe that both random and similarity-based demonstration retrievers are significantly

impacted by the duplicated dataset. This is because the retrieved demonstrations can include duplicates, particularly when using a similarity-based retriever, as similarity scores are calculated independently for each sample. In contrast, our $\mathcal{D}_{\texttt{FEEDER}}'$ and $\mathcal{D}_{\texttt{FEEDER}}^{*'}$ act as "weak" and "strong" filters, respectively, by effectively removing redundant or unnecessary samples from the input. The "weak" filter provided by $\mathcal{D}_{\texttt{FEEDER}}'$ ensures sufficiency by eliminating a significant portion of redundant data while maintaining the core information needed for the task. On other hand, the "strong" filter represented by $\mathcal{D}_{\texttt{FEEDER}}^{*'}$ not only ensures sufficiency but also guarantees necessity, leading to an even more refined dataset that further enhances model robustness and performance. This differentiation highlights the flexibility and effectiveness of our filtering mechanisms in handling noisy or duplicated datasets.

## A10. Case Study with Artificial Data Points Generated by LLMs

### A10.1. Case Study for Transitivity of LLMs

To illustrate the transitivity of LLMs, we conducted a simple experiment using gpt-3.5-turbo. We prompted the model with the question *which place does Jerry lives in?* LLM responses with *I'm sorry, but I don't have access to personal information about individuals, including your friend Jerry.* Then, let $\mathcal{D}_{\texttt{A}}$, $\mathcal{D}_{\texttt{B}}$, $\mathcal{D}_{\texttt{C}}$ denote the city, the country, and the continent he lives in. Then, we can observe that if we tell the LLM about the city (e.g., London), then the LLM can tell about the country (e.g., United Kingdom); if we tell the LLM about the country (e.g., United Kingdom), then the LLM can tell about the continent (e.g., Europe). Also, if we tell the LLM about the city (e.g., London), then we ask about the continent, and the LLM also can tell (e.g., Europe). This observation is one case to demonstrate the transitivity of sufficiency.

### A10.2. Case Study for Demonstrations Selection Should be Aware of LLMs in Use

Subsequently, we conduct a case study to substantiate the central proposition of this paper: Whether the assessment of the quality of a demonstration should depend on the specific LLM in use.

We consider the factual error made by Google Bard in the first demo[4]. Bard said *The James Webb Space Telescope took the very first pictures of a planet outside of our own solar system*. However, the fact is *The Very Large Telescope took the very first pictures of a planet outside of our own solar system*. Based on the above statements, we produce

---

[4] https://www.theverge.com/2023/2/8/23590864/google-ai-chatbot-bard-mistake-error-exoplanet-demo

*Table A9.* Results of performance difference between using $\mathcal{D}^*_{\text{FEEDER}}$ (derived by using FEEDER version introduced in Appendix A7), we also evaluate the performance of our variants of FEEDER with duplicated training dataset. We evaluate GPT-neo's performance on the n-shot settings.

| $\Psi_{\text{LLM}}(\cdot)$ | $\mathcal{D}$ | $n$ | SST-2 | | | SST-5 | | | COLA | | |
|---|---|---|---|---|---|---|---|---|---|---|---|
| | | | Random | Similarity | Diversity | Random | Similarity | Diversity | Random | Similarity | Diversity |
| GPT-neo (1.3B) | $\mathcal{D}_{\text{TRAIN}}$ | 2 | 76.8 (3.5) | 81.5 (0.1) | 76.3 (0.4) | 17.9 (3.6) | 26.9 (0.1) | 22.7 (0.1) | 30.7 (3.1) | 55.5 (0.2) | 56.5 (0.4) |
| | | 5 | 65.1 (3.5) | 80.8 (0.2) | 66.1 (0.3) | 19.0 (3.9) | 29.2 (0.1) | 25.1 (0.1) | 40.0 (3.6) | 55.9 (0.1) | 52.5 (0.2) |
| | $\mathcal{D}'_{\text{TRAIN}}$ | 2 | 73.4 (6.6) | 78.4 (0.3) | 75.4 (2.4) | 14.9 (3.8) | 22.7 (2.9) | 21.7 (1.0) | 29.3 (5.4) | 49.8 (1.3) | 52.7 (3.3) |
| | | 5 | 59.4 (3.5) | 75.3 (1.3) | 64.1 (3.5) | 17.5 (2.8) | 23.5 (2.1) | 22.7 (2.8) | 37.8 (4.2) | 51.2 (1.4) | 51.0 (2.3) |
| | $\mathcal{D}_{\text{FEEDER}}$ | 2 | 75.1 (2.8) | 82.6 (2.1) | 78.5 (1.9) | 19.7 (2.7) | 27.4 (2.1) | 22.8 (1.8) | 59.3 (3.7) | 64.7 (1.4) | 64.7 (1.6) |
| | | 5 | 73.2 (4.2) | 82.9 (2.7) | 71.6 (2.4) | 19.2 (3.2) | 30.2 (1.1) | 26.4 (2.4) | 58.7 (3.2) | 67.2 (2.4) | 65.8 (1.8) |
| | $\mathcal{D}'_{\text{FEEDER}}$ | 2 | 74.3 (2.9) | 81.3 (1.1) | 76.4 (1.8) | 18.2 (2.2) | 26.1 (2.1) | 21.0 (1.8) | 58.3 (2.7) | 62.5 (1.4) | 63.5 (1.1) |
| | | 5 | 71.1 (3.2) | 80.0 (2.4) | 69.8 (2.1) | 19.0 (2.0) | 29.4 (1.3) | 25.3 (2.1) | 57.5 (3.0) | 65.0 (2.4) | 64.1 (1.8) |
| | $\mathcal{D}^*_{\text{FEEDER}}$ | 2 | 75.6 (1.8) | 83.1 (1.0) | 79.0 (1.1) | 20.1 (2.0) | 27.8 (2.3) | 23.1 (1.2) | 60.2 (3.2) | 64.9 (1.4) | 65.0 (1.1) |
| | | 5 | 73.7 (4.1) | 82.8 (2.2) | 71.8 (2.1) | 19.0 (3.0) | 31.2 (1.0) | 26.3 (2.1) | 59.2 (2.7) | 67.3 (2.1) | 65.4 (2.2) |
| | $\mathcal{D}^{*'}_{\text{FEEDER}}$ | 2 | 75.2 (2.0) | 82.8 (2.0) | 78.4 (1.3) | 19.9 (2.2) | 27.0 (2.1) | 22.7 (1.8) | 59.4 (1.7) | 64.9 (1.2) | 64.5 (1.2) |
| | | 5 | 73.5 (4.2) | 82.4 (2.2) | 71.3 (2.2) | 18.9 (2.2) | 29.9 (1.0) | 26.2 (1.2) | 56.5 (2.2) | 65.5 (2.2) | 64.7 (1.4) |

the question *What took the very first pictures of a planet outside of our own solar system?*

We ask the question to gpt-3.5-turbo. The corresponding answer is:

*The first pictures of a planet outside of our own system were taken by the Hubble Space Telescope in 2008. The planet, named Fomalhaut b, orbits the star Fomalhaut, which is located about 25 light-years away from Earth in the constellation Piscis Austrinus.*

We can see that the above answer is incorrect. We prompt the fact into the LLM, then the answer is:

*The Very Large Telescope took the very first pictures of a planet outside of our own solar system.*

This indicates that prompting the fact works. We further ask gpt-3.5-turbo to generate some sufficient statements for the fact. The following is its output:

*1. The Very Large Telescope made history by capturing the first-ever images of a planet beyond our solar system.*

*2. Thanks to the advanced technology of the Very Large Telescope, scientists were able to capture the first-ever photographs of an exoplanet.*

*3. The groundbreaking images captured by the Very Large Telescope marked a major milestone in the study of exoplanets.*

*4. The discovery of a planet outside of our solar system was made possible by the cutting-edge capabilities of the Very Large Telescope.*

*5. The Very Large Telescope's ability to capture images of distant celestial bodies allowed scientists to observe an*

*exoplanet for the first time in history.*

We also evaluate separately using the above statements as a prompt to gpt-3.5-turbo, and either one of them can lead to the correct answer. We provide detailed answers as follows.

*1. The Very Large Telescope took the very first pictures of a planet outside of our own solar system.*

*2. The Very Large Telescope took the very first pictures of a planet outside of our own solar system.*

*3. The Very Large Telescope took the very first pictures of a planet outside of our own solar system.*

*4. The Very Large Telescope took the very first pictures of a planet outside of our own solar system.*

*5. The Very Large Telescope took the very first pictures of a planet outside of our own solar system.*

We can see, in this case, that either one of the statements generated by gpt-3.5-turbo, is a sufficient and necessary instance to answer *What took the very first pictures of a planet outside of our own solar system?*

We further evaluate the performance of gpt-j-6b. Without any prompt (i.e., in the zero-shot setting), its answer is:

*The Hubble Space Telescope.*

We then prompt the above 5 statements provided by gpt-3.5-turbo into gpt-j-6b, then the answer is:

*1. The first-ever images of a planet beyond our solar system have been captured by the Very Large Telescope in Chile.*

*2. The Hubble Space Telescope.*

*3. A team of astronomers led by the University of Arizona's*

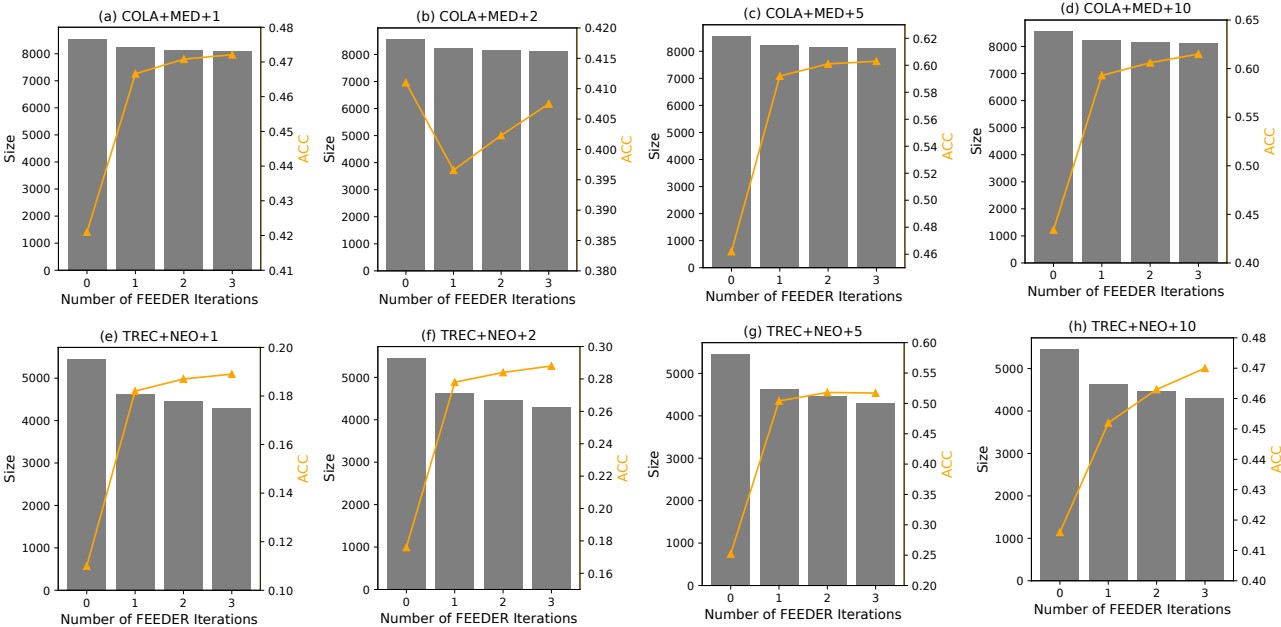

*Figure A6.* Performance comparisons for running our approximation algorithm to pre-select `FEEDER` with different iterations $K$ are evaluated in terms of accuracy (denoted as ACC) with `RAN` as the retriever and the size of the resulting `FEEDER` set (denoted as Size). Each sub-figure is entitled with Dataset+LLM base+n shots.

*Michael Liu.*

*4. The Hubble Space Telescope.*

*5. The Very Large Telescope.*

We can see that only the 1-st statement or the 5-th statement is a sufficient and necessary instance. Combining the results of `gpt-j-6b` and the results of `gpt-3.5-turbo` can verify the core insight of our paper: *the measurement over prompting a demonstration should consider what LLM is in use.*

Furthermore, we also evaluate the performance of 3 GPT variants used in the paper. We begin by evaluating the zero-shot performance of `gpt-neo-1.3B` (denoted as `NEO` in the experiment), and its result is:

*The first pictures of a planet outside of our own solar system were taken by the Voyager 1 spacecraft in 1977.*

We then prompt the above 5 statements into `gpt-neo-1.3B`, then the answer is:

*1. The very large telescope was built in the early 1990s by the European Southern Observatory (ESO) in Chile. The Very Large Telescope.*

*2. The Very Large Telescope in Chile.*

*3. The Very Large Telescope (VLT) in Chile.*

*4. The Very Large Telescope.*

*5. The Very Large Telescope in Chile.*

The above results show that either one of the latter 4 statements is a sufficient instance. The results of `gpt2-large` (denoted as `LAR` in the experiment) show that neither of the 5 statements is a sufficient and necessary instance:

*1. The very large telescope was built in the early 1990s by the European Southern Observatory (ESO) in Chile. The Very Large Telescope.*

*2. The Hubble Space Telescope.*

*3. The first pictures of a planet outside of our own solar system were taken by the Hubble Space Telescope in 1990.*

*4. The Hubble Space Telescope.*

*5. The very first pictures of a planet outside of our own solar system were taken by the Hubble Space Telescope.*

The results of `gpt2-medium` (denoted as `MED` in the experiment) show that only the 4-th statement is not a sufficient and necessary instance:

*1. The Very Large Telescope.*

*2. The Very Large Telescope.*

*3. The Very Large Telescope.*

*4. The Hubble Space Telescope.*

*5. The Very Large Telescope.*

All the above results verify that *quality* of one demonstration

should be LLM-specific, which is the key idea of our paper.

---

**Algorithm 4** Alternative Exact Algorithm for FEEDER

---

**Input:** Training dataset $\mathcal{D}_{\text{TRAIN}}$.
**Output:** Exact FEEDER $\widetilde{\mathcal{D}}_{\text{FEEDER}}$.
Initialize the number of rounds $r = 0$.
Initialize the set of unnecessary data $\mathcal{D}_r = \emptyset$.
**repeat**
 Initialize $k = 1$.
 Initialize $\mathscr{H}_0 = \emptyset$.
 Update input data by removing the unnecessary part $\mathcal{D}_{\text{IN}} = \mathcal{D}_{\text{TRAIN}} - \mathcal{D}_r$.
 **for** *each instance* $(\boldsymbol{x}_n, \boldsymbol{y}_n) \in \mathcal{D}_{\text{IN}}$ **do**
  Check $Y_{(\{\boldsymbol{x}_{n'} | \boldsymbol{x}_{n'} \in \mathcal{D}_{\text{IN}}\})} = \mathbf{1}_{|\mathcal{D}_{\text{IN}}|} | \text{unplug}((\boldsymbol{x}_n, \boldsymbol{y}_n))$; $C, S$ (a), $C = \mathcal{D}_{\text{IN}}$, $S = (Y_{(\{\boldsymbol{x}_{n'} | \boldsymbol{x}_{n'} \in \mathcal{D}_{\text{IN}}\})} = \mathbf{1}_{|\mathcal{D}_{\text{IN}}|})$.
  If (a) holds, let $\mathcal{H}_n = \{(\boldsymbol{x}_n, \boldsymbol{y}_n)\}$ and append $\mathcal{H}_n$ to $\mathscr{H}_0$.
 **repeat**
  **for** *each pair* $(\mathcal{H}_i, \mathcal{H}_j)$ *where* $\mathcal{H}_i, \mathcal{H}_j \in \mathscr{H}_{k-1}$ **do**
   Check $Y_{(\{\boldsymbol{x}_n | \boldsymbol{x}_n \in \mathcal{D}_{\text{IN}}\})} = \mathbf{1}_{|\mathcal{D}_{\text{IN}}|} | \text{unplug}(\mathcal{H}_i \cup \mathcal{H}_j)$; $C, S$ (b), where $C = \mathcal{D}_{\text{IN}}$ and $S = (Y_{(\{\boldsymbol{x}_{n'} | \boldsymbol{x}_{n'} \in \mathcal{D}_{\text{IN}}\})} = \mathbf{1}_{|\mathcal{D}_{\text{IN}}|})$.
   If (b) holds, generate a new node $\mathcal{H}_i \cup \mathcal{H}_j$, append it to $\mathscr{H}_k$, and assign $\mathcal{H}_i \cup \mathcal{H}_j$; otherwise, append $\mathcal{H}_i$ and $\mathcal{H}_j$ to $\mathscr{H}_k$.
   Remove $\mathcal{H}_i, \mathcal{H}_j$ from $\mathscr{H}_{k-1}$, i.e., $\mathscr{H}_{k-1} = \mathscr{H}_{k-1} - \{\mathcal{H}_i, \mathcal{H}_j\}$.
  Grow tree from bottom to top via $k = k + 1$.
 **until** $|\mathscr{H}_k| = 1$ *where we assume the current round is* $K$;
 Let $\mathcal{H}_{\text{UNNCESSARY}}$ denote only one element (i.e. the root node) in $\mathscr{H}_K$.
 Update the number of rounds, i.e., $r = r + 1$.
 Update $\mathcal{D}_r$ to include the unnecessary part $\mathcal{H}_{\text{UNNCESSARY}}$, i.e., $\mathcal{D}_r = \mathcal{D}_r \cup \mathcal{H}_{\text{UNNCESSARY}}$.
**until** $|\mathcal{H}_{\text{UNNCESSARY}}| \leq 1$;
Assign $\widetilde{\mathcal{D}}_{\text{FEEDER}}$ as removing $\mathcal{D}_r$ from $\mathcal{D}_{\text{TRAIN}}$, i.e., $\widetilde{\mathcal{D}}_{\text{FEEDER}} = \mathcal{D}_{\text{TRAIN}} - \mathcal{D}_r$.

---

