# OpenReview forum: "Large Language Models are Demonstration Pre-Selectors for Themselves"
_ICML.cc/2025/Conference — ICML 2025 poster_

### Official Review · Reviewer_mKZf · 2025-03-10

**Overall Recommendation:** 2

**Summary:**

This paper introduces FEEDER, a demonstration pre-selection framework designed to improve the efficiency and effectiveness of large language models (LLMs) in in-context learning (ICL) and fine-tuning tasks. FEEDER identifies a representative subset of training data using two new metrics: "sufficiency" and "necessity." By leveraging this pre-selected subset, the approach reduces computational costs while enhancing model performance. Experimental results show that FEEDER can effectively enhance both ICL and fine-tuning.

**Claims And Evidence:**

The author validate the effectiveness of the method using a small dataset and a small-scale model. However, the current experiments are not sufficient, and larger-scale models need to be used. Additionally, the performance improvement of the model is not significant enough.

**Essential References Not Discussed:**

No.

**Experimental Designs Or Analyses:**

I carefully examine the validity of the experiments, and the current experimental setup is reasonable.

**Methods And Evaluation Criteria:**

The author proposed a new method for selecting data. While it has some scalability, the performance is not particularly impressive. Additionally, the size of the tested model is still too small.

**Other Comments Or Suggestions:**

1.	The paper primarily compares FEEDER with a few selection methods (e.g., similarity and clustering-based approaches). Including comparisons with more recent approaches, such as reinforcement learning-based selection methods, would provide a more comprehensive evaluation.
2.	The concepts of sufficiency and necessity are somewhat abstract, and the paper could benefit from more intuitive, visual examples to help readers understand their impact on demonstration selection.

**Other Strengths And Weaknesses:**

Strengths:
1. The introduction of FEEDER as a demonstration pre-selection framework, along with the novel sufficiency and necessity metrics, offers a fresh perspective on data selection.
2. By reducing redundant data, FEEDER enhances the efficiency of ICL while improving performance.
3. The study evaluates multiple LLMs ranging from 300M to 8B parameters across multiple tasks (text classification, reasoning, semantic parsing), making the results convincing.
4. FEEDER is compatible with various existing demonstration selection strategies.

Weaknesses:
1. The current comparisons primarily involve Random, Similarity, and Diversity selection strategies. However, it remains unclear whether combining FEEDER with previous ICL approaches would yield better results. A more informative comparison would involve integrating FEEDER with advanced ICL techniques instead of only evaluating it against basic selection methods.
2. While FEEDER shows performance improvements in most cases, the reported gains are not particularly significant. The paper should further analyze whether these improvements justify the additional computational complexity.
3. This study evaluates models with up to 8B parameters, but does not include state-of-the-art LLMs such as GPT-3 (175B), GPT-4, or the larger LLaMA model. Testing on larger models would better demonstrate the effectiveness of FEEDER.

**Questions For Authors:**

1. Does FEEDER maintain its effectiveness as dataset sizes increase, or does its selection quality degrade with larger datasets?
2. How would FEEDER perform on significantly larger models like GPT-4 or LLaMA-3 65B?

**Relation To Broader Scientific Literature:**

The key difference between this work and previous studies is the introduction of sufficiency and necessity metrics for data selection. This approach effectively identifies examples that are both representative and maximize information coverage, thereby improving the model's efficiency and performance.

**Theoretical Claims:**

I check the rationality of the theoretical claims, and I haven't found unreasonable aspects so far.

---

> ### Author Rebuttal · Authors · 2025-04-01
>
> We thank Reviewer mKZf for recognizing our novelty, acknowledging its efficiency and performance improvement, appreciating our comprehensive evaluations on multiple LLMs and tasks, and noting its compatibility with existing demonstration selection strategies.
>
> **Q1. More comparison beyound basic selection methods (Random, Similarity, and Diversity)**
>
> **R1.** Thank you for your issue. We have compared FEEDER with advanced ICL techniques, specifically including the uncertainty-based, the clustering-based, and the recent latent variable-based demonstration selection method proposed in [1]. This method is explicitly denoted as "Latent" in our paper (Section 5.1, line 317-319, Section A5.1, line 1186-1188 left).
>
> Corresponding experimental results are reported in Appendix Tables A2, A4, and A6, clearly demonstrating FEEDER’s performance relative to this advanced baseline.
>
> To further address your suggestion, we will highlight this comparison more explicitly in revision and include additional similarity-based methods.
>
> [1] Large Language Models Are Latent Variable Models: Explaining and Finding Good Demonstrations for In-Context Learning. NeurIPS 2023
>
> **Q2. While FEEDER shows performance improvements in most cases, the reported gains are not particularly significant.**
>
> We have added a significance test for the main results on the GSM8K dataset using the Llama-3 8B model in the ICL setting, as shown below. Results for other datasets and LLMs will be included in our revision.
>
> **Table R2**. * indicates *p* < 0.05 in significance tests compared to using TRAIN.
> | Model | Mode | n | GSM8K (Random) | GSM8K (Similarity) | GSM8K (Diversity) |
> |--|--|--|--|--|--|
> | Llama-3 8B | TRAIN | 1 | 78.24 | 79.56 | 79.56 |
> | Llama-3 8B | TRAIN | 2 | 79.55 | 83.40 | 83.67 |
> | Llama-3 8B | TRAIN | 5 | 81.45 | 83.47 | 84.52 |
> | Llama-3 8B | TRAIN | 10 | 82.31 | 84.42 | 84.53 |
> | Llama-3 8B | FEEDER | 1 | **80.23*** | **81.21*** | **81.21*** |
> | Llama-3 8B | FEEDER | 2 | **82.13*** | **84.43*** | **83.88** |
> | Llama-3 8B | FEEDER | 5 | **82.55*** | **85.03*** | **84.77** |
> | Llama-3 8B | FEEDER | 10 | **84.56*** | **85.79*** | **85.43*** |
>
> **Q3. Larger models (>8B)**
>
> **R3.** **We add **Qwen 2.5 32B model** as our base model on **GSM8K** and **GPQA**. Please refer to the R1 in response of Reviewer oHZY.** Results for the other benchmarks and LLMs will be included in the revision.
>
> **Q4. Whether these improvements justify the additional computational complexity.**
>
> Thanks for your question. We have included the runtime of running FEEDER in Figure A6. Running FEEDER only requires pre-running the inference over half of the number of training data points (33.45s for GPT-neo 1.3B on COLA dataset).
>
> For the fine-tuning, it costs 867.93s for training GPT-neo 1.3B on COLA dataset for one-epoch on the entire data points, whereas training GPT-neo 1.3B on COLA dataset on our FEEDER data costs 707.79s. As training is significantly more costly than inference, performing pre-selection is always cost-effective if around 10\% can be filtered out.
>
> Furthermore, we also would like to emphaize that our FEEDER operates at the pre-selection stage which allows for pre-computation over the entire training dataset, and serves various test data.
>
> **Q5. The concepts of sufficiency and necessity are somewhat abstract, and the paper could benefit from more intuitive, visual examples to help readers understand their impact on demonstration selection**
>
> **R5.** We have provided detailed descriptions of sufficiency and necessity, along with examples, in Appendix A2. Additionally, we also presented a case study in A11.2 to illustrate the impact of sufficiency and necessity on LLM performance.
>
> **Q6. Does FEEDER maintain its effectiveness and selection quality degrade as dataset sizes increase?**
>
> **R6.** Thanks for your question. To examine the impact of dataset size, we evaluate the effect of randomly reducing the SUBJ dataset by 20%, 40%, 60%, and 80% using the Similarity demonstration selector. Results are summarized below.
>
> **Table R3**.
> | Model | Mode | n | SUBJ (20%) | SUBJ (40%) | SUBJ (60%) | SUBJ (80%) | SUBJ (100%) |
> |--|--|--|--|--|--|--|--|
> | Llama-2 (7B) | TRAIN | 1 | 45.72 | 47.21 | 48.25 | 48.44 | 48.50 |
> | Llama-2 (7B) | TRAIN | 2 | 87.58 | 88.96 | 89.80 | 90.53 | 90.76 |
> | Llama-2 (7B) | TRAIN | 5 | 84.25 | 85.46 | 86.09 | 86.56 | 86.88 |
> | Llama-2 (7B) | TRAIN | 10 | 80.10 | 80.68 | 81.52 | 81.45 | 81.37 |
> | Llama-2 (7B) | FEEDER | 1 | **47.50*** | **48.95*** | **49.25*** | **49.50*** | **49.73*** |
> | Llama-2 (7B) | FEEDER | 2 | **89.80*** | **90.94*** | **91.06** | **92.17*** | **92.54*** |
> | Llama-2 (7B) | FEEDER | 5 | **86.28*** | **87.05*** | **87.56*** | **87.89*** | **87.95*** |
> | Llama-2 (7B) | FEEDER | 10 | **84.50*** | **85.14*** | **85.67*** | **85.88*** | **85.87*** |
>
> The results above demonstrate that our FEEDER is effective across different dataset scales.
>
> Please let us know if you have further questions -- thank you so much!

---

### Official Review · Reviewer_S3Tb · 2025-03-14

**Overall Recommendation:** 4

**Summary:**

The paper introduces FEEDER, a pre-selection framework designed to improve in-context learning (ICL) in large language models by identifying a representative subset of training data demonstrations. FEEDER uses "sufficiency" and "necessity" metrics to balance representativeness with redundancy, and employs a tree-based algorithm to efficiently identify the pre-selected examples. This pre-selection set replaces the full training set to improve example selection for ICL or for fine-tuning through a bi-level optimization method. Experimental validation across multiple different LLMs with sizes ranging from 300M to 8B parameters demonstrates that FEEDER can reduce training data size by over 20% while maintaining ICL performance, integrate with other ICL selection strategies for further improved performance, and improve fine-tuning performance compared to other methods. The authors also conduct several ablations on selection strategy and number of rounds to identify the optimal design.

## Update after rebuttal
The authors addressed most of my major concerns, so I have increased my score accordingly.

**Claims And Evidence:**

The paper claims that FEEDER can significantly reduce the size of training data required for selecting examples for ICL without compromising performance. These claims are substantiated by experiments showing consistent results across different model sizes and ICL strategies. However, it does not provide runtime comparisons, which would help to fully validate the efficiency improvements.

**Essential References Not Discussed:**

The paper does not sufficiently discuss its relationship with established active learning techniques, particularly around the concept of representativeness and informativeness, which have been studied for many years. This might obscure potential similarities and differences that could be crucial for contextualizing the methodological contributions.

**Experimental Designs Or Analyses:**

The experimental design appears sound, with FEEDER tested across a spectrum of LLMs and ICL strategies. The results are extensively documented, particularly in the appendix, which includes detailed methodology and additional results. However, the paper would be strengthened by including actual runtime data and comparisons to fine-tuning on the full dataset, which are crucial for evaluating the practical efficiency gains of the proposed approach (I imagine this is not the case, but if somehow full fine-tuning is faster than searching for this pre-selector set in a large dataset, the only benefit I see to this method is for improving proprietary model performance where fine-tuning cannot be done).

**Methods And Evaluation Criteria:**

The proposed methods and evaluation criteria make sense for the LLM ICL and fine-tuning setting. However, the work would benefit from benchmarking on other types of tasks besides text classification.

**Other Comments Or Suggestions:**

Table 2 should bold all highest numbers, not just highest that are their results.

**Other Strengths And Weaknesses:**

Strengths:
1. Novel approach for improving ICL in LLMS.
2/ Strong results across multiple LLM architectures, sizes, and datasets.
3. Solid theoretical contributions to back methodological design and results.
4. The paper is well-written and accessible.

Weaknesses:
1. The paper is not contextualized well within the active learning literature, which has explored similar ideas well before LLMs.
2. Performance of the approach is only benchmarked on text classification tasks.
3. No runtime numbers are presented which makes it difficult to assess the practicality of the method.

**Questions For Authors:**

1. Do the authors know why COLA+LAG+2 performance is worse with an increasing number of rounds?
2. How do runtimes compare to fine-tuning on the whole dataset?

Strong responses to the weaknesses as well as these questions could result in an improvement in my evaluation of the paper.

**Relation To Broader Scientific Literature:**

The key contributions of the paper have important implications on how to improve example selection for ICL in LLMs, which is a topic of broad interest in the AI community due to the massive increase in interest in and use of LLMs. Their proposed pre-selector method in combination with selection strategies is considerably better than naive selection strategies on the full dataset, which is a new finding that may be of interest in the field.

**Theoretical Claims:**

I briefly reviewed all three proofs which seemed correct but it is possible that I missed something.

---

> ### Author Rebuttal · Authors · 2025-04-01
>
> We thank Reviewer S3Tb for recognizing our novelty, solid theoretical contributions and good writing. Below, we respond to each of your questions in detail.
>
> **Q1. The paper is not contextualized well within the active learning literature.**
>
> **R1.** We clarify the relationship between our FEEDER method and active learning research line as follows:
>
> 1. Similarity: Both FEEDER and active learning methods focus on selecting representative and informative samples.
> 2. Difference: Active learning typically selects samples for annotation during training, while FEEDER selects a representative subset from already labeled data to reduce computation.
> 3. Our novelty: FEEDER proposes sufficiency and necessity metrics tailored specifically to LLMs, efficiently identifying critical demonstrations and removing redundancy.
>
> We listed some related active learning works discussed some of them in Appendix A1.3, we will highlight it and add more discussion in revision.
>
>
> **Q2. Performance of the approach is only benchmarked on text classification tasks.**
>
> **R2.**
> 1. As mentioned in our experiment section, in addition to 6 text classification tasks, **we have included the reasoning task GSM8K and the semantic parsing task SMCALFlow,** with corresponding results shown in Tables 2 and A6.
> 2. Furthermore, we add the science QA task GPQA in our revision. Partial results with the Qwen2.5 32 model are available in Table R1 below. These results further demonstrates FEEDER's ability to enhance LLM performance in math and science tasks.
>
> **Table R1**. Performance comparisons on GSM8K and GPQA datasets are conducted in the ICL setting. * indicates *p* < 0.05 in significance tests compared to using TRAIN.
> | Model | Mode | n | GSM8K (Random) | GSM8K (Similarly) | GSM8K (Diversity) | GPQA (Random) | GPQA (Similarly) | GPQA (Diversity) |
> |-----------------|--------|----|---------------|----------------|----------------|--------------|----------------|----------------|
> | Qwen 2.5 32B | TRAIN | 1 | 81.0 | 82.1 | 82.1 | 40.1 | 42.0 | 42.0 |
> | Qwen 2.5 32B | TRAIN | 2 | 83.2 | 84.2 | 84.7 | 42.0 | 44.5 | 45.2 |
> | Qwen 2.5 32B | TRAIN | 5 | 85.2 | 89.5 | 89.6 | 43.3 | 46.2 | 46.8 |
> | Qwen 2.5 32B | TRAIN | 10 | 86.2 | 90.4 | 89.9 | 43.2 | 46.8 | 46.7 |
> | Qwen 2.5 32B | FEEDER | 1 | **81.8** | **83.5*** | **83.5*** | **40.5** | **42.7** | **42.7** |
> | Qwen 2.5 32B | FEEDER | 2 | **84.5*** | **85.7*** | **86.0*** | **43.5*** | **45.8*** | **45.6** |
> | Qwen 2.5 32B | FEEDER | 5 | **86.7*** | **90.2** | **90.1** | **44.9*** | **48.0*** | **48.0*** |
> | Qwen 2.5 32B | FEEDER | 10 | **87.6*** | **91.2*** | **90.7** | **44.5*** | **47.8*** | **47.9*** |
>
>
> **Q3. No runtime numbers. How do runtimes compare to fine-tuning on the whole dataset?**
>
>
> **R3.**
>
> Thanks for your question. We have included the runtime of running FEEDER in Figure A6. Running FEEDER only requires pre-running the inference over half of number of training data points (33.45s for GPT-neo 1.3B on COLA dataset).
>
> For the fine-tuning, it costs 867.93s for training GPT-neo 1.3B on COLA dataset for one-epoch on the entire data points, whereas training GPT-neo 1.3B on COLA dataset on our FEEDER data costs 707.79s. As training is significantly more costly than inference, performing pre-selection is always cost-effective if around 10\% can be filtered out.
> Furthermore, we also would like to emphaize that our FEEDER operates at the pre-selection stage which allows for pre-computation over the entire training dataset, and serves various test data.
>
>
> **Q4. Do the authors know why COLA+LAG+2 performance is worse with an increasing number of rounds?**
>
> **R4.** We apologize for mixing the results of different demonstration retrievers when drawing this subfigure in Figure 3. The corrected results for COLA+LAG+2 should be 0.389 (for #Round = 0), 0.412 (for #Round = 1), 0.438 (for #Round = 5), and 0.430 (for #Round = 10), which follow the same trend as other settings.
>
> Please let us know if we have properly addressed your questions and we are more than happy to discuss more!

---

### Official Review · Reviewer_s2sv · 2025-03-14

**Overall Recommendation:** 3

**Summary:**

This submission presents a pre-selection framework for in-context learning designed to identify a representative subset of examples from the training set. The proposed framework FEEDER evaluates demonstration examples based on their sufficiency and necessity. Aside from benefiting in-context learning, the framework can also benefit model training, particularly by accelerating the fine-tuning process with the selected subset.

**Claims And Evidence:**

Yes, the claims made in the submission supported by clear and convincing evidence.

**Essential References Not Discussed:**

As mentioned in the "Experimental Designs Or Analyses" section above, the authors should include more discussion with *"Large Language Models Are Latent Variable Models: Explaining and Finding Good Demonstrations for In-Context Learning"*. This work is listed in the literature, but without thorough discussion/comparison.

**Experimental Designs Or Analyses:**

### Strengths
This submission conducts extensive experiments on several commonly used benchmarks and applies the proposed framework to various LLMs, including GPT-2/3, Gemma-2, and Llama-2. The experimental results demonstrate mostly consistent improvements, providing strong evidence of the framework's effectiveness.


### Weaknesses
This submission is closely related to the paper *"Large Language Models Are Latent Variable Models: Explaining and Finding Good Demonstrations for In-Context Learning" (NeurIPS 2024)*, with significant overlap in the selection of datasets and LLMs for the experiments. However, there is no direct comparison between the two methods regarding their effectiveness for ICL. It would be valuable to include a more detailed discussion and comparison between this submission and the related work.

**Methods And Evaluation Criteria:**

Yes, the proposed method and the evaluation criteria make sense for the problem.

**Other Comments Or Suggestions:**

* In Table 1, for the COLA results on Gemma-2, the 1-shot results under Similarity and Diversity should highlight the D_train values in bold for clarity.

**Other Strengths And Weaknesses:**

N/A

**Questions For Authors:**

N/A

**Relation To Broader Scientific Literature:**

This work builds upon existing research on enhancing ICL through improved sample selection. Its key contribution lies in introducing "sufficiency" and "necessity" metrics to guide the demonstration pre-selection process. Additionally, it proposes a novel tree-based algorithm to efficiently identify optimal demonstration examples, distinguishing it from prior approaches in the field.

**Theoretical Claims:**

No issue found.

---

> ### Author Rebuttal · Authors · 2025-04-01
>
> We thank Reviewer s2sv for recognizing our extensive experiments across multiple benchmarks and LLMs, showing consistent improvements of our framework.
>
> Below, we respond to each of your questions in detail.
>
> **Q1.1 The differences discussion between this submission and related work [1]. Some overlap in the selection of datasets and LLMs for the experiments.**
>
> **R1.1** Thanks for pointing it out. **Our approach differs significantly from theirs in both theoretical analysis and experimental implementation.**
>
> 1. Theoretically, their work investigates the causal relationship between the input X and the output Y, whereas ours analyzes the causal relationship between the input X and the selected demonstration C.
> 2. Empricially, their method operates at the demonstration selection stage, while our approach introduces a **pre-selection** stage before demonstration selection. This allows us to pre-compute a core subset of the training dataset as the demonstration selection pool.
> 3. Furthermore, their approach focuses on the demonstration selection stage and requires fine-tuning the embeddings of latent variables, which limits its applicability to smaller LLMs. In contrast, our method, which operates at the **pre-selection** stage, does not require fine-tuning LLMs.
>
> **Q1.2 No direct comparison and more detailed discussion between the two methods (FEEDER and [1]) regarding their effectiveness for ICL.**
>
> **R1.2** We directly compared FEEDER with [1] in our experiments, we have included [1] as one of the baseline demonstration selectors, note that we denoted method [1] as **"Latent"** (Section 5.1 line 317-319, Section A5.1. line 1186-1188 left). We provided the corresponding results **in Appendix Tables A1&A2, A3&A4, 2&A6**, main findings of effectiveness are listed below:
>
> 1. Comparing FEEDER (ours) + SUBJ (Similarity) against TRAIN + SUBJ (Latent [1]), we observed that applying a similarity-based demonstration selector on our FEEDER-generated core subset outperforms their advanced demonstration selector applied to the entire dataset. One possible explanation is that their method does not explicitly consider the relationships among candidate demonstrations.
> 2. Comparing FEEDER (ours) + SUBJ (Latent [1]) with FEEDER [1] + SUBJ (Similarity), we find that our method also improves LLM performance. However, the improvements brought by our method and theirs would have some overlap, as both approaches condition the selected demonstrations on the LLMs being used.
>
> **Q2. Clear writing sugguestion:** In Table 1, COLA results on Gemma-2, the 1-shot results under Similarity and Diversity should highlight the D_train values in bold.
>
> **R2.** Thanks for your suggestion. We will address these issues in our revision.
>
> **Reference:**
>
> [1] Large Language Models Are Latent Variable Models: Explaining and Finding Good Demonstrations for In-Context Learning. NeurIPS 2023
>
> Please let us know if you have further questions -- thank you so much!

---

> > ### Comment · Reviewer_s2sv · 2025-04-03
> >
> > Thank you for your responses. The authors have addressed my concerns. I keep my recommendation of weak accept.

---

> > > ### Author Response · Authors · 2025-04-07
> > >
> > > We are glad that all your concerns have been addressed! Thank you for supporting the acceptance of our paper.

---

### Official Review · Reviewer_oHZY · 2025-03-15

**Overall Recommendation:** 2

**Summary:**

This paper introduces FEEDER (FEw yet Essential Demonstration prE-selectoR), a novel pre-selection framework designed to improve In-Context Learning (ICL) and fine-tuning in large language models (LLMs). The key contribution of FEEDER is a pre-selection stage, where a representative subset of training data is selected based on two new metrics: sufficiency (how well a demonstration represents other samples) and necessity (whether removing a demonstration leads to loss of critical information). The paper proposes a tree-based algorithm to efficiently identify such representative subsets, reducing the computational cost of ICL while maintaining or even improving performance.

**Claims And Evidence:**

The claim of effectiveness is validated in the experiments.

**Essential References Not Discussed:**

NA

**Experimental Designs Or Analyses:**

Experiments are solid across various datasets.

**Methods And Evaluation Criteria:**

The method proposed is applicable to various tasks.

**Other Comments Or Suggestions:**

NA

**Other Strengths And Weaknesses:**

Strengths:

1.One of the major limitations of ICL is its high computational overhead, as each new query requires retrieving demonstrations from a large dataset. FEEDER reduces this cost without sacrificing accuracy, making ICL more practical for real-world deployment.

2.The sufficiency and necessity framework provides a formal and interpretable way to measure the contribution of a demonstration.

Weakness:

1.Limited Evaluation of Models: The work mainly considers several LLMs that are very small in parameter size, with the largest being 8B. This is particularly insufficient for the evaluation of ICL tasks, as the more practical LLMs are generally larger.

2.The paper writing could be further improved. There exists a lot of blank spaces in the paper between sections.
Lack of Evaluation of Tasks. For evaluation, the authors mainly consider 6 text classification tasks. However, many ICL methods are evaluated on a variety of tasks, including NLI or translation. The authors should consider adding more tasks.

**Questions For Authors:**

See weaknesses.

**Relation To Broader Scientific Literature:**

The proposed framework is inspiring and insightful for various tasks in the literature

**Theoretical Claims:**

No theoretical analysis.

---

> ### Author Rebuttal · Authors · 2025-04-01
>
> We would like to thank Reviewer oHZY for recognizing our method is efficient and practical for real-world deployment, as well as we proposing an interpretable way to explain the demonstration.
>
> Below, we respond to each of your questions in detail.
>
> **Q1. Should consider larger LLM.**
>
> **R1.** We add **Qwen 2.5 32B model** as our base model. The corresponding results on **GSM8K** and **GPQA** are listed below. Results for the other benchmarks and LLMs will be included in the revision.
>
> **Table R1**. Performance comparisons on GSM8K and GPQA datasets are conducted in the ICL setting. * indicates *p* < 0.05 in significance tests compared to using TRAIN.
> | Model | Mode | n | GSM8K (Random) | GSM8K (Similarly) | GSM8K (Diversity) | GPQA (Random) | GPQA (Similarly) | GPQA (Diversity) |
> |-----------------|--------|----|---------------|----------------|----------------|--------------|----------------|----------------|
> | Qwen 2.5 32B | TRAIN | 1 | 81.0 | 82.1 | 82.1 | 40.1 | 42.0 | 42.0 |
> | Qwen 2.5 32B | TRAIN | 2 | 83.2 | 84.2 | 84.7 | 42.0 | 44.5 | 45.2 |
> | Qwen 2.5 32B | TRAIN | 5 | 85.2 | 89.5 | 89.6 | 43.3 | 46.2 | 46.8 |
> | Qwen 2.5 32B | TRAIN | 10 | 86.2 | 90.4 | 89.9 | 43.2 | 46.8 | 46.7 |
> | Qwen 2.5 32B | FEEDER | 1 | **81.8** | **83.5*** | **83.5*** | **40.5** | **42.7** | **42.7** |
> | Qwen 2.5 32B | FEEDER | 2 | **84.5*** | **85.7*** | **86.0*** | **43.5*** | **45.8*** | **45.6** |
> | Qwen 2.5 32B | FEEDER | 5 | **86.7*** | **90.2** | **90.1** | **44.9*** | **48.0*** | **48.0*** |
> | Qwen 2.5 32B | FEEDER | 10 | **87.6*** | **91.2*** | **90.7** | **44.5*** | **47.8*** | **47.9*** |
>
> These results demonstrate FEEDER can be extended to LLMs larger than 8B.
>
>
> **Q2. Paper writing issue: lot of blank spaces between sections.**
>
> **R2.** Thanks for your suggestion. We will address these issues in our revision.
>
> **Q3. Lack of Evaluation of Tasks. Authors should consider ICL methods on variety tasks, now only consider 6 text classification tasks.**
>
> **R3.** As mentioned in our experiment section, in addition to 6 text classification tasks, we have included the reasoning task GSM8K and the semantic parsing task SMCALFlow, with corresponding results shown in Tables 2 and A6 in the paper.
>
> Furthermore, we add the science QA task GPQA in our revision. Partial results with the Qwen2.5 32B model are available in Table R1 above, which further demonstrates FEEDER's ability to enhance LLM performance in math and science tasks.
>
> We are eager to hear your feedback. We’d deeply appreciate it if you could let us know whether your concerns have been addressed.

---

### Decision · Program_Chairs · 2025-05-01

**Decision:**

Accept (poster)

**Comment:**

This paper addresses the inefficiency of standard few-shot in-context learning (ICL). The authors propose FEEDER, a framework that performs a pre-selection step before ICL inference. FEEDER identifies a core subset of the training data (the "FEEDER set") containing the most informative demonstrations. Informativeness is defined using two novel metrics: sufficiency (how well a demonstration represents others) and necessity (the information loss if a demonstration is removed). A tree-based algorithm is proposed to efficiently find this FEEDER set. Once identified, this smaller FEEDER set replaces the original large dataset as the pool from which demonstrations are selected during ICL inference, leading to improved efficiency and potentially better accuracy.

Overall, the paper received a mixed review. On the positive side, reviewers acknowledged the importance of the problem, the soundness and relevance of the sufficiency/necessity framework, sensible method and convincing evaluation. On the negative sides, reviewers questioned the limited evaluation on task/models (addressed in the rebuttal to a large extent), comparison to important baselines (again, the authors mentioned that some of the baselines requested are already in the paper, and they clarified the relationship of FEEDER w.r.t. active learning) -- the negative reviewers did not respond to the rebuttal, and the AC has checked the authors' response and agree that their original concerns have been largely addressed. As such, the AC recommends acceptance, and encourages the authors to incorporate all promised changes into the camera ready version of the paper. On a side note, the authors should also consider incorporating some qualitative discussions of some more recent works discussing similar phenomenon (e.g., that a small set of demonstrations can perform on par with a large number of examples) like [1] published in ICLR 2025.

[1] Wan, X., Zhou, H., Sun, R., Nakhost, H., Jiang, K., & Arık, S. Ö. (2025). From Few to Many: Self-Improving Many-Shot Reasoners Through Iterative Optimization and Generation. arXiv preprint arXiv:2502.00330.